# Uniform-in-time propagation of chaos for the mean-field gradient Langevin dynamics

**Taiji Suzuki**
The University of Tokyo
RIKEN Center for Advanced Intelligence Project
taiji@mist.i.u-tokyo.ac.jp

**Atsushi Nitanda**
Kyushu Institute of Technology
RIKEN Center for Advanced Intelligence Project
nitanda@ai.kyutech.ac.jp

**Denny Wu**
The University of Toronto
Vector Institute for Artificial intelligence
dennywu@cs.toronto.edu

## ABSTRACT

The mean-field Langevin dynamics is characterized by a stochastic differential equation that arises from (noisy) gradient descent on an infinite-width two-layer neural network, which can be viewed as an interacting particle system. In this work, we establish a quantitative weak propagation of chaos result for the system, with a finite-particle discretization error of $\mathcal{O}(1/N)$ *uniformly over time*, where $N$ is the width of the neural network. This allows us to directly transfer the learning guarantee for infinite-width networks to practical finite-width models without excessive overparameterization. On the technical side, our analysis differs from most existing studies on similar mean field dynamics in that we do not require the interaction between particles to be sufficiently weak to obtain a uniform propagation of chaos, because such assumptions may not be satisfied in neural network optimization. Instead, we make use of a logarithmic Sobolev-type condition which can be verified in appropriate regularized risk minimization settings.

## 1 INTRODUCTION

**Mean-field neural networks.** We consider the optimization of a two-layer neural network in the *mean-field regime*, which is represented as an average over $N$ neurons:

$$f_{\boldsymbol{X}}(z) = \frac{1}{N} \sum_{i=1}^{N} h_z(x_i),$$

where given the input $z \in \mathbb{R}^{d'}$, each neuron computes a nonlinear transformation based on trainable parameters $x \in \mathbb{R}^d$; for example, we may set $h_z(x) = \tanh(w^\top z + b)$ for $x = (w, b) \in \mathbb{R}^{d'+1}$.

Importantly, the mean-field parameterization allows for the parameters to move away from initialization during gradient descent and hence learn informative features (Yang and Hu, 2020) even when the network width is large ($N \to \infty$), in contrast to the *Neural Tangent Kernel* (NTK) parameterization (Jacot et al., 2018) (corresponding to a $1/\sqrt{N}$ prefactor), which freezes the model at initialization under overparameterization. This *feature learning* ability enables mean-field neural networks to outperform the NTK counterpart (or linear estimators in general) in learning a wide range of target functions (Ghorbani et al., 2019; Li et al., 2020; Abbe et al., 2022; Ba et al., 2022).

Optimization guarantees for mean-field neural networks are typically obtained by lifting the finite-width model to the infinite-dimensional space of parameter distributions and then exploiting convexity of the objective function. Using this viewpoint, convergence of gradient flow on infinite-width neural networks to the global optimal solution can be shown under appropriate conditions (Nitanda and Suzuki, 2017; Chizat and Bach, 2018; Mei et al., 2018; Rotskoff and Vanden-Eijnden, 2018; Sirignano and Spiliopoulos, 2020). However, most existing results are *qualitative* in nature, in that they do not characterize the rate of convergence and the finite-particle discretization error.

**Mean-field Langevin dynamics.** An often-studied optimization method for mean-field neural networks is the noisy particle gradient descent (NPGD) algorithm (Mei et al., 2018; Hu et al., 2019; Chen et al., 2020b), where Gaussian noise is injected to the gradient to encourage "exploration" and enable global optimality to be shown under milder conditions than the noiseless case. The large particle and vanishing step size limit is termed the *mean-field Langevin dynamics* (Hu et al., 2019), which globally minimizes an entropy-regularized convex functional in the space of measures.

Recently, Nitanda et al. (2022); Chizat (2022) established exponential convergence for the mean-field Langevin dynamics under certain logarithmic Sobolev inequalities which can be easily verified in regularized risk minimization problems using two-layer neural networks (1). This represents a significant step towards a *quantitative* optimization analysis of neural networks in the presence of feature learning, yet the limitation is also clear: these results are obtained from directly analyzing the large particle limit (i.e., the limiting McKean-Vlasov stochastic differential equation), and cannot be easily transferred to practical finite-width networks. In fact, naively applying the quantitative results in Mei et al. (2018; 2019) leads to discretization error bounds that blow up exponentially in time, rendering the guarantee vacuous beyond the very early stages of gradient descent learning. Therefore, for the purpose of characterizing the optimization behavior of finite-width neural networks, it is important to derive a finite-particle discretization error bound that holds *uniformly over time*, that is, the error remains stable even when $t$ is large.

## 1.1 OUR CONTRIBUTIONS

In this paper, we establish finite-particle guarantees for the mean-field Langevin dynamics via a *propagation of chaos* calculation (Sznitman, 1991) which controls the weak error between the empirical distribution of the interacting particle system and the corresponding infinite particle limit along the optimization trajectory. This allows us to bound the difference in the function value between the finite-width neural network optimized by NPGD and the infinite-width counterpart.

In particular, starting from $N$ initialized particles $\boldsymbol{X}_0 \overset{\text{i.i.d}}{\sim} \mu_0$, if we denote the finite-particle model at time $t$ of optimization as $f_{\boldsymbol{X}_t}$, and its corresponding infinite-particle limit as $f_{\mu_t}$, then our propagation of chaos result is the following.

**Theorem** (informal). *Under suitable regularity conditions,* $\mathbb{E}\big[(f_{\boldsymbol{X}_t}(z) - f_{\mu_t}(z))^2\big] = \mathcal{O}(1/N)$ *for any $t > 0$ and $z \in \mathbb{R}^{d'}$.*

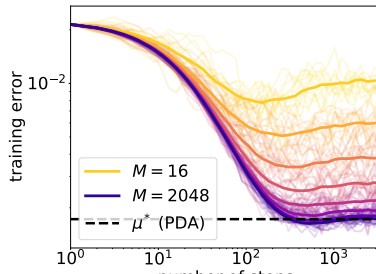
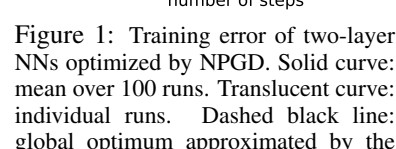

Figure 1: Training error of two-layer NNs optimized by NPGD. Solid curve: mean over 100 runs. Translucent curve: individual runs. Dashed black line: global optimum approximated by the PDA algorithm (Nitanda et al., 2021).

We make the following remarks on the main theorem:

- To our knowledge, we provide the first rigorous *uniform-in-time* propagation of chaos result in the context of mean-field neural networks. This is in contrast to prior works where the discretization error typically increases as optimization proceeds (e.g., $|f_{\boldsymbol{X}_t} - f_{\mu_t}| = \mathcal{O}(\exp(t) \cdot N^{-1/2})$ as in Mei et al. (2018, Theorem 3)). The theorem implies that as the width $N$ becomes larger, the difference between the finite-width and infinite-width model output diminishes rapidly, as shown in Figure 1.

- Our analysis assumes a modified Log-Sobolev condition which is satisfied in regularized risk minimization problems using neural network when the convex regularizer on the parameters has super-quadratic tail. Noticeably, we do not impose any constraint on the strength of regularization and interaction; this differs from many existing results where uniform propagation of chaos is only achieved under weak interaction or large noise (Eberle et al., 2019; Delarue and Tse, 2021).

Due to the space constraint, we defer discussions on additional related works to Appendix A.

## 2 PRELIMINARIES

In this section, we formulate the problem setting and introduce some useful notations for the following sections. We optimize a two-layer neural network by minimizing the empirical or expected risk in a supervised learning setting, where the input is included in a set $\mathcal{Z} \subset \mathbb{R}^{d'}$ and the output is in a bounded set $\mathcal{Y} \subset \mathbb{R}$. As defined in the Introduction, $h_{(\cdot)}(x) : z \in \mathcal{Z} \mapsto h_z(x) \in \mathcal{Y}$ represents

one neuron (particle) with parameters $x \in \mathbb{R}^d$, and the mean-field neural network is written as the average over $N$ neurons: $f_{\boldsymbol{X}}(z) = \frac{1}{N} \sum_{i=1}^{N} h_z(x_i)$, where $\boldsymbol{X} = (x_i)_{i=1}^{N} \subset \mathbb{R}^d$ denotes the collection of parameters and $z \in \mathcal{Z}$. The continuous limit of the neural network is obtained by taking $N \to \infty$, and in analogy to the law of large numbers, $f_{\boldsymbol{X}}$ converges to the following integral form:

$$f_\mu(z) = \int h_z(x) \mathrm{d}\mu(x),$$

where $\mu$ is a probability measure on $(\mathbb{R}^d, \mathcal{B}(\mathbb{R}^d))$ representing the weight of each parameter. Let $\mathcal{P}$ be the set of probability measures on $(\mathbb{R}^d, \mathcal{B}(\mathbb{R}^d))$, and $\mathcal{P}_p$ be those with a finite $p$-th moment ($p \geq 1$). As typical for the mean-field analysis, we aim to optimize the density function $\mu$ so that the neural network $f_\mu$ accurately predicts the output $y \in \mathcal{Y}$ from the input $z \in \mathcal{Z}$.

In the following, we take (regularized) empirical risk minimization as a concrete example, and we note that the exact same analysis applies to the minimization of expected risk. Let $\ell(z, y) : \mathcal{Y} \times \mathcal{Y} \to \mathbb{R}$ be a convex loss function, such as the squared loss $\ell(z, y) = (z - y)^2/2$ for regression, or the logistic loss $\ell(z, y) = \log(1 + \exp(-yz))$ for classification. For each $(z_i, y_i)$ in the given training data $(z_i, y_i)_{i=1}^{n} \subset \mathcal{Z} \times \mathcal{Y}$, we use the notation $h_i(x)$ and $\ell_i(f)$ to indicate $h_{z_i}(x)$ and $\ell(f(z_i), y_i)$ respectively. Our goal is to find an *approximate* minimizer of the following objective over $\mathcal{P}$:

$$F(\mu) := \frac{1}{n} \sum_{i=1}^{n} \ell_i(f_\mu) + \lambda_1 \int r(x) \mathrm{d}\mu(x), \tag{1}$$

where $\lambda_1 > 0$ is the regularization strength and $r(\cdot)$ is a convex regularizer. More specifically, we will analyze the mean-field Langevin dynamics which solves an *entropy-regularized* version of (1). It is worth noting that this entropy-regularized objective can also be globally optimized by the recently proposed particle gradient-type methods in Nitanda et al. (2021); Oko et al. (2022), for which finite-width convergence rates have been provided. However, those methods employ an intricate double-loop structure which does not mirror the commonly-used gradient descent algorithm. Therefore, an important question to be addressed is whether noisy gradient descent also enjoys similar quantitative convergence guarantee — this is precisely the motivation of the current paper.

## 3 MEAN-FIELD GRADIENT LANGEVIN DYNAMICS

**Derivation of the continuous dynamics.** The basic idea of the mean-field Langevin dynamics is to optimize the aforementioned objective via *Wasserstein gradient flow* over a set of measures $\mathcal{P}$. To define the gradient with respect to the measure, we introduce the *first-variation* $\frac{\delta G}{\delta \mu}$ of a functional $G : \mathcal{P}_2 \to \mathbb{R}$ at $\mu \in \mathcal{P}_q$ (for a given $q \geq 1$) as a continuous functional $\mathcal{P}_q \times \mathbb{R}^d \to \mathbb{R}$ that satisfies

$$\lim_{\epsilon \to 0} \frac{G(\epsilon \nu + (1 - \epsilon)\mu)}{\epsilon} = \int \frac{\delta G}{\delta \mu}(\mu)(x) \mathrm{d}(\nu - \mu),$$

for any $\nu \in \mathcal{P}_q$. If there exists such a functional $\frac{\delta G}{\delta \mu}$, we say $G$ admits a first-variation at $\mu$, or simply $G$ is differentiable at $\mu$. To avoid the ambiguity of $\frac{\delta G}{\delta \mu}$ up to constant shift, we follow the convention of imposing $\int \frac{\delta G}{\delta \mu}(\mu) \mathrm{d}\mu = 0$. In our setting, the first-variation of the objective $F$ is given by

$$\frac{\delta F}{\delta \mu}(\mu)(x) = \frac{1}{n} \sum_{j=1}^{n} \ell_j'(f_\mu) h_j(x) + \lambda_1 r(x).$$

We track $F(\mu_t)$ along a trajectory of measures $(\mu_t)_t$ in $\mathcal{P}_2$ following a *continuity equation*:

$$\partial_t \mu_t = \nabla \cdot (\mu_t v_t),$$

where $v_t : \mathbb{R}^d \to \mathbb{R}^d$ is a vector field included in $L^2(\mu_t)$, and the time-derivative and the divergence operator are defined in a *weak sense*, that is, for any continuously differentiable function $\phi$ with a compact support, $\int \phi \mathrm{d}\mu_t - \int \phi \mathrm{d}\mu_s = -\int_s^t \int \nabla \phi \cdot v_\tau \mathrm{d}\mu_\tau \mathrm{d}\tau$. Then, the time-derivative of $G(\mu_t)$ can be written as

$$\partial_t G(\mu_t) = \int v_t \cdot \nabla \frac{\delta}{\delta \mu} G(\mu_t) \mathrm{d}\mu_t. \tag{2}$$

We refer readers to Villani (2009); Ambrosio et al. (2005); Bakry et al. (2014) for more details. In this sense, $\nabla \frac{\delta}{\delta \mu} F(\mu_t)$ can be seen as a *gradient direction* in the measure space (endowed with a

Wasserstein metric). The mean-field Langevin dynamics approximately minimizes the objective $F$ based on the Wasserstein gradient flow. Specifically, define the nonlinear drift term:

$$b(x, \mu) = \nabla \frac{\delta F}{\delta \mu}(\mu)(x) = \frac{1}{n} \sum_{j=1}^{n} \ell_j'(f_\mu) \nabla h_j(x) + \lambda_1 \nabla r(x),$$

the *mean-field Langevin dynamics* is then given by the following stochastic differential equation:

$$\mathrm{d}X_t = -b(X_t, \mu_t)\mathrm{d}t + \sqrt{2\lambda}\mathrm{d}W_t, \tag{3a}$$
$$\mu_t = \mathrm{Law}(X_t), \tag{3b}$$

for $X_0 \sim \mu_0$, where $\mathrm{Law}(X)$ denotes the distribution (probability law) of the random variable $X$, and $(W_t)_{t \geq 0}$ is the $d$-dimensional standard Brownian motion. The existence and uniqueness of the solution are ensured by Theorem 3.3 of Huang et al. (2021) (see also Corollary 3 for more details).

For concise presentation in the subsequent analysis, we follow the notation in (Delarue and Tse, 2021) and denote the law $\mu_t$ in the mean-field Langevin dynamics with the initial condition $\mu_0$ as $\mu_t = m(t, \mu_0)$. It is known that $\mu_t$ satisfies the following *nonlinear* Fokker-Planck equation:

$$\partial_t m(t, \mu_0) = \lambda \Delta m(t, \mu_0) + \nabla \cdot [m(t, \mu_0)b(\cdot, m(t, \mu_0))], \tag{4}$$

with $m(0, \mu_0) = \mu_0$ (this is again defined in a weak sense, that is, $\int \phi \mathrm{d}(m(t, \mu_0) - m(s, \mu_0)) = \int_s^t \int (\lambda \Delta \phi - b(\cdot, m(\tau, \mu_0))^\top \nabla \phi) \mathrm{d}m(\tau, \mu_0) \mathrm{d}\tau$ for smooth test function $f$ with compact support). This dynamics is an example of distribution dependent SDEs originating from the study of interacting particle systems which dates back to 1950s (Kahn and Harris, 1951; Kac, 1956; 1959; McKean, 1966; 1967). A fundamental characterization of the mean-field Langevin dynamics is that it is a Wasserstein gradient flow that minimizes the following objective (Mei et al., 2018; Hu et al., 2019):

$$\mathcal{L}(\mu) = F(\mu) + \lambda \mathrm{Ent}(\mu), \tag{5}$$

where $\mathrm{Ent}(\mu) = -\int \log(\mathrm{d}\mu(z)/\mathrm{d}z)\mathrm{d}\mu(z)$ is the negative entropy of $\mu$. Indeed, it is known that $\nabla \frac{\delta \mathcal{L}(\mu)}{\delta \mu} = \nabla \frac{\delta F}{\delta \mu} + \lambda \nabla \log(\mu) = \lambda \nabla \log(\mu) + b(\cdot, \mu_t)$ (e.g, Theorem 4.16 of Ambrosio et al. (2005)) and the continuity equation corresponding to $\mu_t$ can be rewritten as $\partial_t \mu_t = \nabla \cdot [(\lambda \nabla \log(\mu_t) + b(\cdot, \mu_t))\mu_t] = \nabla \cdot (\nabla \frac{\delta \mathcal{L}(\mu)}{\delta \mu} \mu_t)$, which, in combination with the identity (2), yields that

$$\frac{\mathrm{d}}{\mathrm{d}t}\mathcal{L}(\mu_t) = -\int \left\| \nabla \frac{\delta \mathcal{L}(\mu)}{\delta \mu} \right\|^2 \mathrm{d}\mu_t.$$

We therefore see that $\mu_t$ decreases $\mathcal{L}(\mu_t)$ unless $\frac{\delta \mathcal{L}(\mu)}{\delta \mu} = 0$, which is a crucial property that guarantees the convergence of $\mu_t$ to the global optimal solution (Lemma 1). This can be seen as a nonlinear extension of the usual gradient Langevin dynamics (e.g., see Bakry et al. (2014)), where $F(\mu)$ is a linear functional in the form of $F(\mu) = \int L(x)\mathrm{d}\mu$ with some objective function $L$.

It is easy to see that the objective $\mathcal{L}$ can be reformulated as the following objective (up to constant) that employs the KL divergence from a distribution characterized by the regularization term $r$:

$$\mathcal{L}(\mu) = \frac{1}{n} \sum_{i=1}^{n} \ell_i(f_\mu) + \lambda \mathrm{KL}(\mu, \nu_r), \tag{6}$$

where $\nu_r$ is a distribution with density proportional to $\exp(-\lambda_1 r/\lambda)$ and $\mathrm{KL}(\cdot, \cdot)$ is the KL divergence (relative entropy) defined as $\mathrm{KL}(\mu, \nu) := \int \log(\mathrm{d}\mu/\mathrm{d}\nu)\mathrm{d}\mu$.

**Particle discretization.** One of the main difficulties to simulate the mean-field Langevin dynamics is that we cannot access the exact information of $\mu_t$ in the practical setting. Instead, we approximate this infinite-dimensional objective by a finite set of particles, which yields the following SDE:

$$\mathrm{d}\hat{X}_t^i = -b(\hat{X}_t^i, \mu_t^N)\mathrm{d}t + \sqrt{2\lambda}\mathrm{d}W_t^i, \quad \mu_t^N = \frac{1}{N} \sum_{i=1}^{N} \delta_{\hat{X}_t^i}, \tag{7}$$

and $\hat{X}_0^i \sim \mu_0$. Our goal is to quantify the finite-particle approximation error due to replacing $\mu_t$ with $\mu_t^N$. The main mechanism of this approximation is the *propagation of chaos* (Sznitman, 1991),

which roughly refers to the phenomenon that as the number of particles $N \to \infty$, the correlation between particles vanishes and $\mu_t^N \to \mu_t$. However, it is far from trivial to establish a *quantitative* estimate of such approximation that provides a meaningful guarantee for the finite-particle system. In the following sections, we will show that under appropriate conditions, the finite-particle error can be (weakly) controlled uniformly over time, which allows us to transfer learning guarantees in the mean-field limit to finite-width neural networks that are not excessively overparameterized.

## 4 MAIN ASSUMPTIONS AND LOGARITHMIC SOBOLEV INEQUALITY

In this section we present our main theoretical result – the quantitative propagation of chaos. First, we introduce the main assumption in our analysis.

**Assumption 1.** *We assume $h_i, \ell_i, r \in \mathcal{C}^\infty$ and satisfy the following conditions:*

1. ***Convexity of loss:*** *$\ell_i$ is a convex function.*

2. ***Boundedness and smoothness:*** *There exists $B > 0$ such that $\|h_i\|_\infty \leq B$, $\|\nabla h_i\|_\infty \leq B$, $\|\nabla \nabla^\top h_i\|_\infty \leq B$, $\max\{|\ell_j(f_\mu)|, |\ell_j'(f_\mu)|, |\ell_j''(f_\mu)|\} \leq B$ uniformly over $\mathcal{P}$.*

3. ***Regularity of $r$:*** *The regularization term $r$ is a convex function satisfying $c_r \|x\|^{2+\delta} \leq r(x) \leq C_r(1 + \|x\|^{2+\delta})$[1], $\nabla r(x) \cdot x \geq c_r \|x\|^{2+\delta}$ and $0 \preceq \nabla \nabla^\top r(x) \preceq C_r(1 + \|x\|^\delta)I$ for constants $0 < \delta$ and $c_r, C_r > 0$.*

We make the following remarks on the assumptions.

- The loss convexity is a standard assumption to ensure that the objective $\mathcal{L}$ is convex with respect to $\mu$ (note that this does not imply convexity with respect to the parameters $\{x_i\}_{i=1}^N$ of the network).
- The second assumption is satisfied for standard two-layer models under the following conditions: $(i)$ $\|z\| \leq C$; $(ii)$ smooth loss function, such as the squared loss and logistic loss. For example, we may set $h_z(x) = \tanh(r\sigma(w^\top z))$ with smooth activation function $\sigma$ and $x = (r, w)$, or $h_z(x) = \sigma(w^\top z + b)$ with smooth and bounded activation and $x = (w, b)$.
- The constraint on $r(\cdot)$ requires the regularization term to have a super-quadratic tail, which is satisfied, for example, by $r(x) = \|x\|^4$. While this does not cover the standard weight decay, we note that such regularizers with stronger tail growth have been employed in the theoretical analysis of neural networks (Chen et al., 2020a; Allen-Zhu and Li, 2022). The purpose of this assumption is to ensure good isoperimetry of $\mu_t$ along the trajectory (see Corollary 1).
- The infinite differentiability condition is imposed only for the simplicity of our analysis.

**Proximal Gibbs distribution.** An important quantity in the convergence analysis is the *proximal Gibbs distribution*: for $\mu \in \mathcal{P}$, we define the proximal density function as

$$p_\mu(x) = \frac{1}{Z(\mu)} \exp\left(-\frac{1}{\lambda} \frac{\delta F(\mu)}{\delta \mu}(x)\right),$$

where $Z(\mu)$ is the normalization constant. One may check that this corresponds to the minimizer of the *linearized potential*: $\min_{\nu \in \mathcal{P}} \int \frac{\delta F(\mu)}{\delta \mu} d\nu + \lambda \mathrm{Ent}(\nu)$. For a given $\mu_t$, we denote by $\tilde{\mu}_t$ its proximal Gibbs measure, that is, the probability measure with the density $p_{\mu_t}$. Then, we have the following characterization of the minimizer of $\mathcal{L}$.

**Proposition 1.** *Under Assumption 1, the functional $\mathcal{L}$ has a unique minimizer in $\mathcal{P}$ that is absolutely continuous with respect to the Lebesgue measure. Moreover, $\mu^* \in \mathcal{P}_{2+\delta}$ is the optimal solution if and only if $\mu^*$ is absolutely continuous and its density function is given by $p_{\mu^*}$.*

This proposition can be shown in the same manner as Proposition 2.5 of Hu et al. (2019). We remark that although this prior result assumed $r$ to have at most quadratic growth, its proof does not require such growth condition but requires only the integrability of $\nu_r$ and $\nu_r(x) \log(\nu_r(x))$.

Many convergence properties of the mean-field Langevin dynamics can be characterized by properties of $\mu^*$ and the proximal Gibbs distribution $\tilde{\mu}_t$. We first introduce the *logarithmic Sobolev inequality* (LSI) which will be very useful in the subsequent analysis.

---

[1]This condition can be easily relaxed to $c_r \|x\|^{2+\delta} \leq r(x) \leq C_r(1 + \|x\|^{2+\delta'})$ with $\delta \neq \delta' > 0$. Here we consider $\delta = \delta'$ just for simplicity of presentation.

**Definition 1** (Logarithmic Sobolev inequality). *Let $p(\theta)$ be a smooth probability density function on $\mathbb{R}^d$. $p(\theta)$ (or its corresponding probability measure on $(\mathbb{R}, \mathcal{B}(\mathbb{R}^d))$) satisfies the LSI with constant $\alpha' > 0$ if and only if, for any smooth function $\phi : \mathbb{R}^d \to \mathbb{R}$ with $\mathbb{E}_p[\|\phi\|^2] < \infty$, it holds that*

$$\mathbb{E}_p[\phi^2 \log(\phi^2)] - \mathbb{E}_p[\phi^2] \log(\mathbb{E}_p[\phi^2]) \leq \frac{2}{\alpha'} \mathbb{E}_p[\|\nabla\phi\|_2^2].$$

We can verify that in our setting, the proximal Gibbs measure satisfies the LSI condition.

**Proposition 2.** *Under Assumption 1, $p_\mu$ satisfies the log-Sobolev inequality with a constant $\alpha$ that depends on $d, c_r, B, \lambda, \delta$. If additionally $\nabla\nabla^\top r \succeq I$, then the LSI holds with $\alpha = \frac{2\lambda_1}{\lambda} \exp\left(-\frac{4B}{\lambda}\right)$.*

The proof is given in Corollary 5 in the Appendix. Note what similar characterization was obtained in Nitanda et al. (2022); Chizat (2022) under a quadratic regularizer $r(x) = \|x\|^2$, via the standard Bakry–Emery and Holley–Stroock arguments (Bakry and Émery, 1985; Holley and Stroock, 1987) (see also Corollary 5.7.2 and 5.1.7 of Bakry et al. (2014)). However, our Assumption 1 does not entail $\nabla\nabla^\top r \succeq I$ and thus our proof follows a different strategy. This LSI condition is crucial to the geometric ergodicity of the mean-field Langevin dynamics described in Theorem 1 below.

## 5 CONVERGENCE GUARANTEE FOR FINITE-WIDTH NEURAL NETWORKS

To present our (weak) convergence result, we first introduce an objective function in the form of

$$\mathcal{U}(t, \mu) := \Phi(m(t, \mu)),$$

where $\Phi : \mathcal{P} \to \mathbb{R}$ is assumed to be sufficiently smooth, that is, $\Phi$ is twice differentiable with respect to $\mu$ and $x$, and the derivatives are bounded as $\sup_{x_1,\ldots,x_k \in \mathbb{R}^d} |\partial_{x_1}^{j_1} \ldots \partial_{x_k}^{j_k} \frac{\delta^k \Phi(\mu)}{\delta\mu^k}(x_1, \ldots, x_k)| < C$ for $k = 0, 1, 2$ and $j_i = 0, 1, 2$ uniformly over all $\mu \in \mathcal{P}$ with some constant $C$ (see Delarue and Tse (2021) for related definition).

**Example 1.** *Under Assumption 1, we allow for the following objective functions.*

*(i)* **Neural network function value:** *$\Phi(\mu) = \int h_z(x)\mathrm{d}\mu(x)$ with a fixed $z \in \mathcal{Z}$.*

*(ii)* **Training and test error:** *$\Phi(\mu) = \mathbb{E}_{(Z,Y)\sim P}[\ell(f_\mu(Z), Y)]$ where $P$ is a distribution on $\mathcal{Z} \times \mathcal{Y}$. For a smooth loss $\ell$, $\Phi$ satisfies the smoothness condition. If $P = \frac{1}{n}\sum_{i=1}^n \delta_{(z_i,y_i)}$, then $\Phi$ is the training loss, and if $P$ is the test distribution, it is the test loss.*

We proceed by bounding the weak difference between the finite-particle system at time $t$ and the optimal $\mu^*$ (see Proposition 1): $\mathbb{E}[\Phi(\mu_t^N)] - \Phi(\mu^*)$. We utilize the following decomposition:

$$\mathbb{E}[\Phi(\mu_t^N)] - \Phi(\mu^*) = \underbrace{\mathbb{E}[\mathcal{U}(t, \mu_0^N) - \Phi(\mu^*)]}_{\text{(I), ergodicity term}} + \underbrace{\mathbb{E}[\mathcal{U}(0, \mu_t^N) - \mathcal{U}(t, \mu_0^N)]}_{\text{(II), propagation of chaos term}}, \qquad (8)$$

where the ergodicity term (I) monitors the convergence of the infinite-particle dynamics (3) (starting from $\mu_0^N$) to the optimal solution $\mu^*$, and the propagation of chaos term (II) controls the fluctuation due to the finite-particle update (4). The two terms are bounded separately in the ensuing subsections. Note that while we focus on two-layer neural networks under Assumption 1, the same computation can be performed under certain isoperimetric conditions on the trajectory. In particular,

- Analysis of the **ergodicity term (I)** only requires the proximal Gibbs measure $\tilde{\mu}_t$ to satisfy the LSI. This condition can be easily verified for both quadratic regularizer as in (Nitanda et al., 2022; Chizat, 2022) and super-quadratic regularizers as shown in Proposition 2.

- To control the **propagation of chaos term (II)**, we require an LSI condition on $\mu_t$ along the trajectory. Similar assumption also appeared in Lacker and Flem (2022) to obtain a uniform-in-time evaluation, and is very challenging to establish in the mean-field neural network setting. We prove this assumption by transferring the LSI constant from $\tilde{\mu}_t$ to $\mu_t$ via a *super LSI* condition, which is verified under the super-quadratic regularization in Assumption 1 (see Lemma 2).

### 5.1 BOUNDING THE ERGODICITY TERM (I)

Let $W_p(\mu, \nu)$ denote the $p$-Wasserstein distance between $\mu, \nu \in \mathcal{P}_p$. We first show that $m(t, \mu_0^N)$ converges to $\mu^*$ in an exponential order (geometric ergodicity) in the following sense.

**Lemma 1.** *Under Assumption 1, for $\mu_t = m(t, \mu_0^N)$, it holds that $\mathcal{L}(\mu_t) < \infty$ for any $t > 0$, and*

$$\mathcal{L}(\mu_t) - \mathcal{L}(\mu^*) \leq \exp(-2\alpha\lambda(t - \tau_0))\lambda\psi_2(\tau_0)W_2(\mu_0, \mu^*)^2,$$

$$\mathrm{KL}(\mu_t, \mu^*) \leq \exp(-2\alpha\lambda(t - \tau_0))\psi_2(\tau_0)W_2(\mu_0, \mu^*)^2,$$

*for any $t > \tau_0$ where $\tau_0 > 0$ is arbitrary, $\psi_2(t) = \frac{1}{2\lambda}\left(\frac{B^2}{1-\exp(-B^2t)} + \frac{tB^2\exp(4B^2t)}{2}\right)$, and $\alpha$ is the LSI constant of $\tilde{\mu}_t$ given in Proposition 2.*

The proof can be found in Corollary 4 in the Appendix. This is an extension of the "entropy sandwich" argument in Nitanda et al. (2022); Chizat (2022), in which the right hand side of the bound is given by $\mathrm{KL}(\mu_0, \mu^*)$ instead of the Wasserstein distance. However, in our setting, $\mu_0 = \mu_0^N$ is a discrete distribution and thus the KL divergence from $\mu^*$ is not finite. To resolve this issue, our analysis shows an upper bound of the KL divergence at $t > \tau_0$ via the Wasserstein distance $W_2(\mu_0, \mu^*)$ (see Corollary 3). Then, we obtain the following theorem on the convergence of term (I).

**Theorem 1** (Geometric ergodicity). *Under Assumption 1, the term (I) converges as*

$$\mathbb{E}[\mathcal{U}(t, \mu_0^N)] - \Phi(\mu^*) \leq C\sqrt{2\alpha^{-1}\psi_2(\tau_0)}\exp(-\alpha\lambda(t - \tau_0))\mathbb{E}[W_2(\mu_0^N, \mu^*)],$$

*for any $t > \tau_0$ where $\tau_0 > 0$ is an arbitrary positive real number.*

*Proof.* By Otto-Villani's theorem (Otto and Villani, 2000), LSI implies Talagrand's inequality: $W_2(\mu, \mu^*) \leq \sqrt{\frac{2}{\alpha}\mathrm{KL}(\mu, \mu^*)}$. Also, the smoothness of $\Phi$ entails $\Phi(\mu) - \Phi(\mu^*) \leq CW_2(\mu, \mu^*)$ (see Lemma 10). The assertion is obtained by combining Lemma 1 and Talagrand's inequality. $\square$

## 5.2 BOUNDING THE PROPAGATION OF CHAOS TERM (II)

Bounding the second term (II) is much more involved. We utilize the following evaluation adapted from Delarue and Tse (2021) (see Arnaudon and Del Moral (2020) for similar calculation):

$$\mathbb{E}[\mathcal{U}(0, \mu_t^N) - \mathcal{U}(t, \mu_0^N)] = \frac{1}{N}\sum_{i=1}^{d}\int_0^t \mathbb{E}\left[\int\left(\partial_{(x_1)_i}\partial_{(x_2)_i}\frac{\delta^2\mathcal{U}}{\delta\mu^2}(t - s, \mu_s^N)(x, x)\right)\mu_s^N(\mathrm{d}x)\right]\mathrm{d}s.$$

Intuitively, the integrand on the right hand side approximately represents $\mathbb{E}[\mathcal{U}(0, \mu_{s+\epsilon}^N) - \mathcal{U}(t, \mu_s^N)]$ for small $\epsilon$, that is, how a small time difference between the finite particle and continuous limit propagates to the terminal time $t$. Here, for a linear operator $q$ acting on a function $f : \mathbb{R}^d \to \mathbb{R}$, we write $f(q) := q(f)$. Then from Delarue and Tse (2021) (see also Appendix C.2, C.3) it holds that

$$\partial_{(x_1)_i}\partial_{(x_2)_i}\frac{\delta^2\mathcal{U}}{\delta\mu^2}(t, \mu_0)(x_1, x_2)$$

$$= \frac{\delta^2\Phi}{\delta\mu^2}(\mu_t)(d_i^{(1)}(t; \mu_0, \xi, x_1), d_i^{(1)}(t; \mu_0, \xi, x_2)) + \frac{\delta\Phi}{\delta\mu}(\mu_t)(d_{i,i}^{(2)}(t; \mu_0, x_1, x_2)),$$

where $d_i^{(1)}$ and $d_{i,j}^{(2)}$ are linear operators defined by

$$d_i^{(1)}(t; \mu_0, \xi, x_1)(\phi) = \partial_{(x_1)_i}\frac{\delta}{\delta\mu}(m(t; \cdot)(\phi))|_{\mu_0}(x_1),$$

$$d_{i,j}^{(2)}(t; \mu_0, x_1, x_2)(\phi) = \partial_{(x_1)_i}\partial_{(x_2)_j}\frac{\delta^2}{\delta\mu^2}(m(t; \cdot)(\phi))|_{\mu_0}(x_1, x_2),$$

for a smooth test function $\phi : \mathbb{R}^d \to \mathbb{R}$, where $m(t, \mu)(\phi) = \int\phi(x)\mathrm{d}m(t, \mu)(x)$ (we will also use the same notation for a general measure $\mu$). The dynamics of these operators is characterized by Proposition 6 in Appendix C.1, which is adapted from Delarue and Tse (2021). To obtain a uniform-in-time evaluation of term (II), we aim to show a rapid decay of $d_i^{(1)}$ and $d_{i,j}^{(2)}$.

**Isoperimetry of $\mu_t$ via *super LSI*.** Our strategy is to establish the boundedness of the integral $\int_0^t \mathbb{E}\left[\int \partial_{(x_1)_i}\partial_{(x_2)_i}\frac{\delta^2\mathcal{U}}{\delta\mu^2}(t - s, \mu_s^N)(x, x)\mu_s^N(\mathrm{d}x)\right]\mathrm{d}s$ by proving the exponential convergence of $\partial_{(x_1)_i}\partial_{(x_2)_i}\frac{\delta^2\mathcal{U}}{\delta\mu^2}(t - s, \mu_s^N)(x, x)$. However, this requires a local evaluation around $m(t - s; \mu_s^N)$, for which we cannot exploit "global" properties such as the log-Sobolev condition of the optimal

solution $\mu^*$. Instead, our current analysis requires $m(t - s; \mu_s^N)$ to also satisfy the LSI condition, which is technically demanding to establish. To our knowledge, similar condition has only been recently verified in Guillin et al. (2021); Lacker and Flem (2022) for a limited class of interaction potentials which cannot cover the case of mean-field neural networks.

To overcome this difficulty, we impose a stronger-than-quadratic tail growth condition on the regularizer, i.e., $r(x) = \Omega(\|x\|^{2+\delta})$ (see third point in Assumption 1). Under this assumption, we can show that the proximal Gibbs measure $\tilde{\mu}_\tau$ associated with $\mu_\tau = m(\tau + s; \mu_s^N)$ satisfies the *super logarithmic Sobolev inequality* defined below.

**Definition 2** (super logarithmic Sobolev inequality (super LSI)). *We say that a probability measure $\mu$ satisfies super log-Sobolev inequality if there exists a monotonically non-increasing function $\beta : (0, \infty) \to \mathbb{R}$ such that for any $\phi$ satisfying $\mathbb{E}_\mu[\phi^2] = 1$ and $\mathbb{E}_\mu[\|\nabla \phi\|^2] < \infty$, it holds that*

$$\mu(\phi^2 \log \phi^2) \leq r \int \|\nabla \phi\|^2 \mathrm{d}\mu + \beta(r) \quad (\forall r > 0).$$

It is known that super LSI implies LSI if there exists $r > 0$ such that $\beta(r) = 0$.

**Lemma 2.** *For any $\mu \in \mathcal{P}$, the probability measure $\tilde{\mu}$ corresponding to the proximal Gibbs density $p_\mu$ satisfies the super log-Sobolev inequality with $\beta(r) = C' - \frac{4+2\delta}{\delta} \log(r/2)$ with a constant $C' > 0$. Furthermore, it satisfies the log-Sobolev inequality with LSI constant $\tilde{\alpha} = \exp(-\frac{\delta C'}{4+2\delta})$.*

The proof is given in Appendix B.3. An important consequence of this lemma is that $\mu_t$ and $\tilde{\mu}_t$ have a *bounded density ratio* when $t$ is sufficiently large. This implies that many properties of $\tilde{\mu}_t$ are also inherited by $\mu_t$. Crucially, the bound on density ratio is strong enough for the LSI condition to be transferred from $\tilde{\mu}_t$ to $\mu_t$, which allows us to establish the exponential convergence of $d_i^{(1)}$ and $d_{i,j}^{(2)}$.

**Corollary 1.** *Under Assumption 1, there exists some $T_0 > 0$ depending on $d, B, \delta, \alpha, \lambda, c_r, C_r$ and $\tilde{Q}_0$ depending on $W_2(\mu_0, \mu^*)$ such that for all $t \geq T_0 + \tilde{Q}_0$ we have*

$$\frac{1}{\sqrt{2}} \leq \left| \frac{\mathrm{d}\mu_t}{\mathrm{d}\tilde{\mu}_t}(x) \right| \leq \sqrt{2} \ (\forall x \in \mathbb{R}^d), \quad \left\| \frac{\mathrm{d}\mu_t}{\mathrm{d}\tilde{\mu}_t} - 1 \right\|_\infty \leq C' \exp(-\alpha\lambda(t - T_0)),$$

*where $C'$ is some positive constant. Moreover, for $t \geq T_0 + \tilde{Q}_0$, $\mu_t$ satisfies $(\alpha/2)$-LSI.*

**Uniform-in-time propagation of chaos.** Equipped with the LSI condition on $\mu_t$, we can now control term (II) in the error decomposition by proving exponential convergence of $d_i^{(1)}$ and $d_{i,j}^{(2)}$. In particular, LSI implies the Poincaré inequality which then roughly ensures the KL-divergence behaves like a strongly convex function around $\mu_t$. Therefore, a small perturbation from $\mu_t$ exponentially converges to 0 as $t$ grows, which entails the fast convergence of $d_i^{(1)}$ and $d_{i,j}^{(2)}$ because these quantities represent infinitesimal displacement of $\mu_t$.

**Theorem 2** (Uniform Propagation of Chaos). *Suppose that the support of $\mu_0$ is bounded. Then for any $0 < s < t$ and $1 \leq i \leq d$, it holds that*

$$\mathbb{E}\left[ \int \left( \partial_{(x_1)_i} \partial_{(x_2)_i} \frac{\delta^2 \mathcal{U}}{\delta \mu^2}(t - s, \mu_s^N)(x, x) \right) \mu_s^N(\mathrm{d}x) \right] = \mathcal{O}(\exp(-\lambda\alpha(t - s - T_0)/2)),$$

*with some constant $T_0 > 0$. This implies that*

$$\mathbb{E}[\mathcal{U}(0, \mu_t^N) - \mathcal{U}(t, \mu_0^N)] = \mathcal{O}(N^{-1}).$$

The proof of this theorem can be found in Appendix C.6. In addition, note that the informal theorem stated in Section 1 is a direct consequence of the above theorem (see Corollary 7 for details).

## 5.3 PUTTING THINGS TOGETHER

By combining the previous calculations, we arrive at the following characterization on the difference between the finite-width neural network and the optimal (infinite-width) solution $\mu^*$.

**Corollary 2.** *Under Assumption 1, if the initial distribution $\mu_0$ has bounded support, then we have*

$$\mathbb{E}[\mathcal{U}(0, \mu_t^N)] - \Phi(\mu^*) \leq C_1 N^{-1} + C_2 \sqrt{2\alpha^{-1}\psi_2(\tau_0)} \exp(-\alpha\lambda(t - \tau_0)),$$

*for some constants $C_1, C_2$ and any $\tau_0 > 0$.*

Noticeably, since both Theorem 1 and Theorem 2 do not require sufficiently large regularizations, our finite-particle guarantee holds for *any* choice of regularization strength $\lambda, \lambda_1$; in contrast, prior works on uniform propagation of chaos typically assume weak interaction or large noise, which limits the applicability to the optimization of neural networks in the mean-field regime.

As an important consequence of our general convergence guarantee, we know that the finite particle dynamics also exhibits a similar decay of the training and test losses compared to the continuous limit up to a $\mathcal{O}(1/N)$ discretization error; this can be straightforwardly checked by considering the special setting where $\Phi(\mu) = \mathbb{E}_{(Z,Y)\sim P}[\ell(f_\mu(Z), Y)]$ (see Example 1).

**Remark 1.** *Our current convergence result holds in expectation. To obtain a high probability statement, as discussed in Arnaudon and Del Moral (2020), one may apply a martingale concentration inequality (e.g., see Lemma 3.2 of Nishiyama (1997)) to obtain a guarantee in the form of $\mathcal{U}(0, \mu_t^N) - \mathcal{U}(t, \mu_0^N) - \mathbb{E}[\mathcal{U}(0, \mu_t^N) - \mathcal{U}(t, \mu_0^N)] = c\sqrt{\log(\epsilon^{-1})/N}$ with probability $1 - \epsilon$.*

## 6 NUMERICAL EXPERIMENTS

We provide empirical support for our propagation of chaos result in a synthetic student-teacher setting. We consider the empirical risk minimization problem, where the training labels are generated by a teacher model which is a Gaussian function defined as $f^*(z) = \exp\left(-\frac{\|z-a\|^2}{2d}\right)$. We set $n = 2000, d = 20$. The loss is chosen to be the squared error, and for the regularization term we set $r(x) = \|x\|^2$ or $r(x) = \|x\|^4$, and the regularization strength $\lambda_1 = \lambda = 10^{-2}$. The student model is a two-layer neural network with tanh activation, and the width $N$ is taken to be $\{16, 32, 64, 128, 256, 512, 1024, 2048\}$. We optimize the student model using NPGD with step size $\eta = 10^{-2}$. The global optimal solution $\mu^*$ is approximated via the particle dual averaging (PDA) algorithm (Nitanda et al., 2021): we

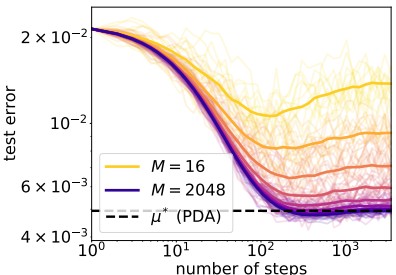

Figure 2: Test error of NNs optimized by NPGD ($r(x) = \|x\|^2$). Solid curve: mean over 100 runs. Translucent curve: individual runs.

set the model width $N = 2048$ and number of outer loop steps $T = 250$; we scale the number of inner loop steps $T_t$ with $t$, and the step size $\eta_t$ with $1/\sqrt{t}$, where $t$ is the outer loop iteration.

For the figures, we report the training or test error without the regularization terms. In Figure 1 in the Introduction, we plot the training error for $r(x) = \|x\|^4$; whereas in Figure 2, we plot the test error for $r(x) = \|x\|^2$. Observe that even though our current theoretical analysis does not cover the latter setting, the empirical trends are almost identical: as the width $N$ increases, the performance of the finite-width network improves and approaches that of the (approximate) optimal solution $\mu^*$.

## 7 CONCLUSION

In this paper, we established the first uniform-in-time propagation of chaos for the mean-field Langevin dynamics in the context of neural network optimization. In contrast to most existing works, our analysis gives a quantitative discretization error and does not blow up through time, and we do not impose the commonly-assumed weak interaction condition. This is achieved by utilizing a super logarithmic Sobolev inequality that is satisfied by a regularization term with super-quadratic tail. This condition then enables us to establish good isoperimetry of the intermediate solution $\mu_t$, which gives an exponential convergence of the error propagation.

**Limitations and future directions.** Our current analysis requires a super-quadratic tail of the regularization term, which does not cover the commonly-used $\ell_2$ regularization (weight decay). We note that after our initial submission, Chen et al. (2022a) developed a different proof technique based on the tensorization of LSI which handles the case of quadratic regularization. Another important future direction is to extend the analysis discrete-time dynamics. Finally, the mean-field Langevin dynamics has found applications beyond neural network optimization (Chizat et al., 2022); hence we are optimistic that our technique can provide finite-particle guarantee for interacting particle algorithms in other applications.

ACKNOWLEDGMENT

TS was partially supported by JSPS KAKENHI (20H00576) and JST CREST. AN was partially supported by JSPS Kakenhi (22H03650) and JST-PRESTO (JPMJPR1928). DW was partially supported by a Borealis AI Fellowship.

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

TABLE OF CONTENTS

# —————————— **Appendix** ——————————

## A ADDITIONAL RELATED WORKS

### A.1 OPTIMIZATION OF MEAN-FIELD NEURAL NETWORKS

Mean-field analysis of two-layer neural networks describes the optimization dynamics as a partial differential equation (PDE) of the parameter distribution, from which convergence to the global optimal solution can be shown (Nitanda and Suzuki, 2017; Chizat and Bach, 2018; Mei et al., 2018; Rotskoff and Vanden-Eijnden, 2018). However, quantitative convergence results usually requires additional assumptions on the learning problem (Chizat, 2019; Akiyama and Suzuki, 2021; Chen et al., 2022b), or modification of the dynamics (Rotskoff et al., 2019; Wei et al., 2019). For the entropy-regularized objective (5), efficient optimization algorithms have been proposed in Nitanda et al. (2021); Oko et al. (2022); Nishikawa et al.; noticeably, the quantitative convergence rate guarantees for these particle-based methods remain valid in both finite-width and discrete-time settings.

If we restrict ourselves the standard gradient descent-based methods, then fluctuation around the mean-field limit has been studied in Rotskoff and Vanden-Eijnden (2018); Sirignano and Spiliopoulos (2020); Pham and Nguyen (2021). Closely related to our work are the quantitative propagation of chaos results for two-layer neural network from De Bortoli et al. (2020); Chen et al. (2020c). In particular, De Bortoli et al. (2020) studied the impact of learning rate in the stochastic gradient descent update, but did not provide a convergence rate or uniform control of the discretization error over time (due to the lack of regularization). Chen et al. (2020c) showed that the long-time fluctuation induced by finite width can be controlled assuming that the mean-field dynamics converges at a specific rate, but such condition is very challenging to establish in their setting. In contrast, our result provides a uniform-in-time bound on the finite-width discretization error under conditions that can be verified for regularized risk minimization problems.

### A.2 INTERACTING PARTICLE SYSTEMS AND PROPAGATION OF CHAOS

Propagation of chaos has been analyzed mainly in the context of McKean-Vlasov equations whose drift term has the form of $b(x, \mu) = \nabla V(x) - \nabla \int W(\cdot, y) \mathrm{d}\mu(y)|_x$. Generally, neural network optimization is not included in this class, but techniques to analyze such equations can be applied to the neural network setting. Many existing works analyze the discretization error in a bounded time horizon (see Lacker (2021) and references therein). That is, for a fixed time horizon $T$, it has been shown that $\sup_{t \in [0,T]} |\Psi(\mu_t^N) - \Psi(\mu_t)| \leq C_T/N$. However, the constant $C_T$ depends on $T$ and the dependency is often exponential.

To obtain a uniform in time control, there are roughly two approaches: $(i)$ the uniform log-Sobolev (or Poincaré) inequality approach, and $(ii)$ the local Taylor expansion approach. The first approach $(i)$ directly derives the LSI constant of the $N$-particle dynamics $(X = (X_t^i)_{i=1}^N)$ and show that the constant can be bounded from below *uniformly* over all $N$. For example, Guillin et al. (2022) (see also references therein) established a uniform LSI constant under a weak interaction assumption. Ren and Wang (2021); Delgadino et al. (2021) showed geometric ergodicity based on a similar evaluation. Salem (2018) considered a uniform *WJ-inequality* instead of the log-Sobolev inequality also based on weak interaction conditions. The second approach $(ii)$ is what we employed; in particular, we follow the framework developed in Arnaudon and Del Moral (2020); Delarue and Tse (2021). In addition, Durmus et al. (2020) devised a different technique using a sophisticated coupling argument. We note that these prior results all assume weak interaction between particles to establish a uniform-in-time evaluation, and thus cannot be applied to the neural network setting.

## B BASIC PROPERTIES OF THE SOLUTION

### B.1 BOUNDEDNESS AND UNIQUENESS OF THE SOLUTION

**Proposition 3** (Theorem 2.1 and Corollary 4.3 of Wang (2018), adapted)**.** *Suppose that there exist $K_1, K_2, K_3 \in C([0, \infty); (0, \infty))$ such that*

- $-2\langle b(x,\mu)-b(y,\nu), x-y\rangle \leq K_1(t)\|x-y\|^2 + K_2(t)W_2(\mu,\nu)\|x-y\|$ for any $x, y \in \mathbb{R}^d$, $t \geq 0$ and $\mu, \nu \in \mathcal{P}_2$,

- $\|b(0,\mu)\|^2 \leq K_3(t)\{1 + \mu(\|\cdot\|^2)\}$ for any $\mu \in \mathcal{P}_2$ and $t \geq 0$.

*Then, the mean-filed Langevin dynamics* (3) *has a unique strong solution.*

*Moreover, for two different solutions $X_t^{(1)}$ and $X_t^{(2)}$ with different initial distributions $\mu_0^{(1)}, \mu_0^{(2)} \in \mathcal{P}_2$, the corresponding laws $\mu_t^{(k)} = \mathrm{Law}(X_t^{(k)})$ $(k = 1, 2)$ are equivalent for $t > 0$ and satisfy the following contraction property:*

$$W_2(\mu_t^{(1)}, \mu_t^{(2)})^2 \leq \psi_1(t)W_2(\mu_0^{(1)}, \mu_0^{(2)})^2, \quad \mathrm{KL}(\mu_t^{(1)}, \mu_t^{(2)}) \leq \psi_2(t)W_2(\mu_0^{(1)}, \mu_0^{(2)})^2, \quad (9)$$

*for $t > 0$, where $\psi_j : (0, \infty) \to [0, \infty)$ $(j = 1, 2)$ depends only on $K_1, K_2, K_3, \lambda$ and is an increasing function.*

Note that it is possible that $\lim_{t\to 0} \psi_2(t) = \infty$. Indeed, $\psi_2(t)$ is given as

$$\psi_2(t) = \frac{1}{2\lambda}\left(\frac{K_1(t)}{1 - e^{-K_1(t)t}} + \frac{tK_2(t)\exp(2t(K_1(t) + K_2(t))))}{2}\right).$$

As a consequence of this proposition, we obtain the following corollary.

**Corollary 3.** *Under Assumption 1, the mean-filed Langevin dynamics* (3) *has a unique strong solution. Moreover, the two distributions $\mu_t^{(1)}$ and $\mu_t^{(2)}$ corresponding to different initial distributions in $\mathcal{P}_2$ are equivalent and satisfy the contraction property* (9).

*In particular, $\mu_t$ is equivalent to $\mu^*$ and hence is equivalent to the Lebesgue measure. Therefore, $\mu_t$ has a density that is positive for all $x \in \mathbb{R}^d$, and if $\mu_0 \in \mathcal{P}_2$, then the density of $\mu_t$ satisfies $(t, x) \mapsto \frac{d\mu_t}{dx}(x) \in C^{1,\infty}((0, \infty) \times \mathbb{R}^d, \mathbb{R})$.*

*Proof.* Let $H(x, \mu) = \frac{1}{n}\sum_{j=1}^{n} \ell_j'(f_\mu)h_j(x)$. We just need to check the two conditions in Proposition 3. The first condition can be checked as follows: by noticing the convexity of $r$, we have

$$\begin{aligned}
&- 2\langle b(x,\mu) - b(y,\nu), x-y\rangle \\
&= -2\langle\nabla H(x,\mu) - \nabla H(y,\nu), x-y\rangle - 2\lambda_1\langle\nabla r(x) - \nabla r(y), x-y\rangle \\
&= -2\langle\nabla H(x,\mu) - \nabla H(y,\mu), x-y\rangle - 2\langle\nabla H(y,\mu) - \nabla H(y,\nu), x-y\rangle \\
&\quad - 2\lambda_1\langle\nabla r(x) - \nabla r(y), x-y\rangle \\
&\leq 2B^2\|x-y\|^2 + 2B^2\|f_\mu - f_\nu\|_\infty\|x-y\| \quad (\because \text{Assumption 1 and convexity of } r) \\
&\leq 2B^2\|x-y\|^2 + 2B^3 W_2(\mu,\nu)\|x-y\|,
\end{aligned}$$

which yields the first condition. Next, the second condition can be guaranteed as

$$\|b(0,\mu)\| \leq \|\nabla H(0,\mu)\| + \lambda_1\|\nabla r(0)\| \leq B^2 + \lambda C_r.$$

Therefore, applying Proposition 3, we obtain the first assertion.

As for the second assertion, we first note that $\mu^*$ is an invariant measure of the mean-field Langevin dynamics. Hence, if $\mu_0 = \mu^*$, then $\mu_t = \mu^*$ for any $t > 0$. Moreover, recall that $p_{\mu^*} = \mu^*$ by Proposition 1. Combining these relations, we have $\mu^* = p_{\mu^*} = \mu_t$ $(t > 0)$ when the initial distribution satisfies $\mu_0 = \mu^*$. Therefore, $\mu_t$ with a general initial distribution is equivalent to $\mu^*$ by Proposition 3. Finally, $(t, x) \mapsto \frac{d\mu_t}{dx}(x) \in C^{1,\infty}((0, \infty) \times \mathbb{R}^d, \mathbb{R})$ follows from Theorem 5.1 of Jordan et al. (1998). $\square$

In the subsequent analysis, it is important to ensure the boundedness of the moments of $X_t$. Indeed, we have the following estimate.

**Lemma 3.** *Under Assumption 1, for any $p \geq 2$, $\mathbb{E}[\|X_0\|^p] < \infty$ implies*

$$\mathbb{E}\left[\sup_{t\in[0,T]} \|X_t\|^p\right] < \infty$$

*for any $T > 0$.*

This is a direct consequence of Theorem 2.1 of Wang (2018). Therefore, we have that $\mu_t \in \mathcal{P}_p$ as long as $\mu_0 \in \mathcal{P}_p$. Indeed, we are interested in a situation where $\mu_0 = \mu_s^N = \frac{1}{N}\sum_{i=1}^N \hat{X}_s^i$, and hence $\mu_t \in \mathcal{P}_p$ for any $p \geq 2$ because a discrete measure has a finite moment for any $p \geq 2$.

If $p = 2$ or $p = 2 + \delta$, we have a sharper uniform bound as follows.

**Lemma 4.** *Under Assumption 1, we have the following uniform boundedness of the moments:*

$$\sup_{t>0} \mathbb{E}[\|X_t\|^2] \leq \max\left\{\frac{1}{\lambda_1 c(1+\delta/2)}\left[\frac{B^4}{\lambda_1 c(1+\delta/2)} + \lambda_1 c\delta + 2\lambda d\right], \mathbb{E}[\|X_0\|^2]\right\},$$

$$\sup_{t>0} \mathbb{E}[r(X_t)] \leq \max\left\{\mathbb{E}[r(X_0)],\right.$$

$$\left.\frac{C_r(2+\delta)}{\lambda_1 c_r^2(1+\delta)}\left[\frac{1}{2+\delta}\frac{(B^2 C_r)^{2+\delta}}{(\lambda_1 c_r/2)^{1+\delta}} + \frac{(1+2\delta)\lambda_1 c_r}{2+\delta} + \lambda C_r + \frac{2}{2+\delta}(\lambda C_r)^{(2+\delta)/2}\left(\frac{\delta}{\lambda c_r(1+\delta)}\right)^{\delta/2}\right]\right\}.$$

*The same bounds also hold with respect to $\mathbb{E}[\|\hat{X}_t^i\|^2]$ and $\mathbb{E}[r(\hat{X}_t^i)]$.*

*Proof.* Let $H_t(x) = \frac{1}{n}\sum_{j=1}^n \ell_j'(f_{\mu_t})h_j(x)$. By the formula of the infinitesimal generator, we have

$$\frac{\mathrm{d}}{\mathrm{d}t}\mathbb{E}[\|X_t\|^2] = \mathbb{E}[-2X_t^\top b(X_t, \mu_t)] + 2\lambda d.$$

By Young's inequality, the right hand side can be bounded as

$$-2\mathbb{E}\left[X_t^\top\left(\nabla H_t(X_t) + \lambda_1 \nabla r(X_t)\right)\right] + 2\lambda d$$

$$\leq 2B^2 \mathbb{E}[\|X_t\|] - 2\lambda_1 c\mathbb{E}[\|X_t\|^{2+\delta}] + 2\lambda d$$

$$\leq \frac{4B^4}{4\lambda_1 c(1+\delta/2)} + \frac{2\lambda_1 c(1+\delta/2)\mathbb{E}[\|X_t\|^2]}{2} + 2\lambda_1 c\left\{\frac{\delta}{2} - (1+\frac{\delta}{2})\mathbb{E}[\|X_t\|^2]\right\} + 2\lambda d$$

$$\leq \frac{B^4}{\lambda_1 c(1+\delta/2)} + \lambda_1 c\delta + 2\lambda d - \lambda_1 c(1+\frac{\delta}{2})\mathbb{E}[\|X_t\|^2].$$

Hence, we obtain that

$$\mathbb{E}[\|X_t\|^2] \leq \frac{1}{\lambda_1 c(1+\delta/2)}\left[\frac{B^4}{\lambda_1 c(1+\delta/2)} + \lambda_1 c\delta + 2\lambda d\right] \vee \mathbb{E}[\|X_0\|^2].$$

In the same vein, we can show the bound for $r(X_t)$ as follows. First note that

$$\frac{\mathrm{d}}{\mathrm{d}t}\mathbb{E}[r(X_t)] = \mathbb{E}[-\nabla^\top r(X_t)b(X_t, \mu_t)] + \lambda\mathbb{E}[\mathrm{Tr}[\nabla\nabla^\top r(X_t)]].$$

By Young's inequality, the right hand side can be bounded as

$$-\mathbb{E}\left[\nabla^\top r(X_t)\left(\nabla H_t(X_t) + \lambda_1 \nabla r(X_t)\right)\right] + \lambda C_r(1 + \mathbb{E}[\|X_t\|^\delta])$$

$$\leq B^2 \mathbb{E}[\|\nabla r(X_t)\|] - \lambda_1 \mathbb{E}[\|\nabla r(X_t)\|^2] + \lambda C_r(1 + \mathbb{E}[\|X_t\|^\delta])$$

$$\leq B^2 C_r \mathbb{E}[\|X_t\|^{1+\delta}] - \lambda_1 c_r \mathbb{E}[\|X_t\|^{2(1+\delta)}] + \lambda C_r(1 + \mathbb{E}[\|X_t\|^\delta])$$

$$\leq \frac{1}{2+\delta}\frac{(B^2 C_r)^{2+\delta}}{(\lambda_1 c_r/2)^{1+\delta}} + \frac{\lambda_1 c_r(1+\delta)}{2(2+\delta)}\mathbb{E}[\|X_t\|^{2+\delta}] - \lambda_1 c_r\left\{\frac{2(1+\delta)}{2+\delta}\mathbb{E}[\|X_t\|^{2+\delta}] - \frac{\delta}{2+\delta}\right\}$$

$$+ \lambda C_r + \frac{2}{2+\delta}(\lambda C_r)^{(2+\delta)/2}\left(\frac{\delta}{\lambda c_r(1+\delta)}\right)^{\delta/2} + \frac{\lambda_1 c_r(1+\delta)}{2(2+\delta)}\mathbb{E}[\|X_t\|^{2+\delta}]$$

$$\leq \frac{1}{2+\delta}\frac{(B^2 C_r)^{2+\delta}}{(\lambda_1 c_r/2)^{1+\delta}} + \frac{\delta\lambda_1 c_r}{2+\delta} + \lambda C_r + \frac{2}{2+\delta}(\lambda C_r)^{(2+\delta)/2}\left(\frac{\delta}{\lambda c_r(1+\delta)}\right)^{\delta/2}$$

$$- \frac{\lambda_1 c_r(1+\delta)}{2+\delta}\underbrace{\mathbb{E}[\|X_t\|^{2+\delta}]}_{\geq \mathbb{E}[r(X_t)]/C_r - 1}.$$

This gives the second bound.

The same argument can be applied to $\hat{X}_t^i$, which concludes the assertion. $\qquad\square$

**Lemma 5.** *Under Assumption 1, for $\mu_0 \in \mathcal{P}_p$ with $p \geq 2$, it holds that*

$$\nabla \log(\mu_t)(x) = -\frac{1}{t\sqrt{2\lambda}} \mathbb{E}\left[\int_0^t (I + s\nabla b(\cdot, \mu_s)|_{X_s}) \cdot \mathrm{d}W_s | X_t = x\right],$$

*and, if $p/\delta \geq 1$, it also holds that*

$$\mathbb{E}[\|\nabla \log(\mu_t)(X_t)\|^{p/\delta}] < \infty,$$

*for any $t > 0$.*

*Proof.* First, note that $\mu_0 \in \mathcal{P}_p$ ensures differentiability of $\mu_t$. The characterization of $\nabla \log(\mu_t)$ is given by the integration by parts formula investigated by Föllmer (1986) (see also Lemma 6.2 of Hu et al. (2019), Wang (2014) and Theorem 5.1 of Wang (2018)).

As for the moment bound, first we note that

$$\|\nabla b(\cdot, \mu)|_x\| = O(1 + \|x\|^\delta),$$

by the assumption. Then, by Jensen's inequality and the moment inequality of stochastic integral (Kim, 2013), we have that, for $q = p/\delta$,

$$\mathbb{E}[\|\nabla \log(\mu_t)\|^q] \lesssim \left(\frac{1}{t\sqrt{2\lambda}}\right)^q C_{t,q} \mathbb{E}\left[\int_0^t (1 + s\|\nabla b(\cdot, \mu_s)|_{X_s}\|)^q \mathrm{d}s\right]$$

$$\lesssim \left(\frac{1}{t\sqrt{2\lambda}}\right)^q C_{t,q} \mathbb{E}\left[\int_0^t (1 + s(1 + \|X_s\|^\delta))^q \mathrm{d}s\right] < \infty \quad (\because \text{Lemma 3}),$$

where $C_{t,q} = \left(\frac{q(q-1)}{2}\right)^{q/2} t^{\frac{q-2}{2}}$. $\qquad\qquad\qquad\qquad\qquad\qquad\qquad\qquad\qquad\qquad\qquad\square$

According to Lemma 3 and the remark following the lemma, we have $\mu_t \in \mathcal{P}_p$ for any $p \geq 2$ in our situation where $\mu_0$ is a discrete measure like $\mu_0 = \mu_s^N$. Hence, we may assume $\mathbb{E}[\|\nabla \log(\mu_t)(X_t)\|^p] < \infty$ for any $p \geq 2$. In particular, the Fisher divergence $I(\mu_t\|\mu^*)$ is well-defined for any $t > 0$ (but not defined for $t = 0$).

### B.2 GEOMETRIC ERGODICITY

For $\mu, \nu \in \mathcal{P}$ where $\nu$ is absolutely continuous with respect to $\mu$ and thus can be written as $\mathrm{d}\nu = f\mathrm{d}\mu$, the *Fisher* divergence of $\nu$ with respect to $\mu$ is defined as

$$I(\nu\|\mu) = 4\int \|\nabla \sqrt{f}\|^2 \mathrm{d}\mu = \int \|\nabla \log(f)\|^2 \mathrm{d}\nu.$$

**Proposition 4** (Geometric ergodicity of the mean-field Langevin dynamics (Nitanda et al., 2022; Chizat, 2022))**.**

$$\mathcal{L}(\mu_t) - \mathcal{L}(\mu^*) \leq \exp(-2\alpha\lambda t)(\mathcal{L}(\mu_0) - \mathcal{L}(\mu^*)).$$
$$\lambda \mathrm{KL}(\mu\|\mu^*) \leq \mathcal{L}(\mu) - \mathcal{L}(\mu^*) \leq \lambda \mathrm{KL}(\mu\|p_\mu).$$

Although Nitanda et al. (2022); Chizat (2022) assumed $r(x) = \Theta(\|x\|^2)$, we can adapt the same argument also to our situation. Indeed, the quadraticity of the regularization term is used to ensure the well-posedness of the solution, and in our setting this is ensured by Corollary 3, which yields the assertion of the proposition.

Combining Corollary 3 and Propositions 4, we obtain the following corollary.

**Corollary 4.** *Under Assumption 1, for any initial condition $\mu_0 \in \mathcal{P}$ with $W_2(\mu_0, \mu^*) < \infty$, it holds that $\mathcal{L}(\mu_t) < \infty$, and*

$$\mathcal{L}(\mu_t) - \mathcal{L}(\mu^*) \leq \exp(-2\alpha\lambda(t - \tau_0))\lambda\psi_2(\tau_0)W_2(\mu_0, \mu^*)^2,$$
$$\mathrm{KL}(\mu_t, \mu^*) \leq \exp(-2\alpha\lambda(t - \tau_0))\psi_2(\tau_0)W_2(\mu_0, \mu^*)^2,$$

*for any $t > \tau_0$ where $\tau_0 > 0$ can be arbitrary.*

*Proof.* We know that $\mu^* = p_{\mu^*} = \mu_t$ $(t > 0)$ when the initial distribution is $\mu_0 = \mu^*$ from the proof of Corollary 3. Plugging this relation into Corollary 3 and Propositions 4 gives the assertion. $\square$

As remarked in Section 5, in the convergence analysis we need to assume an LSI (or Poincaré inequality) on $\mu_t$ instead of $\tilde{\mu}_t$. This is not generally ensured. However, we can verify this condition if the semi-group satisfies the *super log-Sobolev inequality*. Indeed, the super log-Sobolev inequality entails the *ultra-contractivity* yielding an $L^\infty$-convergence of the density ratio between $\mu_t$ and $\tilde{\mu}_t$. This is remarkably useful to transfer the LSI property of $\tilde{\mu}_t$ to $\mu_t$.

### B.3 SUPER LOGARITHMIC SOBOLEV INEQUALITY

**Definition 3** (super log-Sobolev inequality (super LSI)). *We say that a probability measure $\mu$ satisfies super log-Sobolev inequality if there exits a monotonically non-increasing function $\beta : (0, \infty) \to \mathbb{R}$ such that*

$$\mu(f^2 \log f^2) \leq r \int \nabla f \cdot \nabla f \mathrm{d}\mu + \beta(r) \quad (\forall r > 0, \ \forall f \in D(\mathcal{E}), \ \mu(f^2) = 1).$$

**Proposition 5.** *If a probability measure $\mu$ is given by $\mu = \exp(-V)$ where $V(x) = \lambda_1 r(x) + H(x)$ with a convex function $r : \mathbb{R}^d \to \mathbb{R}$ satisfying $\lambda_1 \nabla r(x) \cdot x \geq c\|x\|^{2+\delta}$ for $\delta > 0$ and $h : \mathbb{R}^d \to \mathbb{R}$ satisfying $\|\nabla H\|_\infty \leq C < \infty$, then $\mu$ satisfies the super log-Sobolev inequality with $\beta(r) = C' - \frac{4+2\delta}{\delta} \log(r/2)$ where $C' > 0$ is a constant depending on $d, c, C, \delta$. In particular, it satisfies the log-Sobolev inequality with a constant $\alpha' > 0$ such that $\beta(2/\alpha') = 0$.*

*Proof.* This can be proven by adapting Corollary 5.7.5 of Wang (2005). Let $P_t$ be the semigroup that corresponds to the generator $L^* \phi = \Delta \phi - \nabla V \cdot \nabla \phi$. Then, we have that

$$\begin{aligned}
L^* \|x\|^2 &= d - (\lambda_1 \nabla r + \nabla H) \cdot (2x) \\
&\leq d - 2c\|x\|^{2+\delta} + 2\|\nabla H\|_\infty \|x\| \\
&\leq d - 2c\|x\|^{2+\delta} + c\|x\|^{2+\delta} + \frac{\|\nabla H\|_\infty^{\frac{2+\delta}{\delta}}}{c} \quad (\because \text{Young's inequality}) \\
&= d + \frac{C^{\frac{2+\delta}{\delta}}}{c} - c\|x\|^{2+\delta}.
\end{aligned}$$

Corollary 5.7.5 of Wang (2005) implies that

$$\|P_t\|_{L^2(\mu) \to L^\infty(\mu)} \leq \exp[c' t^{-(1+\delta/2)/(\delta/2)}] = \exp[c' t^{-(2+\delta)/\delta}],$$

for a constant $c' > 0$ depending on $d, c, C$. Indeed, (5.7.9) of Wang (2005) holds for $c \leftarrow d + \frac{C^{\frac{2+\delta}{\delta}}}{c}$ and $\gamma(r) \leftarrow c r^{1+\delta/2}$ in their notations, which yields the bound.

Then, Theorem 5.1.7 of Wang (2005) states that the super log-Sobolev inequality holds for $\beta(r) = 2 \log \|P_{r/2}\|_{L^2(\mu) \to L^\infty(\mu)} \leq 2 \log(c') - 2 \frac{2+\delta}{\delta} \log(r/2)$. By resetting $C' \leftarrow 2 \log(c')$, we obtain the assertion. $\square$

Due to our assumption on the regularization term in Assumption 1, we know that the proximal Gibbs measure satisfies the super LSI condition.

**Corollary 5.** *$\tilde{\mu}_t$ satisfies the super log-Sobolev inequality with $\beta(r) = C' - \frac{4+2\delta}{\delta} \log(r/2)$. In addition, it satisfies the log-Sobolev inequality with the LSI-constant $\tilde{\alpha} = \exp(-\frac{\delta C'}{4+2\delta})$.*

Let $P_{s,t}$ $(s < t)$ be the semigroup associated with $X_t$, i.e., $(P_{s,t} f)(x) = \mathbb{E}[f(X_t)|X_s = x]$ and $\mu_s P_{s,t} f = \mu_t f$.

**Theorem 3.** *There exists $t_0 \in (0, 1]$ such that*

$$\|P_{s,s+t_0}\|_{L^2(\tilde{\mu}_{s+t_0}) \to L^\infty(\tilde{\mu}_s)} \leq \exp\left[ \int_2^\infty \frac{\beta(1/p)}{p^2} \mathrm{d}p + C t_0 \right]$$

*for some constant $C > 0$. In particular, there exists $C_0 > 0$ such that $\|P_{s,s+t_0}\|_{L^2(\tilde{\mu}_{s+t_0}) \to L^\infty(\tilde{\mu}_s)} \leq C_0 < \infty$ uniformly over $s > 0$.*

*Proof.* We first note that

$$\tilde{\mu}_t(f^p \log(f)) \leq -r\tilde{\mu}_t(f^{p-1}L_{\mu_t}^* f) + \beta\left(\frac{4(p-1)r}{p}\right)p^{-1}\tilde{\mu}_t(f^p) + \tilde{\mu}_t(f^p)\log(\|f\|_{L^p(\tilde{\mu}_t)}). \quad (10)$$

for all $f \in \mathcal{D}(L_{\mu_t}^*)$ such that $f \geq 0$ (which we denote as $f \in \mathcal{D}_+$), $2 < p < \infty$, $r > 0$ (by definition the invariant measure corresponding to $L_{\mu_t}^*$ is $\tilde{\mu}_t$, i.e., $L_{\mu_t}\tilde{\mu}_t = 0$).

Let $t_0 := \int_2^\infty \frac{\gamma(p)}{p}\mathrm{d}p \leq 1$ for $\gamma(p) = \frac{1}{2p}$. We define $p(\tau)$ and $N(\tau)$ as functions on $[0, t)$ such that

$$p'(\tau) = \frac{p(\tau)}{\gamma \circ p(\tau)}, \quad p(0) = 2,$$

$$N'(\tau) = \frac{p'(\tau)\beta\left(\frac{4(p(\tau)-1)\gamma\circ p(\tau)}{p(\tau)}\right)}{p(\tau)^2}, \quad N(0) = 0.$$

Then, for $f \in \mathcal{D}_+$, if we rewrite $s \leftarrow s + t_0$, one has

$$\frac{\mathrm{d}}{\mathrm{d}\tau}\|P_{s-\tau,s}f\|_{L^{p(\tau)}(\tilde{\mu}_{s-\tau})}$$

$$=\frac{\tilde{\mu}_{s-\tau}(\frac{\mathrm{d}}{\mathrm{d}\tau}(P_{s-\tau,s}f)^{p(\tau)})}{p(\tau)\|P_{s-\tau,s}f\|_{p(\tau)}^{p(\tau)-1}} - \frac{p'(\tau)}{p(\tau)}\|P_{s-\tau,s}f\|_{L^{p(\tau)}(\tilde{\mu}_{s-\tau})}\log(\|P_{s-\tau,s}f\|_{L^{p(\tau)}(\tilde{\mu}_{s-\tau})})$$

$$+\frac{1}{p(\tau)}\|P_{s-\tau,s}f\|_{L^{p(\tau)}(\tilde{\mu}_{s-\tau})}^{1-p(\tau)}\int|P_{s-\tau,s}f|^{p(\tau)}\frac{\mathrm{d}}{\mathrm{d}\tau}\tilde{\mu}_{s-\tau}\mathrm{d}x. \quad (11)$$

The last term of the right hand side can be bounded as

$$\frac{1}{p(\tau)}\|P_{s-\tau,s}f\|_{L^{p(\tau)}(\tilde{\mu}_{s-\tau})}^{1-p(\tau)}\int|P_{s-\tau,s}f|^{p(\tau)}\frac{\mathrm{d}}{\mathrm{d}\tau}\tilde{\mu}_{s-\tau}\mathrm{d}x$$

$$\leq\frac{1}{p(\tau)}\|P_{s-\tau,s}f\|_{L^{p(\tau)}(\tilde{\mu}_{s-\tau})}^{1-p(\tau)}\int|P_{s-\tau,s}f|^{p(\tau)}\left|\frac{1}{n}\sum_{j=1}^n\ell_j''(f_{\mu_{s-\tau}})h_j(\cdot)\mu_{s-\tau}(L_{\mu_{s-\tau}}^*h_j)\right|\tilde{\mu}_{s-\tau}\mathrm{d}x$$

$$\leq C\frac{1}{p(\tau)}\|P_{s-\tau,s}f\|_{L^{p(\tau)}(\tilde{\mu}_{s-\tau})}^{1-p(\tau)}\int|P_{s-\tau,s}f|^{p(\tau)}\tilde{\mu}_{s-\tau}\mathrm{d}x = C\frac{\|P_{s-\tau,s}f\|_{L^{p(\tau)}(\tilde{\mu}_{s-\tau})}}{p(\tau)}. \quad (12)$$

By the backward Kolmogorov equation, and combining these inequalities (11) and (12) with the super log-Sobolev inequality (10), we have

$$\frac{\mathrm{d}}{\mathrm{d}\tau}(e^{-N(\tau)}\|P_{s-\tau,s}f\|_{L^{p(\tau)}(\tilde{\mu}_{s-\tau})})$$

$$=\frac{p'(\tau)e^{-N(s)}}{p(\tau)\|P_{s,s-\tau}f\|_{L^{p(\tau)}(\tilde{\mu}_{s-\tau})}^{p(\tau)-1}}\left\{\tilde{\mu}_{s-\tau}((P_{s-\tau,s}f)^{p(\tau)}\log P_{s-\tau,s}f)\right.$$

$$+\frac{p(\tau)}{p'(\tau)}\tilde{\mu}_{s-\tau}((P_{s-\tau,s}f)^{p(\tau)-1}L_{\mu_{s-\tau}}^*P_{s-\tau,s}f)$$

$$\left.-\frac{N'(\tau)p(\tau)}{p'(\tau)}\tilde{\mu}_{s-\tau}((P_{s-\tau,s}f)^{p(\tau)}) - \tilde{\mu}_{s-\tau}((P_{s-\tau,s}f)^{p(\tau)})\log\|P_{s-\tau,s}f\|_{L^{p(\tau)}(\tilde{\mu}_{s-\tau})}\right\}$$

$$+e^{-N(\tau)}\frac{1}{p(\tau)}\|P_{s-\tau,s}f\|_{L^{p(\tau)}(\tilde{\mu}_{s-\tau})}^{1-p(\tau)}\int|P_{s-\tau,s}f|^{p(\tau)}\frac{\mathrm{d}}{\mathrm{d}\tau}\tilde{\mu}_{s-\tau}\mathrm{d}x$$

$$\leq C\frac{1}{p(\tau)}e^{-N(\tau)}\|P_{s-\tau,s}f\|_{L^{p(\tau)}(\tilde{\mu}_{s-\tau})}.$$

Hence, we obtain that

$$\|P_{s-\tau,s}f\|_{L^{p(\tau)}(\tilde{\mu}_{s-\tau})} \leq e^{N(\tau)+C\int_0^\tau p(\tau')^{-1}\mathrm{d}\tau'}\|f\|_{L^2(\tilde{\mu}_s)} \leq e^{N(\tau)+C\tau}\|f\|_{L^2(\tilde{\mu}_s)}.$$

Here, notice that

$$\int_2^{p(s-t_0)}\frac{\gamma(p)}{p}\mathrm{d}p = \int_0^{t_0}\frac{p'(\tau)\gamma(p(\tau))}{p(\tau)}\mathrm{d}\tau = \int_0^{t_0}1\mathrm{d}\tau = t_0.$$

On the other hand, since we also have $t_0 = \int_2^\infty \frac{\gamma(p)}{p}\mathrm{d}p$ by its definition, it must be the case that $\lim_{\tau \nearrow t_0} p(\tau) = \infty$ and hence

$$\lim_{\tau \nearrow t_0} N(\tau) = \int_2^\infty \frac{\beta\left(\frac{4(p-1)\gamma(p)}{p}\right)}{p^2}\mathrm{d}p \leq \int_2^\infty \frac{\beta\left(2\gamma(p)\right)}{p^2}\mathrm{d}p.$$

This yields the first assertion. By Corollary 5 we can see that $\int_2^\infty \frac{\beta(2\gamma(p))}{p^2}\mathrm{d}p = \int_2^\infty \frac{\beta(1/p)}{p^2}\mathrm{d}p \lesssim \int_2^\infty \frac{1+\log(p)}{p^2}\mathrm{d}p =: C_0 < \infty$. This yields the second assertion. $\qquad\square$

Let $p(t, x, y)$ be the density function of the distribution $X_t$ conditioned by $X_0 = x \in \mathrm{supp}(\mu_0)$ with respect to $\tilde{\mu}_t$, i.e., $p(t, x, y) = \frac{\mathrm{d}P_{0,t}^*(\delta_x)}{\mathrm{d}\tilde{\mu}_t}(y)$. Then, we have that

$$\left| \int (p(t + t_0, x, y) - 1)f(y)\mathrm{d}\tilde{\mu}_{t+t_0}(\mathrm{d}y) \right| \leq |P_{0,t_0} \cdot P_{t_0,t+t_0}(f - \tilde{\mu}_{t+t_0}f)(x)|$$

$$\leq C_0 \|P_{t_0,t+t_0}f - \tilde{\mu}_{t+t_0}f\|_{L^2(\tilde{\mu}_{t_0})}, \tag{13}$$

where we used the bound $\|P_{0,t_0}\|_{L^2(\tilde{\mu}_{t_0}) \to L^\infty(\tilde{\mu}_0)} \leq C_0$ and the fact that the support of $\tilde{\mu}_0$ is the whole space $\mathbb{R}^d$. We will show that the right hand side converges to 0 in an exponential order by taking its differentiation with respect to $s$:

$$\partial_s \|P_{s,t}f - \tilde{\mu}_t f\|_{L^2(\tilde{\mu}_s)}^2 = 2\int \partial_s(P_{s,t}f - \tilde{\mu}_t f)(P_{s,t}f - \tilde{\mu}_t f)\mathrm{d}\tilde{\mu}_s + \int (P_{s,t}f - \tilde{\mu}_t f)^2 \partial_s \tilde{\mu}_s \mathrm{d}x.$$

We evaluate each term as follows.

(i) The second term in the right hand side can be evaluated as

$$\left| \int (P_{s,t}f - \tilde{\mu}_t f)^2 \partial_s \tilde{\mu}_s \mathrm{d}x \right|$$

$$\leq \left| \int (P_{s,t}f - \tilde{\mu}_t f)^2 \left( \frac{1}{n}\sum_{j=1}^n \ell_j''(f_{\mu_s})h_j(\cdot)\partial_s\mu_s(h_j) \right) \tilde{\mu}_s \mathrm{d}x \right|$$

$$\leq C\int (P_{s,t}f - \tilde{\mu}_t f)^2 \tilde{\mu}_s \mathrm{d}x \cdot \frac{1}{n}\sum_{j=1}^n |\partial_s\mu_s(h_j)|$$

$$\leq C\int (P_{s,t}f - \tilde{\mu}_t f)^2 \tilde{\mu}_s \mathrm{d}x \cdot \frac{1}{n}\sum_{j=1}^n \left| \int \nabla h_j(\nabla \log(\mu_s) - \nabla \log(\tilde{\mu}_s))\mathrm{d}\mu_s \right|$$

$$\leq C^2 \|P_{s,t}f - \tilde{\mu}_t f\|_{L^2(\tilde{\mu}_s)}^2 \sqrt{I(\mu_s \| \tilde{\mu}_s)}. \tag{14}$$

(ii) Next, we evaluate the first term. By the Poincaré inequality (PI), we have that

$$\int \partial_s(P_{s,t}f - \tilde{\mu}_t f)(P_{s,t}f - \tilde{\mu}_t f)\mathrm{d}\tilde{\mu}_s$$

$$= -\int L_{\mu_s}^*(P_{s,t}f)(P_{s,t}f - \tilde{\mu}_t f)\mathrm{d}\tilde{\mu}_s$$

$$= \int \lambda \|\nabla P_{s,t}f\|^2 \mathrm{d}\tilde{\mu}_s$$

$$\geq \lambda\alpha\|P_{s,t}f + \tilde{\mu}_s(P_{s,t}f)\|_{L^2(\tilde{\mu}_s)}^2 = \lambda\alpha\|P_{s,t}f - \tilde{\mu}_t f + \tilde{\mu}_t f - \tilde{\mu}_s(P_{s,t}f)\|_{L^2(\tilde{\mu}_s)}^2$$

$$= \lambda\alpha[\|P_{s,t}f - \tilde{\mu}_t f\|_{L^2(\tilde{\mu}_s)}^2 + 2\tilde{\mu}_s(P_{s,t}f - \tilde{\mu}_t f)(\tilde{\mu}_t f - \tilde{\mu}_s(P_{s,t}f)) + (\tilde{\mu}_t f - \tilde{\mu}_s(P_{s,t}f))^2]$$

$$= \lambda\alpha[\|P_{s,t}f - \tilde{\mu}_t f\|_{L^2(\tilde{\mu}_s)}^2 - 2(\tilde{\mu}_s(P_{s,t}f) - \tilde{\mu}_t f)^2 + (\tilde{\mu}_t f - \tilde{\mu}_t(P_{s,t}f))^2]$$

$$= \lambda\alpha[\|P_{s,t}f - \tilde{\mu}_t f\|_{L^2(\tilde{\mu}_s)}^2 - (\tilde{\mu}_s(P_{s,t}f) - \tilde{\mu}_t f)^2]$$

$$= \lambda\alpha\|P_{s,t}f - \tilde{\mu}_t f\|_{L^2(\tilde{\mu}_s)}^2 - \lambda\alpha(\tilde{\mu}_s(P_{s,t}f) - \tilde{\mu}_t f)^2.$$

Denote the total variation norm of two probability measures $\mu$ and $\nu$ by $\|\mu - \nu\|_{\mathrm{TV}}$, we notice that, if $t - s \geq t_0$, it holds that

$$
\begin{aligned}
&|\tilde{\mu}_s(P_{s,t}f) - \tilde{\mu}_t f| \\
&\leq |(P^*_{s,t-t_0}\tilde{\mu}_s - \tilde{\mu}_{t-t_0})(P_{t-t_0,t}f) - (\tilde{\mu}_t - P^*_{t-t_0,t}\tilde{\mu}_{t-t_0})f| \\
&\leq \|P^*_{s,t-t_0}\tilde{\mu}_s - \tilde{\mu}_{t-t_0}\|_{\mathrm{TV}}\|P_{t-t_0,t}f\|_{L^\infty} + |(\tilde{\mu}_t - P^*_{t-t_0,t}\tilde{\mu}_{t-t_0})f| \\
&\leq \|P^*_{s,t-t_0}\tilde{\mu}_s - \tilde{\mu}_{t-t_0}\|_{\mathrm{TV}}\|P_{t-t_0,t}\|_{L^2(\tilde{\mu}_t)\to L^\infty(\tilde{\mu}_{t-t_0})}\|f\|_{L^2(\tilde{\mu}_t)} + |(\tilde{\mu}_t - P^*_{t-t_0,t}\tilde{\mu}_{t-t_0})f| \\
&\leq C_0\|P^*_{s,t-t_0}\tilde{\mu}_s - \tilde{\mu}_{t-t_0}\|_{\mathrm{TV}}\|f\|_{L^2(\tilde{\mu}_t)} + |\tilde{\mu}_t f - \tilde{\mu}_{t-t_0}(P_{t-t_0,t}f)|, \quad (15)
\end{aligned}
$$

and, if $t - s < t_0$, the same bound without the first term in the right hand side holds. By Pinsker's inequality $\|\mu - \nu\|_{\mathrm{TV}} \leq \sqrt{\mathrm{KL}(\mu||\nu)/2}$, we have that

$$
\begin{aligned}
\|\tilde{\mu}_s - \mu^*\|_{\mathrm{TV}} &\leq \sqrt{\mathrm{KL}(\tilde{\mu}_s||\mu^*)/2} \\
&= \frac{1}{\sqrt{2}}\left[\int \lambda^{-1}\left(\frac{\delta F(\mu^*)}{\delta\mu}(\cdot) - \frac{\delta F(\mu_s)}{\delta\mu}(\cdot)\right)\mathrm{d}\tilde{\mu}_s + \log(Z(\mu^*)/Z(\tilde{\mu}_s))\right]^{1/2} \\
&\leq \frac{1}{\sqrt{\lambda}}\sqrt{\sup_x |\frac{\delta F(\mu^*)}{\delta\mu}(x) - \frac{\delta F(\mu_s)}{\delta\mu}(x)|} \\
&\leq \frac{C}{\sqrt{\lambda}}\sqrt{\|f_{\mu_s} - f_{\mu^*}\|_\infty} \leq C'\exp(-\alpha\lambda s)W_2(\mu_0, \mu^*), \quad (16)
\end{aligned}
$$

where $C' > 0$ is a constant depending on $\tau_0, \alpha, \lambda$. Moreover, if we let $\hat{\mu}_\tau = P^*_{s,\tau}\tilde{\mu}_s$, then

$$
\begin{aligned}
\partial_\tau \mathrm{KL}(\hat{\mu}_\tau||\mu^*) &= \int \log(\hat{\mu}_\tau/\mu^*)\partial_\tau\hat{\mu}_\tau\mathrm{d}x \\
&= \int (\lambda\Delta - b_\tau^\top\nabla)\log(\hat{\mu}_\tau/\mu^*)\hat{\mu}_\tau\mathrm{d}x \\
&= \int (-\lambda\nabla\log(\hat{\mu}_\tau) - b_\tau)\nabla\log(\hat{\mu}_\tau/\mu^*)\hat{\mu}_\tau\mathrm{d}x \\
&= -\lambda\int \|\nabla\log(\hat{\mu}_\tau) - \nabla\log(\mu^*)\|^2\mathrm{d}\hat{\mu}_\tau \\
&\quad - \lambda\int (\nabla\log(\mu^*) + \lambda^{-1}b_\tau)^\top(\nabla\log(\hat{\mu}_\tau) - \nabla\log(\mu^*))\mathrm{d}\hat{\mu}_\tau \\
&= -\lambda\int \|\nabla\log(\hat{\mu}_\tau) - \nabla\log(\mu^*)\|^2\mathrm{d}\hat{\mu}_\tau \\
&\quad - \lambda\int (\nabla\log(\mu^*) - \nabla\log(\tilde{\mu}_\tau))^\top(\nabla\log(\hat{\mu}_\tau) - \nabla\log(\mu^*))\mathrm{d}\hat{\mu}_\tau \\
&= -\frac{\lambda}{2}\int \|\nabla\log(\hat{\mu}_\tau) - \nabla\log(\mu^*)\|^2\mathrm{d}\hat{\mu}_\tau + \frac{\lambda}{2}\int \|\nabla\log(\tilde{\mu}_\tau) - \nabla\log(\mu^*)\|^2\mathrm{d}\hat{\mu}_\tau \\
&\leq -\lambda\alpha\mathrm{KL}(\hat{\mu}_\tau||\mu^*) + \sup_x \|\nabla\log(\tilde{\mu}_\tau)(x) - \nabla\log(\mu^*)(x)\|^2 \\
&\leq -\lambda\alpha\mathrm{KL}(\hat{\mu}_\tau||\mu^*) + C'\exp(-2\alpha\lambda\tau)W_2(\mu_0, \mu^*)^2,
\end{aligned}
$$

where we used the same argument as Eq. (16) in the first inequality. This evaluation, together with the bound of $\mathrm{KL}(\tilde{\mu}_s||\mu^*)$ in Eq. (16), implies that

$$
\begin{aligned}
\mathrm{KL}(\hat{\mu}_\tau||\mu^*) &\leq C'\max\{\exp(-\lambda\alpha(\tau - s))\mathrm{KL}(\tilde{\mu}_s||\mu^*), \exp(-2\alpha\lambda\tau)W_2(\mu_0, \mu^*)^2\} \\
&\leq C'\exp(-\alpha\lambda[(\tau - s) + 2s])W_2(\mu_0, \mu^*)^2.
\end{aligned}
$$

Therefore,

$$
\begin{aligned}
\|P^*_{s,t-t_0}\tilde{\mu}_s - \tilde{\mu}_{t-t_0}\|_{\mathrm{TV}} &\leq \|P^*_{s,t-t_0}\tilde{\mu}_s - \mu^*\|_{\mathrm{TV}} + \|\mu^* - \tilde{\mu}_{t-t_0}\|_{\mathrm{TV}} \\
&\lesssim \exp(-\alpha\lambda[(t - t_0 - s)/2 + s])W_2(\mu_0, \mu^*).
\end{aligned}
$$

Next, we evaluate the second term of the right hand side of Eq. (15), $|\tilde{\mu}_t f - \tilde{\mu}_{t-t_0}(P_{t-t_0,t}f)|$. We notice that

$$
\left|\partial_s\int (P_{s,t}f)\tilde{\mu}_s\mathrm{d}x\right| = \left|\int [(\lambda\Delta - b_s^\top\nabla)(P_{s,t}f)]\tilde{\mu}_s\mathrm{d}x + \int (P_{s,t}f)\partial_s\tilde{\mu}_s\mathrm{d}x\right|
$$

$$
\begin{aligned}
&= \left| \int (-\lambda \nabla \log(\tilde{\mu}_s) - b_s)^\top \nabla (P_{s,t} f) \tilde{\mu}_s \mathrm{d}x + \int (P_{s,t} f) \partial_s \tilde{\mu}_s \mathrm{d}x \right| \\
&= \left| \int (P_{s,t} f) \partial_s \tilde{\mu}_s \mathrm{d}x \right| \\
&\leq C \int |P_{s,t} f| \tilde{\mu}_s \mathrm{d}x \sqrt{I(\mu_s || \tilde{\mu}_s)} \lesssim \|f\|_{L^2(\tilde{\mu}_s)} \sqrt{I(\mu_s || \tilde{\mu}_s)},
\end{aligned}
$$

where the first inequality is obtained by the same argument as (14) and the last inequality is by Theorem 3 and the fact that the density ratio between $\tilde{\mu}_\tau$ and $\tilde{\mu}_{\tau'}$ are bounded from above and below for any $\tau, \tau'$ because of the boundedness of $\frac{\delta F}{\delta \mu}$. Hence, as in Eq. (22) and Eq. (23) below, we have

$$
\begin{aligned}
&|\tilde{\mu}_{t-t_0}(P_{t-t_0,t} f) - \tilde{\mu}_t f| \\
&\leq C \int_{t-t_0}^t \sqrt{I(\mu_s || \tilde{\mu}_s)} \mathrm{d}s \|f\|_{L^2(\tilde{\mu}_{t-t_0})} \\
&\leq C \frac{1}{\lambda} \sqrt{t_0} \exp(-\alpha \lambda(t - t_0 - \tau_0))(1 + \sqrt{\psi_2(\tau_0)} W_2(\mu_0, \mu^*)) \|f\|_{L^2(\tilde{\mu}_{t-t_0})}.
\end{aligned}
$$

Therefore, by applying these bounds to the right hand side of Eq. (15), we arrive at

$$
(\tilde{\mu}_s(P_{s,t} f) - \tilde{\mu}_t f)^2 \leq C \exp(-\lambda \alpha(t - s + 2s))(1 + W_2(\mu_0, \mu^*))^2 \|f\|_{L^2(\tilde{\mu}_t)}^2.
$$

(iii) Finally, by combining the bounds of (i) and (ii), we obtain that

$$
\begin{aligned}
&\partial_s \|P_{s,t} f - \tilde{\mu}_t f\|_{L^2(\tilde{\mu}_s)}^2 \\
&\geq 2\lambda \alpha \|P_{s,t} f - \tilde{\mu}_t f\|_{L^2(\tilde{\mu}_s)}^2 - C\lambda \alpha \exp(-\lambda \alpha(t + s))(1 + W_2(\mu_0, \mu^*))^2 \|f\|_{L^2(\tilde{\mu}_t)}^2 \\
&\quad - C \|P_{s,t} f - \tilde{\mu}_t f\|_{L^2(\tilde{\mu}_s)}^2 \sqrt{I(\mu_s || \tilde{\mu}_s)},
\end{aligned}
$$

which yields that, by taking the differentiation with respect to $s$ in the reverse direction,

$$
\begin{aligned}
&\partial_s \|P_{t-s,t} f - \tilde{\mu}_t f\|_{L^2(\tilde{\mu}_{t-s})}^2 \\
&\leq -(2\lambda \alpha - C\sqrt{I(\mu_{t-s} || \tilde{\mu}_{t-s})}) \|P_{t-s,t} f - \tilde{\mu}_t f\|_{L^2(\tilde{\mu}_{t-s})}^2 \\
&\quad + C\lambda \alpha \exp(-\lambda \alpha(t + s))(1 + W_2(\mu_0, \mu^*))^2 \|f\|_{L^2(\tilde{\mu}_t)}^2,
\end{aligned}
$$

and thus, if we write $C_t = C \int_{t_0}^t \sqrt{I(\mu_s || \tilde{\mu}_s)} \mathrm{d}s$,

$$
\begin{aligned}
&\|P_{t_0, t+t_0} f - \tilde{\mu}_{t+2t_0} f\|_{L^2(\tilde{\mu}_{t_0})}^2 \\
&\leq \exp(-2\lambda \alpha t + C_{t+t_0}) \|P_{t+t_0, t+t_0} f - \tilde{\mu}_{t+t_0} f\|_{L^2(\tilde{\mu}_{t+t_0})}^2 \\
&\quad + \lambda \alpha C \int_{t_0}^{t+t_0} \exp(-\lambda \alpha(t + s))(1 + W_2(\mu_0, \mu^*))^2 \|f\|_{L^2(\tilde{\mu}_{t+t_0})}^2 e^{-2\lambda \alpha(t-s) + C_{t+t_0} - C_s} \mathrm{d}s \\
&\leq \exp(-2\lambda \alpha t) \|f\|_{L^2(\tilde{\mu}_{t+t_0})}^2 \exp(C_{t+t_0})[1 + C(1 + W_2(\mu_0, \mu^*))^2].
\end{aligned}
$$

As in Eq. (23) below, the right hand side can be bounded by

$$
C_1 \exp(-2\lambda \alpha t) \|f\|_{L^2(\tilde{\mu}_{t+t_0})}^2 (1 + W_2(\mu_0, \mu^*)^2) \exp(C_2(1 + W_2(\mu_0, \mu^*)))
$$

for constants $C_1$ and $C_2$. This is further bounded

$$
C_3 \exp(-2\lambda \alpha t) \exp(C_4(1 + W_2(\mu_0, \mu^*))) \|f\|_{L^2(\tilde{\mu}_{t+t_0})}^2
$$

with constants $C_3$ and $C_4$. Therefore, we arrive at

$$
\|P_{t_0, t+t_0} f - \tilde{\mu}_{t+t_0} f\|_{L^2(\tilde{\mu}_{t_0})} \lesssim \exp(-\alpha \lambda t) \exp(C_4'(1 + W_2(\mu_0, \mu^*))) \|f\|_{L^2(\tilde{\mu}_{t+t_0})},
$$

where we used that the density ratio between $\tilde{\mu}_\tau$ and $\tilde{\mu}_{\tau'}$ are bounded from above and below for any $\tau, \tau'$ because of the boundedness of $\frac{\delta F}{\delta \mu}$ and $C_4' = C_4/2$.

Therefore, by applying this bound to the right hand side of (13), we have that

$$
\sup_{x,y} |p(t + t_0, x, y) - 1| \leq C' \exp(-\alpha \lambda t) \hat{Q}_0, \tag{17}
$$

for a constant $C'$ and $\hat{Q}_0 = \exp(C_4'(1 + W_2(\mu_0, \mu^*)))$, which can be checked by noticing the density of $\mu_t$ is smooth and taking $f(x) = \mathbf{1}\{\|x - x'\| \leq \epsilon\}$ for arbitrary $x' \in \mathbb{R}^d$ and letting $\epsilon \to 0$. This indicates that the density function of $\mu_t$ with respect to $\tilde{\mu}_t$ satisfies

$$\left\| \frac{\mathrm{d}\mu_t}{\mathrm{d}\tilde{\mu}_t} - 1 \right\|_\infty \leq C' \exp(-\alpha\lambda(t - t_0))\hat{Q}_0.$$

Importantly, this observation indicates that $\mu_t$ satisfies the $(\alpha/2)$-LSI for sufficiently large $t$ such as $C' \exp(-\alpha\lambda(t - t_0))\hat{Q}_0 \leq \min\{\sqrt{2} - 1, 1 - 1/\sqrt{2}\}$ via the Holley-Stroock bounded perturbation principle (e.g., Proposition 5.1.6 of Bakry et al. (2014)).

**Corollary 6.** *Under Assumption 1, there exits $T_0$ depending on $d, B, \delta, \alpha, \lambda, c_r, C_r$ such that $\mu_t$ satisfies the $(\alpha/2)$-LSI condition for $t \geq T_0 + \log(\hat{Q}_0)/\alpha\lambda$. Moreover,*

$$\frac{1}{\sqrt{2}} \leq \left| \frac{\mathrm{d}\mu_t}{\mathrm{d}\tilde{\mu}_t}(x) \right| \leq \sqrt{2}, \quad \left\| \frac{\mathrm{d}\mu_t}{\mathrm{d}\tilde{\mu}_t} - 1 \right\|_\infty \leq C' \exp(-\alpha\lambda(t - T_0))\hat{Q}_0,$$

*for all $t \geq T_0 + \log(\hat{Q}_0)/\alpha\lambda$, where $C'$ is a constant.*

By setting $\tilde{Q}_0 = \log(\hat{Q}_0)/\alpha\lambda$, we obtain Corollary 1 in the main text.

## C COMPUTATION OF ERROR TERMS

### C.1 DYNAMICS OF THE DERIVATIVE TERMS

Recall that $d_i^{(1)}$ and $d_{i,j}^{(2)}$ are linear operators defined by

$$d_i^{(1)}(t; \mu_0, \xi, x_1)(\phi) = \partial_{(x_1)_i} \frac{\delta}{\delta\mu}(m(t; \cdot)(\phi))|_{\mu_0}(x_1),$$

$$d_{i,j}^{(2)}(t; \mu_0, x_1, x_2)(\phi) = \partial_{(x_1)_i}\partial_{(x_2)_j} \frac{\delta^2}{\delta\mu^2}(m(t; \cdot)(\phi))|_{\mu_0}(x_1, x_2),$$

for a smooth test function $\phi : \mathbb{R}^d \to \mathbb{R}$, where $m(t, \mu)(\phi) = \int \phi(x)\mathrm{d}m(t, \mu)(x)$.

Given a linear operator $q$, we introduce a differential operator $L_\mu$ as follows,

$$L_\mu q = \lambda\Delta q + \nabla \cdot (b(\cdot, \mu)q) + \nabla \cdot \left( \mu \frac{\delta b}{\delta\mu}(\cdot, \mu)(q) \right),$$

which is defined in a weak sense, i.e., $(L_\mu q)(\phi) = q(L_\mu^* \phi) = q(\lambda\Delta\phi - b(\cdot, \mu) \cdot \nabla\phi - \int \frac{\delta b}{\delta\mu}(y, \mu)(\cdot) \cdot \nabla\phi(y)\mu(\mathrm{d}y))$ for a test function $\phi : \mathbb{R}^d \to \mathbb{R}$ with appropriate regularity condition. Following Delarue and Tse (2021), we know that these operators obey the following dynamics.

**Proposition 6.** $d_i^{(1)}$ and $d_{i,j}^{(2)}$ follows the following differential equation: for $t > 0$,

$$\begin{cases} \partial_t d_i^{(1)}(t; \mu, x) = L_{m(t,\mu)} d_i^{(1)}(t; \mu, x), \\ d_i^{(1)}(0; \mu, x) = (D_x')_i, \end{cases}$$

where $(D_x')_i$ is defined by $(D_x')_i(\phi) = \partial_{x_i}\phi(x)$, and

$$\begin{cases} \partial_t d_{i,j}^{(2)}(t; \mu, x_1, x_2) = L_{m(t,\mu)} d_{i,j}^{(2)}(t; \mu, x_1, x_2) \\ \quad + \nabla \cdot \left( d_j^{(1)}(t; \mu, x_2) \frac{\delta b}{\delta\mu}(\cdot, m(t, \mu))(d_i^{(1)}(t; \mu, x_1)) \right) \\ \quad + \nabla \cdot \left( d_i^{(1)}(t; \mu, x_1) \frac{\delta b}{\delta\mu}(\cdot, m(t, \mu))(d_j^{(1)}(t; \mu, x_2)) \right) \\ \quad + \nabla \cdot \left( m(t, \mu), \frac{\delta^2 b}{\delta\mu^2}(\cdot, m(t, \mu))(d_i^{(1)}(t; \mu, x_1), d_j^{(1)}(t; \mu, x_2)) \right), \\ d_{i,j}^{(2)}(0; \mu, x_1, x_2) = 0. \end{cases}$$

*The above equations should be interpreted in a weak sense, i.e., when $\partial_t q_t - L_{m(t;\mu)} - r_t = 0$ means that $q_t(\phi(t, \cdot)) - q_s(\phi(s, \cdot)) = \int_s^t q_\tau(\partial_\tau\phi(\tau, \cdot))\mathrm{d}\tau + \int_s^t q_\tau(L_{m(\tau;\mu)}^* \phi(\tau, \cdot))\mathrm{d}\tau + \int_s^t r_\tau(\phi(\tau, \cdot))\mathrm{d}\tau$ for a smooth test function $\phi : [0, \infty) \times \mathbb{R}^d \to \mathbb{R}$.*

The well-posedness of this differential equation is justified in Delarue and Tse (2021); Tse (2021) for mean-field dynamics on the $d$-dimensional torus. Although our dynamics is defined on $\mathbb{R}^d$ and the regularization term $r$ has unbounded gradient, the arguments there can be applied because $r$ is convex and does not depend on the distribution. Here we omit the technical details.

## C.2 FIRST ORDER DIFFERENTIATION

Let $H_t(x) = \frac{1}{n} \sum_{j=1}^n \ell_j'(f_{\mu_t}) h_j(x)$. Here, we evaluate the first order derivative $d_i^{(1)}(t; \mu, x)$. For that purpose, we define an operator $d^{(1)}(t; \mu, \xi, x)$ where $\xi \in \mathbb{R}^d, x \in \mathbb{R}^d, \mu \in \mathcal{P}$ as (we omit the argument $x$ if there is no confusion)

$$d^{(1)}(t; \mu, \xi, x)(\phi) = \xi^\top \nabla \frac{\partial}{\partial \epsilon} m(t, \epsilon \delta_x + (1 - \epsilon)\mu)(\phi)|_{\epsilon=0}$$

$$= \xi^\top \nabla \frac{\delta}{\delta \mu} m(t, \mu)(\phi)(x).$$

One may check that $d_i^{(1)}(t; \mu, x) = d^{(1)}(t; \mu, e_i, x)$ where $e_i$ is the indicator vector that has 1 in its $i$-th coordinate and 0 in other coordinates. In our case, we are interested in a setting

$$\mu_0 = \mu_s^N = \frac{1}{N} \sum_{i=1}^N \hat{X}_s^i,$$

which is a discrete measure with support consisting of $N$ points. In that case, $d^{(1)}(t; \mu_0, \xi, x)$ can be reformulated as the following gradient flow

$$\mathbb{E}_{\mu_0} \left[ \tilde{\xi}(X_0)^\top \nabla \frac{\delta}{\delta \mu} m(t, \mu_0)(\phi) \right] = \frac{1}{N} \sum_{i=1}^N \tilde{\xi}(\hat{X}_0^i)^\top \nabla \frac{\delta}{\delta \mu} m(t, \mu_0)(\phi).$$

where $\tilde{\xi}(X_0) = N\xi$ when $X_0 = x$ and $\tilde{\xi}(X_0) = 0$ otherwise. Clearly, $\tilde{\xi} \in L^2(\mu_0)$. Accordingly, we define the following SDE:

$$dX_t^\epsilon = -b(X_t^\epsilon, \mu_t^\epsilon)dt + \sqrt{2\lambda}dW_t,$$

$$\mu_t^\epsilon = \text{Law}(X_t^\epsilon), \quad X_0^\epsilon = X_0 + \epsilon\tilde{\xi}(X_0).$$

Then, if we define $\tilde{v}_t^\xi := \lim_{\epsilon \to 0} \frac{X_t^\epsilon - X_t}{\epsilon}$, it holds that

$$d^{(1)}(t; \mu_0, \xi, x)(\phi) = \mathbb{E}[\tilde{v}_t^\xi \cdot \nabla \phi(X_t)].$$

We can see that the infinitesimal displacement $\tilde{v}_t^\xi$ follows the following differential equation:

$$d\tilde{v}_t^\xi = - \left[ \tilde{v}_t^\xi \cdot \nabla b(x, \mu_t)|_{x=X_t} + \frac{\delta}{\delta \mu} b(X_t, \mu_t)(\tilde{q}_t) \right] dt, \tag{18}$$

from which we can derive moment bounds for $\tilde{v}_t^\epsilon$. In particular, for $p \geq 1$,

$$\frac{1}{p} \frac{d\|\tilde{v}_t^\xi\|^p}{dt} = -\|\tilde{v}_t^\xi\|^{p-2} \left[ \tilde{v}_t^\xi \cdot \nabla b(x, \mu_t)|_{x=X_t} \cdot \tilde{v}_t^\xi + \frac{\delta}{\delta \mu} b(X_t, \mu_t)(\tilde{q}_t) \cdot \tilde{v}_t^\xi \right]$$

$$= -\|\tilde{v}_t^\xi\|^{p-2} \left[ (\tilde{v}_t^\xi)^\top \nabla \nabla^\top H_t(X_t)\tilde{v}_t^\xi + \lambda(\tilde{v}_t^\xi)^\top \nabla \nabla^\top r(X_t)\tilde{v}_t^\xi + \frac{\delta}{\delta \mu} b(X_t, \mu_t)(\tilde{q}_t) \cdot \tilde{v}_t^\xi \right]$$

$$\leq B^2 \|\tilde{v}_t^\xi\|^p + B^2 \|\tilde{v}_t^\xi\|^{p-1} \mathbb{E}[\|\tilde{v}_t^\xi\|],$$

where we used Assumption 1 and convexity of $r$ in the last inequality.

First, by taking expectation of both sides for $p = 1$, we know that $\mathbb{E}[\|\tilde{v}_0^\xi\|] = \|\xi\|$ yields that $\mathbb{E}[\|\tilde{v}_t^\xi\|] \leq \exp(2B^2 t)\mathbb{E}[\|\tilde{v}_0^\xi\|] = \exp(2B^2 t)\|\xi\|$. Then, for $p > 1$, noticing that $\tilde{v}_0^\xi = 0$ for $X_0 \neq x$ and $\|\tilde{v}_t^\xi\|^{p-1}\mathbb{E}[\|\tilde{v}_t^\xi\|] \leq (1 - \frac{1}{p})\|\tilde{v}_t^\xi\|^p + \frac{1}{p}\mathbb{E}[\|\tilde{v}_t^\xi\|]^p$, we have that, for $x' \neq x$,

$$\mathbb{E}[\|\tilde{v}_t^\xi\|^p | X_0 = x'] \tag{19}$$

$$\leq \exp((2p - 1)B^2 t)\mathbb{E}[\|\tilde{v}_0^\xi\|^p | X_0 = x'] + \int_0^t \mathbb{E}[\|\tilde{v}_s^\xi\|]^p \exp[(2p - 1)B^2(t - s)]ds$$

$$\leq \int_0^t \exp(2pB^2 s) \exp[(2p - 1)B^2(t - s)]ds \mathbb{E}[\|\tilde{v}_0^\xi\|]^p$$

$$\leq \exp[(2p-1)B^2 t][\exp(B^2 t) - 1]B^{-2}\mathbb{E}[\|\tilde{v}_0^\xi\|]^p$$
$$\leq B^{-2}\exp[(2p-1)B^2 t][\exp(B^2 t) - 1]\|\xi\|^p$$
$$\leq B^{-2}[\exp(2pB^2 t) - 1]\|\xi\|^p. \tag{20}$$

In the same vein, when $X_0 = x$, we have that

$$\|\tilde{v}_t^\xi\|^p = \mathcal{O}(\exp(2pB^2 t))N^p\|\xi\|^p \tag{21}$$

for $p \geq 1$ and $t > 0$.

By Corollary 3, $m(t, \epsilon\delta_x - (1-\epsilon)\mu_0)$ has a smooth density for $t > 0$, which we denote by $\mu_t^{(\epsilon)}$. Corollary 3 also asserts that $\mu_t^{(\epsilon)}(x) > 0$ and is equivalent to $\mu^*$ for any $t > 0$. Let

$$q_{t,x}^{(1)}(x') = \frac{1}{\mu_t(x')}\xi^\top \nabla \frac{\partial}{\partial\epsilon}\mu_t^{(\epsilon)}(x')|_{\epsilon=0}.$$

For concise presentation, we introduce the abbreviated notation

$$q_{t,x}^{(1)}(\phi) := \mathbb{E}_{\mu_t}[q_{t,x}^{(1)}\phi].$$

Let $\mu_{t|x'}^\epsilon$ be the distribution of $X_t^\epsilon$ conditioned by $X_0^\epsilon = x' + \epsilon\tilde{\xi}(x')$. We define the conditional version of $q_{t,x}$ as $q_{t,x|x'}^{(1)}(x'') := \frac{1}{\mu_{t|x'}(x'')}\xi^\top \nabla \frac{\partial}{\partial\epsilon}\mu_{t|x'}^{(\epsilon)}(x'')|_{\epsilon=0}$. Accordingly, we define $q_{t,x|x'}^{(1)}(\phi) := \mathbb{E}_{\mu_{t|x'}}[q_{t,x|x'}^{(1)}\phi]$. Then, we can see that $q_{t,x}^{(1)}(\phi) = \frac{1}{N}\sum_{i=1}^N q_{t,x|\hat{X}_s^i}^{(1)}(\phi)$.

**Lemma 6** (Bismut formula). *Suppose Assumption 1 holds. Then for a bounded measurable function $f : \mathbb{R}^d \to \mathbb{R}$, $\mu_0 \in \mathcal{P}_2$ and $t > 0$, $m(t, (I + \epsilon\xi)_\sharp\mu_0)(f)$ is differentiable with respect to $\epsilon$ at $t = 0$, and we have*

$$\frac{d}{d\epsilon}m(t, (I + \epsilon\xi)_\sharp\mu_0)(f)|_{\epsilon=0} = \mathbb{E}[\nabla f(X_t) \cdot \tilde{v}_t^\xi] = \mathbb{E}\left[f(X_t)\int_0^t \zeta_s^\xi \cdot dW_s\right]$$

*where*

$$\zeta_s^\xi = \frac{(\sqrt{2\lambda})^{-1}}{t}\left(\tilde{v}_s^\xi + s\frac{\delta b}{\delta\mu}(X_s, \mu_s)(q_{t,x}^{(1)})\right).$$

*Also, $q_{t,x}^{(1)}(x') = \mathbb{E}[\int_0^t \zeta_s^\xi \cdot dW_s|X_t = x']$, $q_{t,x|x'}^{(1)}(x'') = \mathbb{E}[\int_0^t \zeta_s^\xi \cdot dW_s|X_t = x'', X_0 = x']$, and*

$$\mathbb{E}[(q_{t,x|x}^{(1)}(X_t))^2|X_0 = x] < K(t)N^2,$$

*and $\mathbb{E}[(q_{t,x|x'}^{(1)}(X_t))^2|X_0 = x'] \leq K(t)$ for $x' \neq x$, where $K(t)$ is a constant depending on $t$.*

*Proof.* The first assertion is obtained by the Bismut formula with respect to the Lions derivative by the initial distribution $\mu_0$ (Ren and Wang, 2019). In particular, Theorem 2.1 of Ren and Wang (2019) yields the assertion by setting $g_s = s/t$ in their notation. We note that although they assumed $b(x, \mu)$ and their derivatives are bounded, the same argument can be directly applied to our setting because $r(x)$ is a convex function that forces $X_t$ to contract to origin instead of diverging.

As for the second assertion, we first observe that $\frac{d}{d\epsilon}\mathbb{E}[f(X_t)] = \int f(x)\frac{\partial_\epsilon\mu_t(x)}{\mu_t(x)}d\mu_t = \int f(x)\partial_\epsilon \log\mu_t(x)d\mu_t$ for all $f$ (note that $\mu_t(x) > 0$ for all $x \in \mathbb{R}^d$ by Corollary 3). This indicates that $\partial_\epsilon \log(\mu_t)(x) = \mathbb{E}[\int_0^t \zeta_t^\xi \cdot dW_s|X_t = x]$ almost surely. In the same vein, we also have the characterization of the conditional version $q_{t,x|x'}^{(1)}(x'')$. Indeed, we may consider a dynamics of $\check{X}_t = (X_t, X_0)$ and apply the same argument on $\mu_t$ to the distribution of $\check{X}_t$.

The moment bound can be ensured by noticing that

$$\mathbb{E}[q_{t,x|x}^{(1)}(X_t)^2|X_0 = x] = \mathbb{E}\left[\mathbb{E}\left[\int_0^t \zeta_s^\xi \cdot dW_s\Big|X_t, X_0 = x\right]^2\Big|X_0 = x\right]$$

$$\leq \mathbb{E}\left[\left(\int_0^t \zeta_t^\xi \cdot dW_s\right)^2|X_0 = x\right]$$

$$\leq C_{t,2} \int_0^t \mathbb{E}[\|\zeta_s^\xi\|^2 | X_0 = x] \mathrm{d}s$$

$$\lesssim \frac{C_{t,2}}{t^2} \int_0^t (1+s)^2 \mathbb{E}[\|\tilde{v}_s^\xi\|^2 | X_0 = x] \mathrm{d}s$$

$$\lesssim \frac{C_{t,2}}{t^2} (1+t)^3 N^2 \|\xi\|^2 = O\left(\frac{C_{t,2}(1+t)^3}{t^2} N^2\right),$$

where the first inequality is due to Jensen's inequality, the second inequality is by the moment inequality of stochastic integral (Kim, 2013) ($C_{t,q} = (\frac{q(q-1)}{2})^{q/2} t^{\frac{q-2}{2}}$), and the last inequality is due to Eq. (21) and Assumption 1. When $X_0 \neq x$, Eq. (20) gives $\mathbb{E}[\|\tilde{v}_s^\xi\|^2 | X_0 = x'] = O(\exp(2pB^2t))$, which gives the bound for $X_0 \neq x$. $\qquad\square$

Let

$$\hat{L}_t \phi = \lambda \Delta \phi - b_t^\top \nabla \phi$$

for $\phi : \mathbb{R}^d \to \mathbb{R}$. Then, the derivative of $q_{t,x}^{(1)}$ with respect to $t$ can be evaluated as

$$\frac{\mathrm{d}}{\mathrm{d}t} \mathbb{E}_{\mu_t}[q_{t,x}^{(1)} f] = \frac{\partial}{\partial t} \frac{\partial}{\partial \epsilon} \mathbb{E}[f(X_t^\epsilon)]|_{\epsilon=0}$$

$$= \frac{\partial}{\partial \epsilon} \frac{\partial}{\partial t} \mathbb{E}[f(X_t^\epsilon)]|_{\epsilon=0}$$

$$= \frac{\partial}{\partial \epsilon} \mathbb{E}[(\lambda \Delta - b_t^\top \nabla) f(X_t^\epsilon)]|_{\epsilon=0}$$

$$= \int q_{t,x}^{(1)} (\lambda \Delta - b_t^\top \nabla) f - \frac{\delta b_t}{\delta \mu}(q_{t,x}^{(1)}) \nabla f \mathrm{d}\mu_t.$$

We can also see that

$$\frac{\mathrm{d}}{\mathrm{d}t} \mathbb{E}_{\mu_{t|x'}}[q_{t,x|x'}^{(1)} f] = \int q_{t,x|x'}^{(1)} (\lambda \Delta - b_t^\top \nabla) f - \frac{\delta b_t}{\delta \mu}(q_{t,x}^{(1)}) \nabla f \mathrm{d}\mu_{t|x'},$$

Note that the second term in the right hand side is $\frac{\delta b_t}{\delta \mu}(q_{t,x}^{(1)})$ instead of $\frac{\delta b_t}{\delta \mu}(q_{t,x|x'}^{(1)})$. We refer readers to Tse (2021) for higher order regularity of the nonlinear PDEs induced by the derivative with respect to the initial distribution in the torus setting. Therefore, by applying Theorem 4, we obtain the following convergence bound.

**Lemma 7.** *Under Assumption 1, it holds that, for any $t > \tau$ with sufficiently small $\tau > 0$,*

$$\mathbb{E}[(q_{t,x}^{(1)}(X_t))^2] \leq \mathcal{O}(\tilde{\Lambda}_{\mu_0} \exp(-\lambda\alpha(t-T_0)/2)\mathbb{E}[(q_{0,x}^{(1)}(X_0))^2])$$

$$= \mathcal{O}(\tilde{\Lambda}_{\mu_0} \exp(-\lambda\alpha(t-T_0)/2)N\|\xi\|^2),$$

*where $\tilde{\Lambda}_{\mu_0} = \exp(\mathcal{O}(W_2(\mu_0, \mu^*)))$.*

*Proof.* We apply Theorem 4 in Appendix C.4. We first note that the conditions of Theorem 4 hold for $\epsilon_t = C \exp(-\alpha\lambda t)W_2(\mu_0, \mu^*)$ by Corollary 4 and $\delta_t = 0$. Moreover, by Corollary 6, we can assume $\alpha_t = \alpha/2$ for $t \geq T_0 + \log(\hat{Q}_t)/(\lambda\alpha)$ and $\alpha_t = 0$ otherwise. Next, we check the integrability of $\sqrt{I(\mu_t\|\tilde{\mu}_t)}$ with respect to $t$. According to Nitanda et al. (2022); Chizat (2022),

$$\frac{\mathrm{d}}{\mathrm{d}t}(\mathcal{L}(\mu_t) - \mathcal{L}(\mu^*)) \leq -\lambda^2 I(\mu_t\|\tilde{\mu}_t) \leq 0,$$

which implies that

$$\int_{t'}^t I(\mu_s\|\tilde{\mu}_s)\mathrm{d}s \leq \frac{1}{\lambda^2}(\mathcal{L}(\mu_{t'}) - \mathcal{L}(\mu^*) - (\mathcal{L}(\mu_t) - \mathcal{L}(\mu^*))).$$

Hence,

$$\int_{t'}^t \sqrt{I(\mu_s\|\tilde{\mu}_s)}\mathrm{d}s \leq \sqrt{\frac{(t-t')}{\lambda^2}(\mathcal{L}(\mu_{t'}) - \mathcal{L}(\mu_t))}$$

$$\leq \mathcal{O}\left(\frac{1}{\lambda}\sqrt{t-t'}\exp(-\lambda\alpha(t'-\tau_0))\sqrt{\mathcal{L}(\mu_{\tau_0})-\mathcal{L}(\mu^*)}\right). \qquad (22)$$

By taking $t = k+1+\tau_0$ and $t' = k+\tau_0$ for $k = 0, 1, \dots$ and taking summation over $k$, we obtain

$$\int_{\tau_0}^{\infty}\sqrt{I(\mu_s||\tilde{\mu}_s)}\mathrm{d}s \lesssim \mathcal{O}\left(\sqrt{\mathcal{L}(\mu_{\tau_0})-\mathcal{L}(\mu^*)}\right) = \mathcal{O}(1+\sqrt{\psi_2(\tau_0)}W_2(\mu_0,\mu^*)) = \mathcal{O}_p(1), \quad (23)$$

where we omitted the dependence on $\lambda, \alpha$ in the order symbol. Therefore, Theorem 4 yields that

$$\mathbb{E}[(q_{t,x}^{(1)}(X_t))^2] \leq \mathcal{O}(\tilde{\Lambda}_{\mu_0}\exp(-\lambda\alpha(t-T_0)(3/4))\mathbb{E}[(q_{t,x}^{(1)}(X_\tau))^2]),$$

for sufficiently small $\tau > 0$, where $\tilde{\Lambda}_{\mu_0} = \exp(\mathcal{O}(W_2(\mu_0,\mu^*)))$. Combining this evaluation and Lemma 6 completes the proof. $\qquad\square$

However, observe that the bound in Lemma 7 has $\mathcal{O}(N)$ factor in the right hand side. We can remove that factor by considering $\mathbb{E}[|q_{t,x}^{(1)}(X_t)|]^2$ instead of $\mathbb{E}[q_{t,x}^{(1)}(X_t)^2]$. Recall that $\mu_0 = \mu_s^N = \frac{1}{N}\sum_{i=1}^{N}\hat{X}_s^i$. Here we define $\mu_{0\backslash x} := \frac{1}{N-1}\sum_{i=1:\hat{X}_s^i \neq x}^{N}\hat{X}_s^i$.

**Lemma 8.** *Under Assumption 1, it holds that, for any $t > \tau$ with sufficiently small $\tau > 0$,*

$$\mathbb{E}[|q_{t,x}^{(1)}(X_t)|]^2 = \mathcal{O}(\Lambda_{\mu_0}\exp(-\lambda\alpha(t-T_0)/2)\|\xi\|^2),$$

*where $\Lambda_{\mu_0} = \exp(\mathcal{O}(W_2(\mu_0,\mu^*)+W_2(\mu_{0\backslash x},\mu^*)))$.*

The proof of which can be found in Appendix C.5.

We finally remark that combining Eq. (21) and Lemma 8, we know that for any $\phi \in C_b(\mathbb{R}^d)$ and $t > 0$, it holds that

$$d^{(1)}(t;\mu_0,\xi,x)(\phi) \leq \mathcal{O}\left(\Lambda_{\mu_0}\exp(-\lambda\alpha(t-T_0)/2)\|\xi\|\|\phi\|_{\infty,1}\right), \qquad (24)$$

where $\|\phi\|_{\infty,1} = \max\{\|\phi\|_\infty, \|\nabla\phi\|_\infty\}$.

## C.3 SECOND ORDER DIFFERENTIATION

Now we evaluate the second order derivatives. Let $x_1, x_2 \in \mathbb{R}^d$ fixed. For $\epsilon = (\epsilon_1, \epsilon_2)$ with $\epsilon_k \geq 0$ and $\xi^{[1]}, \xi^{[2]} \in \mathbb{R}^d$, we note that

$$\xi^{[1]\top}\nabla_{x_1}\nabla_{x_2}^\top\frac{\delta^2}{\delta\mu^2}\mathcal{U}(t,\mu_0)(x_1,x_2)\xi^{[2]}$$

$$=\xi^{[1]\top}\nabla_{x_1}\nabla_{x_2}^\top\frac{\partial^2}{\partial\epsilon_1\partial\epsilon_2}\Phi(m(t,\epsilon_1\delta_{x_1}+\epsilon_2\delta_{x_2}-(1-\epsilon_1-\epsilon_2)\mu_0))|_{\epsilon=(0,0)}\xi^{[2]}$$

$$=\xi^{[2]\top}\nabla_{x_2}\frac{\partial}{\partial\epsilon_2}\frac{\delta}{\delta m}\Phi(m(t,\epsilon_2\delta_{x_2}+(1-\epsilon_2)\mu_0)(d^{(1)}(t;\epsilon_2\delta_{x_2}+(1-\epsilon_2)\mu_0,\xi^{[1]}))$$

$$=\frac{\delta^2}{\delta m^2}\Phi(m(t,\mu_0))(d^{(1)}(t;\mu_0,\xi^{[1]}),d^{(1)}(t;\mu_0,\xi^{[2]}))$$

$$\quad+\frac{\delta}{\delta m}\Phi(m(t,\mu_0))(d^{(2)}(t;\mu_0,\xi^{[1]},\xi^{[2]},x_1,x_2)),$$

where

$$d^{(2)}(t;\mu_0,\xi^{[1]},\xi^{[2]},x_1,x_2)(\phi)$$

$$= \xi^{[1]}\nabla_{x_1}\nabla_{x_2}^\top\frac{\partial^2}{\partial\epsilon_1\partial\epsilon_2}m(t,\epsilon_1\delta_{x_1}+\epsilon_2\delta_{x_2}-(1-\epsilon_1-\epsilon_2)\mu_0))(\phi)|_{\epsilon=(0,0)}\xi^{[2]}.$$

By Corollary 3, $m(t,\epsilon_1\delta_{x_1}+\epsilon_2\delta_{x_2}-(1-\epsilon_1-\epsilon_2)\mu_0)$ has a smooth density for $t > 0$, which is denoted by $\mu_t^{(\epsilon_1,\epsilon_2)}$. Corollary 3 also asserts that $\mu_t^{(\epsilon_1,\epsilon_2)}(x) > 0$ and equivalent to $\mu^*$ for any $t > 0$. If we write

$$q_{t,(x_1,x_2)}^{(2)}(x) = \frac{1}{\mu_t(x)}\xi^{[1]\top}\nabla\nabla^\top\frac{\partial^2}{\partial\epsilon_1\partial\epsilon_2}\mu_t^{(\epsilon_1,\epsilon_2)}(x)\xi^{[2]},$$

then we can notice that $d^{(2)}(t; \mu_0, \xi^{[1]}, \xi^{[2]}, x_1, x_2)(\phi) = \mathbb{E}_{\mu_t}[q^{(2)}_{t,(x_1,x_2)}\phi]$. Here, we denote by $d^{(1)}_{t,[k]}(\phi) = d^{(1)}(t; \mu_0, \xi^{[k]}, x_k)(\phi) = \xi^{\top}\nabla_x \frac{\partial}{\partial \epsilon} m(t, \epsilon \delta_{x_k} - (1-\epsilon)\mu_0))(\phi)|_{\epsilon=0}$. Then, by taking the derivative of $\frac{\mathrm{d}}{\mathrm{d}t}d^{(1)}(t; \cdot)$ with respect to $\epsilon_2$, we know $q^{(2)}_t$ (and $d^{(2)}$) follows the following dynamics:

$$\frac{\mathrm{d}}{\mathrm{d}t}d^{(2)}(t; \mu_0, \xi^{(1)}, \xi^{(2)}, x_1, x_2)(\phi) = \frac{\mathrm{d}}{\mathrm{d}t}\mathbb{E}_{\mu_t}[q^{(2)}_{t,(x_1,x_2)}\hat{L}_t(\phi)]$$

$$=\mathbb{E}_{\mu_t}[q^{(2)}_{t,(x_1,x_2)}\hat{L}_t(\phi)] - \int \frac{\delta b_t}{\delta \mu}(d^{(2)}(t; \mu_0, \xi^{(1)}, \xi^{(2)}, x_1, x_2)) \cdot \nabla\phi \mathrm{d}\mu_t$$

$$- d^{(1)}_{t,[1]}\left(\frac{\delta b_t}{\delta \mu}(d^{(1)}_{t,[2]}) \cdot \nabla\phi\right) - d^{(1)}_{t,[2]}\left(\frac{\delta b_t}{\delta \mu}(d^{(1)}_{t,[1]}) \cdot \nabla\phi\right) - \mathbb{E}_{\mu_t}\left[\frac{\delta^2 b_t}{\delta^2 \mu}(d^{(1)}_{t,[1]}, d^{(1)}_{t,[2]}) \cdot \nabla\phi\right],$$

and also

$$q^{(2)}_{0,(x_1,x_2)}(\cdot) = 0.$$

Now, let

$$q^{(2)}_t = \frac{1}{N}\sum_{i=1}^{N} q_{t,(\hat{X}^i_s,\hat{X}^i_s)},$$

and consider a situation where $x_1 = x_2 \in \{\hat{X}^i_0\}_{i=1}^{N}$. We write $q^{(1)}_{t,[k],i}$ to indicate $q^{(1)}_{t,\hat{X}^i_t}$ for $d^{(1)}(t; \mu_0, \xi^{[k]}, \hat{X}^i_0)$ when $x_k = \hat{X}^i_0$. Then, we have that

$$\frac{\mathrm{d}}{\mathrm{d}t}\mathbb{E}_{\mu_t}[q^{(2)}_t \hat{L}_t(\phi)]$$

$$=\mathbb{E}_{\mu_t}[q^{(2)}_t \hat{L}_t(\phi)] - \int \frac{\delta b_t}{\delta \mu}(q^{(2)}_t) \cdot \nabla\phi \mathrm{d}\mu_t$$

$$- \frac{1}{N}\sum_{i=1}^{N}\mathbb{E}_{\mu_t}\left[q^{(1)}_{t,[1],i}\frac{\delta b_t}{\delta \mu}(q^{(1)}_{t,[2],i}) \cdot \nabla\phi\right] - \frac{1}{N}\sum_{i=1}^{N}\mathbb{E}_{\mu_t}\left[q^{(1)}_{t,[2],i}\frac{\delta b_t}{\delta \mu}(q^{(1)}_{t,[1],i}) \cdot \nabla\phi\right]$$

$$- \mathbb{E}_{\mu_t}\left[\frac{\delta^2 b_t}{\delta^2 \mu}(q^{(1)}_{t,[1],i}, q^{(1)}_{t,[2],i}) \cdot \nabla\phi\right].$$

By Eq. (24),

$$- \frac{1}{N}\sum_{i=1}^{N}\mathbb{E}_{\mu_t}\left[q^{(1)}_{t,[1],i}\frac{\delta b_t}{\delta \mu}(q^{(1)}_{t,[2],i}) \cdot \nabla\phi\right] \le \mathcal{O}\left(\Lambda_{\mu_0}\exp(-\lambda\alpha(t-T_0)/2)\|\xi^{[1]}\|\sqrt{\mathbb{E}_{\mu_t}[\|\nabla\phi\|^2]}\right)$$

$$- \frac{1}{N}\sum_{i=1}^{N}\mathbb{E}_{\mu_t}\left[q^{(1)}_{t,[2],i}\frac{\delta b_t}{\delta \mu}(q^{(1)}_{t,[1],i}) \cdot \nabla\phi\right] \le \mathcal{O}\left(\Lambda_{\mu_0}\exp(-\lambda\alpha(t-T_0)/2)\|\xi^{[2]}\|\sqrt{\mathbb{E}_{\mu_t}[\|\nabla\phi\|^2]}\right).$$

In the same vein, we also have

$$-\mathbb{E}_{\mu_t}\left[\frac{\delta^2 b_t}{\delta^2 \mu}(q^{(1)}_{t,[1],i}, q^{(1)}_{t,[2],i}) \cdot \nabla\phi\right] \le \mathcal{O}\left(\Lambda^2_{\mu_0}\exp(-\lambda\alpha(t-T_0))\|\xi^{[1]}\|\|\xi^{[2]}\|\sqrt{\mathbb{E}_{\mu_t}[\|\nabla\phi\|^2]}\right).$$

Then applying Theorem 4, we obtain the following lemma.

**Lemma 9.** *Under Assumption 1, it holds that*

$$\mathbb{E}[(q^{(2)}_t(X_t))^2] \le \mathcal{O}(\Lambda_{\mu_0}\exp(-\lambda\alpha(t-T_0)/2)\|\xi^{[1]}\|\|\xi^{[2]}\|).$$

*where $\Lambda_{\mu_0} = \exp(\mathcal{O}(W_2(\mu_0, \mu^*) + W_2(\mu_{0\backslash x}, \mu^*)))$.*

*Proof.* From the argument above, the assumption in Theorem 4 guaranteed for $\delta_t = \exp(-\lambda t/2)$. The other conditions are also satisfied as in the proof of Lemma 7. Hence, for arbitrary small positive time $\tau'_0 > 0$, we have that

$$\mathbb{E}[(q^{(2)}_{\tau'_0}(X_{\tau'_0}))^2] = \mathcal{O}(1 + \mathbb{E}[(q^{(2)}_0(X_0))^2]) = \mathcal{O}(1),$$

due to Theorem 4-(i) and $q^{(2)}_0 = 0$.

On the other hand, Corollary 4 yields that $\mathcal{L}(\mu_t) - \mathcal{L}(\mu^*) \leq \lambda\psi_2(\tau_0)W_2(\mu_0,\mu^*)^2$. Combining this with the argument in Eq. (23), we see that

$$\int_{\tau_0'}^{\infty}\sqrt{I(\mu_t,\tilde{\mu}_t)}\mathrm{d}t \leq \mathcal{O}\left(1 + \sqrt{\psi_2(\tau_0')}W_2(\mu_0,\mu^*)\right).$$

Then, by the same argument as Lemma 7 and Theorem 4-(ii), we obtain the assertion. □

## C.4 GENERAL CONVERGENCE GUARANTEE

We can see that $q_t = q_{t,x}^{(1)}$ and $q_t = q_t^{(2)}$ satisfy

$$\frac{\mathrm{d}}{\mathrm{d}t}\mathbb{E}_{\mu_t}[q_t\phi] = \mathbb{E}[q_t\hat{L}_t(\phi)] - \int\frac{\delta b_t}{\delta\mu}(q_t)\cdot\nabla\phi\mathrm{d}\mu_t + r_t(\phi), \tag{25}$$

where $|r_t(\phi)| \leq C\exp(-c_0\lambda\alpha t)\sqrt{\mathbb{E}_{\mu_t}[\|\nabla\phi\|^2]}$, and with slight abuse of notation we write $q_t(\phi) := \mathbb{E}_{\mu_t}[q_t\phi]$. We define

$$D_1(t) := \frac{1}{n}\sum_{j=1}^{n}\mathbb{E}_{\mu_t}[q_th_j]^2\ell_{j,t}'', \quad D_2(t) := \int q_t^2\mathrm{d}\mu_t,$$

where $\ell_{j,t}'' = \ell_j''(f_{\mu_t})$.

In addition, recall the $\mu$ satisfies the *Poincaré inequality* (PI) with constant $\alpha$ if for all smooth functions $f : \mathbb{R}^d \to \mathbb{R}$, we have

$$\mathrm{Var}_\mu(f) \leq \alpha^{-1}\mathbb{E}_\mu[\|\nabla f\|^2].$$

It is well-known that LSI implies PI with the same constant.

The following theorem provides an upper bound on $D_1$ and $D_2$ under the Poincaré inequality.

**Theorem 4.** *Given Assumption 1 and suppose that $|\frac{\mathrm{d}}{\mathrm{d}t}\ell_{j,t}''| \leq \epsilon_t$, $q_t$ satisfies Eq. (25) with $r_t(q_t)$ satisfying $|r_t(q_t)| \leq C\sqrt{\delta_t\mathbb{E}_{\mu_t}[\|\nabla q_t\|^2]}$ with a sequence of $1/2 \geq \delta_t \geq 0$ and $C \geq 0$, and $\mu_t$ satisfies $\alpha_t$-PI for $\alpha_t \geq 0$ ($\alpha_t = 0$ is also allowed in the case that $\mu_t$ does not satisfy the LSI), then the following bounds hold:*

*(i)*

$$D_2(t) \leq \exp\left[t\left(\frac{B^4}{\lambda} + \frac{C^2}{2\lambda}\right)\right]\left(D_2(0) + \frac{C^2}{2\left(\frac{B^4}{\lambda} + \frac{C^2}{2\lambda}\right)}\right).$$

*(ii)*

$$\frac{\mathrm{d}}{\mathrm{d}t}(D_1(t) + \lambda D_2(t)) \leq -2(1-\delta_t)\lambda\alpha_t(D_1(t) + \lambda D_2(t))$$
$$+ B^2\left(2B\lambda\sqrt{I(\mu_t\|\tilde{\mu}_t)} + \epsilon_t + 2B^2\frac{\delta_t}{1-\delta_t}\right)D_2(t) + \frac{C^2}{2}\delta_t.$$

*In particular, it holds that, for $0 \leq \tau \leq t$,*

$$D_1(t) + \lambda D_2(t) \leq \int_\tau^t\frac{C^2}{2}\delta_s e^{A_t - A_s}\mathrm{d}s + \exp(A_t - A_\tau)(D_1(\tau) + \lambda D_2(\tau)),$$

*where*

$$A_s = \int_0^s -2(1-\delta_s)\lambda\alpha_s + C_1\left(\lambda\sqrt{I(\mu_s\|\tilde{\mu}_s)} + \epsilon_s + \frac{\delta_s}{1-\delta_s}\right)\mathrm{d}s,$$

*and $C_1 = \max\{2B^3, 1, 2B^4\}/\lambda$.*

*Proof.* By substituting $\phi \leftarrow q_t$, it holds that

$$\frac{\mathrm{d}}{\mathrm{d}t}\int(q_t)^2\mathrm{d}\mu_t = 2\mathbb{E}_{\mu_t}[q_t\hat{L}_t(q_t)] - \int q_t^2\frac{\partial}{\partial t}\mathrm{d}\mu_t - 2\int\frac{\delta b_t}{\delta\mu}(q_t)\cdot\nabla q_t\mathrm{d}\mu_t + 2r_t(q_t).$$

Here, the first two terms in the right hand side can be evaluated as

$$2\mathbb{E}_{\mu_t}[q_t \hat{L}_t(q_t)] - \int (q_t)^2 \frac{\partial}{\partial t} \mathrm{d}\mu_t$$

$$= 2\mathbb{E}_{\mu_t}[q_t(\lambda\Delta - b_t^\top \nabla)(q_t)] - \int (\lambda\Delta - b_t^\top \nabla)(q_t)^2 \mathrm{d}\mu_t$$

$$= 2\mathbb{E}_{\mu_t}[q_t(\lambda\Delta - b_t^\top \nabla)(q_t)] - 2\mathbb{E}_{\mu_t}[\lambda(\Delta q_t)q_t + \lambda\|\nabla q_t\|^2 - b_t^\top (\nabla q_t)q_t]$$

$$= -2\lambda\mathbb{E}_{\mu_t}[\|\nabla q_t\|^2].$$

**Part $(i)$.** By the assumption, the Cauchy-Schwarz inequality, and the arithmetic-geometric mean inequality, we can see that

$$-2\int \frac{\delta b_t}{\delta \mu}(q_t) \cdot \nabla q_t \mathrm{d}\mu_t + 2r_t(q_t)$$

$$\leq \frac{B^4}{\lambda}\mathbb{E}_{\mu_t}[q_t^2] + \lambda\mathbb{E}_{\mu_t}[\|\nabla q_t\|^2] + \frac{C^2}{\lambda}\delta_t + \lambda\mathbb{E}_{\mu_t}[\|\nabla q_t\|^2]$$

$$= 2\lambda\mathbb{E}_{\mu_t}[\|\nabla q_t\|^2] + \frac{B^4}{\lambda}\mathbb{E}_{\mu_t}[q_t^2] + \frac{C^2}{\lambda}\delta_t.$$

Therefore,

$$\frac{\mathrm{d}}{\mathrm{d}t}D_2(t) \leq \frac{B^4}{\lambda}D_2(t) + \frac{C^2}{\lambda}\delta_t.$$

This yields that

$$D_2(t) \leq \exp\left(t\frac{B^4}{\lambda} + \frac{C^2}{\lambda}\int_0^t \delta_s \mathrm{d}s\right) D_2(0) + \frac{C^2}{2}\int_0^t \delta_s \exp\left((t-s)\frac{B^4}{\lambda} + \frac{C^2}{\lambda}\int_s^t \delta_\tau \mathrm{d}\tau\right) \mathrm{d}s$$

$$\leq \exp\left[t\left(\frac{B^4}{\lambda} + \frac{C^2}{2\lambda}\right)\right] D_2(0) + \frac{C^2}{2\left(\frac{B^4}{\lambda} + \frac{C^2}{2\lambda}\right)} \exp\left[t\left(\frac{B^4}{\lambda} + \frac{C^2}{2\lambda}\right)\right].$$

This gives the first inequality.

**Part $(ii)$.** Next, we evaluate the time differentiation of $D_1$ as

$$n\frac{\mathrm{d}}{\mathrm{d}t}D_1(t)$$

$$= \frac{\mathrm{d}}{\mathrm{d}t}\sum_{j=1}^n (\mathbb{E}_{\mu_t}[q_t h_j])^2 \ell''_{j,t}$$

$$= \sum_{j=1}^n \mathbb{E}_{\mu_t}[q_t h_j]\left\{2\ell''_{j,t}\frac{\mathrm{d}}{\mathrm{d}t}\mathbb{E}_{\mu_t}[q_t h_j] + \mathbb{E}_{m_t}[q_t h_j]\frac{\mathrm{d}}{\mathrm{d}t}\ell''_{j,t}\right\}$$

$$= 2\sum_{j=1}^n \mathbb{E}_{\mu_t}[q_t h_j]\ell''_{j,t}\mathbb{E}_{\mu_t}\left[q_t(\lambda\Delta h_j - b_t^\top \nabla h_j) - \frac{\delta b_t^\top}{\delta \mu}(q_t)\nabla h_j\right] + B^2 n\epsilon_t D_2(t)$$

$$= -2\lambda\sum_{j=1}^n \mathbb{E}_{\mu_t}[q_t h_j]\ell''_{j,t}\int \nabla q_t \cdot \nabla h_j \mathrm{d}\mu_t$$

$$+ \lambda\sum_{j=1}^n \mathbb{E}_{\mu_t}[q_t h_j]\ell''_{j,t}\int q_t(b_t^\top + \lambda\nabla\log(\mu_t)) \cdot \nabla h_j \mathrm{d}\mu_t$$

$$- \frac{2}{n}\|(\ell''_{i,t}\mathbb{E}_{\mu_t}[q_t h_j])_{j=1}^n\|_{Q_t}^2 + B^2 n\epsilon_t D_2(t) \quad \left(\text{where } (Q_t)_{i,j} := \int \nabla h_i \cdot \nabla h_j \mathrm{d}\mu_t\right).$$

Here, we notice that

$$\sum_{j=1}^n \mathbb{E}_{\mu_t}[q_t h_j]\ell''_{j,t}\int q_t(b_t + \lambda\nabla\log(\mu_t)) \cdot \nabla h_j \mathrm{d}\mu_t$$

$$\leq \lambda \sum_{j=1}^{n} \sqrt{\int q_t^2 \mathrm{d}\mu_t} \sqrt{\int \|\nabla \log(\tilde{\mu}_t) - \nabla \log(\mu_t)\|^2 \mathrm{d}\mu_t} \sqrt{\int q_t^2 \mathrm{d}\mu_t} \ell_{j,t}'' \|\nabla h_j\|_\infty \|h_j\|_\infty$$

$$\leq \lambda n B^3 \sqrt{I(\mu_t \| \tilde{\mu}_t)} \mathbb{E}_{\mu_t}[q_t^2] = \lambda n B^3 \sqrt{I(\mu_t \| \tilde{\mu}_t)} D_2(t).$$

In addition, note that

$$\frac{1}{n} \sum_{j=1}^{n} \mathbb{E}_{\mu_t}[q_t h_j] \ell_{j,t}'' \int \nabla q_t \cdot \nabla h_j \mathrm{d}x = \int \nabla q_t \cdot \frac{\delta b_t}{\delta \mu}(q_t) \mathrm{d}x.$$

Therefore,

$$\frac{\mathrm{d}}{\mathrm{d}t}(D_1(t) + \lambda D_2(t))$$

$$\leq -\frac{2}{n^2} \|(\ell_{i,t}'' \mathbb{E}_{\mu_t}[q_t h_j])_{j=1}^n\|_{Q_t}^2 - 4\lambda \int \nabla q_t \cdot \frac{\delta b_t}{\delta \mu}(q_t) \mathrm{d}\mu_t - 2\lambda^2 \int \|\nabla q_t\|^2 \mathrm{d}\mu_t$$

$$+ 2B^3 \lambda \sqrt{I(\mu_t \| \tilde{\mu}_t)} D_2(t) + B^2 \epsilon_t D_2(t) + 2\lambda r_t(q_t)$$

$$\leq -\frac{2}{n^2} \|(\ell_{i,t}'' \mathbb{E}_{\mu_t}[q_t h_j])_{j=1}^n\|_{Q_t}^2 - 4\lambda \int \nabla q_t \cdot \frac{\delta b_t}{\delta \mu}(q_t) \mathrm{d}\mu_t - 2\lambda^2 \int \|\nabla q_t\|^2 \mathrm{d}\mu_t$$

$$+ 2B^3 \lambda \sqrt{I(\mu_t \| \tilde{\mu}_t)} D_2(t) + B^2 \epsilon_t D_2(t) + 2\delta_t \lambda^2 \int \|\nabla q_t\|^2 \mathrm{d}\mu_t + \frac{C^2}{2}\delta_t$$

$$= -2 \int \left\| \frac{1}{\sqrt{1-\delta_t}} \frac{\delta b_t}{\delta \mu}(q_t) + \sqrt{1-\delta_t}\lambda \nabla q_t \right\|^2 \mathrm{d}\mu_t + 2\left(\frac{1}{1-\delta_t} - 1\right) \int \left\| \frac{\delta b_t}{\delta \mu}(q_t) \right\|^2 \mathrm{d}\mu_t$$

$$+ B^2(2B\lambda\sqrt{I(\mu_t \| \tilde{\mu}_t)} + \epsilon_t) D_2(t) + \frac{C^2}{2}\delta_t$$

$$\leq -2 \int \left\| \frac{1}{\sqrt{1-\delta_t}} \frac{\delta b_t}{\delta \mu}(q_t) + \sqrt{1-\delta_t}\lambda \nabla q_t \right\|^2 \mathrm{d}\mu_t$$

$$+ B^2\left(2B\lambda\sqrt{I(\mu_t \| \tilde{\mu}_t)} + \epsilon_t + 2B^2\frac{\delta_t}{1-\delta_t}\right) D_2(t) + \frac{C^2}{2}\delta_t.$$

When $\mu_t$ satisfies $\alpha_t$-PI (which is implied by $\alpha_t$-LSI), it holds that

$$\alpha_t \int \left[ \theta_1 \frac{1}{n} \sum_{j=1}^{n} \ell_{j,t}'' h_j \mathbb{E}_{\mu_t}[q_t h_j] + \theta_2 \lambda q_t \right]^2 \mathrm{d}\mu_t \leq \int \left\| \theta_1 \frac{\delta b_t}{\delta \mu}(q_t) + \theta_2 \lambda \nabla q_t \right\|^2 \mathrm{d}\mu_t$$

for any $\theta_1, \theta_2 > 0$, which gives that

$$\frac{\mathrm{d}}{\mathrm{d}t}(D_1(t) + \lambda D_2(t))$$

$$\leq -2\alpha_t \int \left[ \frac{1}{\sqrt{1-\delta_t}} \frac{1}{n} \sum_{j=1}^{n} \ell_{j,t}'' h_j \mathbb{E}_{\mu_t}[q_t h_j] + \sqrt{1-\delta_t}\lambda q_t \right]^2 \mathrm{d}\mu_t$$

$$+ B^2\left(2B\lambda\sqrt{I(\mu_t \| \tilde{\mu}_t)} + \epsilon_t + 2B^2\frac{\delta_t}{1-\delta_t}\right) D_2(t) + \frac{C^2}{2}\delta_t$$

$$\leq -2\alpha_t \frac{1}{1-\delta_t} \frac{1}{n^2} \sum_{i,j=1}^{n} \ell_{i,t}'' \ell_{j,t}'' \mathbb{E}_{\mu_t}[q_t h_i] \mathbb{E}_{\mu_t}[q_t h_j] \int h_i h_j \mathrm{d}\mu_t$$

$$- 4\lambda\alpha_t \frac{1}{n} \sum_{j=1}^{n} \ell_{j,t}'' \mathbb{E}_{\mu_t}[q_t h_j] \mathbb{E}_{\mu_t}[q_t h_j] - (1-\delta_t)2\alpha_t \lambda^2 \int q_t^2 \mathrm{d}\mu_t$$

$$+ B^2\left(2B\lambda\sqrt{I(\mu_t \| \tilde{\mu}_t)} + \epsilon_t + 2B^2\frac{\delta_t}{1-\delta_t}\right) D_2(t) + \frac{C^2}{2}\delta_t$$

$$\leq -2(1-\delta_t)\lambda\alpha_t(D_1(t) + \lambda D_2(t))$$

$$+ B^2 \left( 2B\lambda\sqrt{I(\mu_t||\tilde{\mu}_t)} + \epsilon_t + 2B^2 \frac{\delta_t}{1-\delta_t} \right) D_2(t) + \frac{C^2}{2}\delta_t.$$

Using $\lambda D_2(t) \le D_1(t) + \lambda D_2(t)$ and Grönwall's inequality (Mischler, 2019), we arrive at

$$D_1(t) + \lambda D_2(t) \le \int_0^t \frac{C^2}{2}\delta_s e^{A_t - A_s} \mathrm{d}s + \exp(A_t)(D_1(0) + \lambda D_2(0)),$$

where

$$A_s = \int_0^s -2(1-\delta_s)\lambda\alpha_s + C_1 \left( \lambda\sqrt{I(\mu_s||\tilde{\mu}_s)} + \epsilon_s + \frac{\delta_s}{1-\delta_s} \right) \mathrm{d}s,$$

for a constant $C_1 = \max\{2B^3, 1, 2B^4\}/\lambda$. $\hfill\square$

## C.5 Proof of Lemma 8

Recall that in Lemma 7 we obtained a bound on $\mathbb{E}[(q_{t,x}^{(1)}(X_t))^2]$ that contains a factor of $N$. Now we refine this result by considering a bound $\mathbb{E}[q_{t,x}^{(1)}(X_t)\phi] \le \mathbb{E}[|q_{t,x}^{(1)}(X_t)|]\|\phi\|_\infty$.

Define the events $\mathcal{I}_1 := \{X_0 = x\}$ and $\mathcal{I}_1^c := \{X_0 \ne x\}$. We let $\mu_{t|\mathcal{I}_1}$ be the conditional distribution of $X_t$ conditioned by $\mathcal{I}_1$ and $\mu_{t|\mathcal{I}_1^c}$ be that conditioned by $\mathcal{I}_1^c$. For notation simplicity, we write $q_t = q_{t,x}^{(1)}$, $q_{t|\mathcal{I}_1} = q_{t,x|x}^{(1)}$ and $q_{t|\mathcal{I}_1^c}(\cdot) = \sum_{x' \ne x} P(X_0 = x'|X_t = \cdot)q_{t,x|x'}^{(1)}(\cdot) = \sum_{x' \ne x} \frac{\mu_{t|x'}(\cdot)}{\sum_{x'' \ne x}\mu_{t|x''}(\cdot)}q_{t,x|x'}^{(1)}(\cdot)$. Accordingly, we write $q_{t|\mathcal{I}_1}(\phi) := \mathbb{E}_{\mu_{t|\mathcal{I}_1}}[q_{t|\mathcal{I}_1}\phi]$ and $q_{t|\mathcal{I}_1^c}(\phi) := \mathbb{E}_{\mu_{t|\mathcal{I}_1^c}}[q_{t|\mathcal{I}_1^c}\phi]$ for a test function $\phi$.

We control $\mathbb{E}[|q_{t,x}^{(1)}(X_t)|]$ by utilizing the following bound:

$$\mathbb{E}[|q_{t,x}^{(1)}(X_t)|] \le \frac{1}{N}\sqrt{\mathbb{E}[(q_{t,x|\mathcal{I}_1}^{(1)}(X_t))^2|X_0 = x]} + \frac{N-1}{N}\sqrt{\mathbb{E}[(q_{t,x|\mathcal{I}_1^c}^{(1)}(X_t))^2|X_0 \ne x]}. \quad (26)$$

We first evaluate $\mathbb{E}[(q_{t,x|\mathcal{I}_1}^{(1)}(X_t))^2|X_0 = x]$. If we let $D_2(t) = \mathbb{E}_{\mu_t}[q_t^2]$, then it holds that, for a positive sequence $\delta_t > 0$,

$$\begin{aligned}
\frac{\mathrm{d}}{\mathrm{d}t}\mathbb{E}_{\mu_{t|\mathcal{I}_1}}[q_{t|\mathcal{I}_1}^2] &= -2\lambda\mathbb{E}_{\mu_{t|\mathcal{I}_1}}[\|\nabla q_{t|\mathcal{I}_1}\|^2] - 2\int \frac{\delta b_t}{\delta\mu}(q_t) \cdot \nabla q_{t|\mathcal{I}_1}\mathrm{d}\mu_{t|\mathcal{I}_1} \\
&\le -2\lambda\mathbb{E}_{\mu_{t|\mathcal{I}_1}}[\|\nabla q_{t|\mathcal{I}_1}\|^2] + 2B^2\sqrt{D_2(t)}\sqrt{\mathbb{E}_{\mu_{t|\mathcal{I}_1}}[\|\nabla q_{t|\mathcal{I}_1}\|^2]} \\
&\le -2\lambda(1-\delta_t)\mathbb{E}_{\mu_{t|\mathcal{I}_1}}[\|\nabla q_{t|\mathcal{I}_1}\|^2] + B^4 D_2(t)/(2\delta_t) \\
&\le -2\lambda\alpha_t(1-\delta_t)\mathbb{E}_{\mu_{t|\mathcal{I}_1}}[q_{t|\mathcal{I}_1}^2] + B^4 D_2(t)/(2\delta_t).
\end{aligned}$$

Thus, Grönwall's inequality (see also the proof of Theorem 4) yields that

$$\mathbb{E}_{\mu_{t|\mathcal{I}_1}}[q_{t|\mathcal{I}_1}^2] \le \int_\tau^t \frac{B^4 D_2(s)}{2\delta_s}\exp(A_t - A_s)\mathrm{d}s + \exp(A_t)\mathbb{E}_{\mu_{\tau|\mathcal{I}_1}}[q_{\tau|\mathcal{I}_1}^2],$$

where $A_s = \int_0^s -2(1-\delta_s)\lambda\alpha_s\mathrm{d}s$. Here, we recall that $D_2(t) \lesssim \tilde{\Lambda}_{\mu_0} N \exp(-\alpha\lambda(t-T_0)(3/4))\|\xi\|^2$ by Lemma 7. Hence, if we set $\delta_t = \exp(-\alpha\lambda(t-T_0)(1/8))$, then $A_s \le -2\lambda\alpha(s-T_0)_+ + C$ for a constant $C$ which can depend on $\alpha, \lambda, T_0$. This argument and Lemma 6 give

$$\mathbb{E}_{\mu_{t|\mathcal{I}_1}}[q_{t|\mathcal{I}_1}^2] \lesssim \tilde{\Lambda}_{\mu_0} N \exp(-\alpha\lambda(t-T_0)(5/8))\|\xi\|^2. \quad (27)$$

Next, we evaluate $\mathbb{E}[(q_{t,x|\mathcal{I}_1^c}^{(1)}(X_t))^2|X_0 \ne x]$. Let $D_{1,c}(t) := \frac{1}{n}\sum_{j=1}^n q_{t|\mathcal{I}_1^c}(h_j)^2\ell_{j,t}''$ and $D_{2,c}(t) := \int q_{t|\mathcal{I}_1^c}^2\mathrm{d}\mu_{t|\mathcal{I}_1^c}$. For $D_{1,c}(t) + D_{2,c}(t)$, we can see that

$$\sum_{j=1}^n \mathbb{E}_{\mu_{t|\mathcal{I}_1^c}}[q_{t|\mathcal{I}_1^c}h_j]\ell_{j,t}''\int q_{t|\mathcal{I}_1^c}(b_t + \lambda\nabla\log(\mu_{t|\mathcal{I}_1^c})) \cdot \nabla h_j\mathrm{d}\mu_{t|\mathcal{I}_1^c}$$

$$\leq \lambda \sum_{j=1}^{n} \sqrt{D_{2,c}(t)} \sqrt{\int q_{t|\mathcal{I}_1^c}^2 \mathrm{d}\mu_{t|\mathcal{I}_1}} \sqrt{\int \|\nabla \log(\tilde{\mu}_t) - \nabla \log(\mu_{t|\mathcal{I}_1^c})\|^2 \mathrm{d}\mu_{t|\mathcal{I}_1^c}} \ell_{j,t}'' \|\nabla h_j\|_\infty \|h_j\|_\infty$$

$$\leq \lambda n B^3 \underbrace{\sqrt{I(\mu_{t|\mathcal{I}_1^c}\|\tilde{\mu}_t)}}_{=:S_t} D_{2,c}(t).$$

We also notice that

$$\sum_{j=1}^{n} q_{t|\mathcal{I}_1^c}(h_j)^2 \frac{\mathrm{d}}{\mathrm{d}t}\ell_{j,t}'' \leq B^2 \epsilon_t D_{2,c}(t).$$

Hence by the same reasoning as the proof of Theorem 4, we have

$$\frac{\mathrm{d}}{\mathrm{d}t}(D_{1,c}(t) + \lambda D_{2,c})$$

$$= -\frac{2}{n^2}\|(\ell_{j,t}''\mathbb{E}_{\mu_{t|\mathcal{I}_1^c}}[q_{t|\mathcal{I}_1^c}h_j])_{j=1}^n\|_{Q_t}^2 - 2\lambda \int \frac{\delta b_t}{\delta \mu}(q_t) \cdot \nabla q_{t|\mathcal{I}_1^c} \mathrm{d}\mu_{t|\mathcal{I}_1^c} + B^2(2B\lambda S_t + \epsilon_t)D_{2,c}(t)$$

$$\quad - 2\lambda^2 \mathbb{E}_{\mu_{t|\mathcal{I}_1^c}}[\|\nabla q_{t|\mathcal{I}_1^c}\|^2] - 2\lambda \int \frac{\delta b_t}{\delta \mu}(q_t) \cdot \nabla q_{t|\mathcal{I}_1^c} \mathrm{d}\mu_{t|\mathcal{I}_1^c}$$

$$= -2\frac{N-1}{N}\int \left\|\frac{\delta b_t}{\delta \mu}(q_{t|\mathcal{I}_1^c}) + \lambda \nabla q_{t|\mathcal{I}_1^c}\right\|^2 \mathrm{d}\mu_{t|\mathcal{I}_t^c} + B^2(2\lambda S_t + \epsilon_t)D_{2,c}(t)$$

$$\quad - 2\frac{1}{N}\int \left\|\frac{\delta b_t}{\delta \mu}(q_{t|\mathcal{I}_1^c})\right\|^2 \mathrm{d}\mu_{t|\mathcal{I}_1} - 4\frac{\lambda}{N}\int \frac{\delta b_t}{\delta \mu}(q_{t|\mathcal{I}_1})\nabla q_{t|\mathcal{I}_1^c}\mathrm{d}\mu_{t|\mathcal{I}_1^c} - 2\frac{\lambda^2}{N}\int \|\nabla q_{t|\mathcal{I}_1^c}\|^2\mathrm{d}\mu_{t|\mathcal{I}_1^c}$$

$$\leq -2\frac{N-1}{N}\int \left\|\frac{\delta b_t}{\delta \mu}(q_{t|\mathcal{I}_1^c}) + \lambda \nabla q_{t|\mathcal{I}_1^c}\right\|^2 \mathrm{d}\mu_{t|\mathcal{I}_t^c} + B^2(2\lambda S_t + \epsilon_t)D_{2,c}(t)$$

$$\quad + 4\lambda B\left(\frac{1}{N}\mathbb{E}_{\mu_{t|\mathcal{I}_1}}\left[|q_{t|\mathcal{I}_1}h_j|\right]\right)\sqrt{\int \|\nabla q_{t|\mathcal{I}_1^c}\|^2\mathrm{d}\mu_{t|\mathcal{I}_1^c}}$$

$$\leq -2\frac{N-1}{N}\int \left\|\frac{\delta b_t}{\delta \mu}(q_{t|\mathcal{I}_1^c}) + \lambda \nabla q_{t|\mathcal{I}_1^c}\right\|^2 \mathrm{d}\mu_{t|\mathcal{I}_t^c} + B^2(2\lambda S_t + \epsilon_t)D_{2,c}(t)$$

$$\quad + 4\lambda B^2\sqrt{\frac{1}{N^2}\mathbb{E}_{\mu_{t|\mathcal{I}_1}}[q_{t|\mathcal{I}_1}^2]}\sqrt{\int \|\nabla q_{t|\mathcal{I}_1^c}\|^2\mathrm{d}\mu_{t|\mathcal{I}_1^c}}$$

$$\leq -2\frac{N-1}{N}\int \left\|\frac{\delta b_t}{\delta \mu}(q_{t|\mathcal{I}_1^c}) + \lambda \nabla q_t\right\|^2 \mathrm{d}\mu_{t|\mathcal{I}_t^c} + B^2(2\lambda S_t + \epsilon_t)D_{2,c}(t)$$

$$\quad + 2\lambda B^2 \mathcal{O}\left(\sqrt{\tilde{\Lambda}_{\mu_0}}\exp(-\lambda\alpha(t-T_0)(5/16))\|\xi\|\right)\sqrt{\int \|\nabla q_{t|\mathcal{I}_1^c}\|^2\mathrm{d}\mu_{t|\mathcal{I}_1^c}},$$

where we used Eq. (27) in the last inequality. Then, by noticing that the same argument as Corollary 6 holds for the conditional distribution $\mu_{t|\mathcal{I}_1^c}$ since Eq. (17) holds uniformly for any $x \in \{\hat{X}_s^i\}_{i=1}^N$, we may apply the same reasoning as the proof of Theorem 4 to obtain that, for sufficiently small $\tau > 0$, the following holds

$$D_{1,c}(t) + \lambda D_{2,c}(t)$$

$$\lesssim \tilde{\Lambda}_{\mu_0}\exp[-\alpha\lambda(t-T_0)/2 + C(1 + W_2(\mu_{0|\mathcal{I}_1^c}, \mu^*))](\|\xi\|^2 + D_{1,c}(\tau) + D_{2,c}(\tau))$$

$$\lesssim \exp(\mathcal{O}(W_2(\mu_0, \mu^*) + W_2(\mu_{0|\mathcal{I}_1^c}, \mu^*)))\exp(-\alpha\lambda(t-T_0)/2)\|\xi\|^2, \quad (28)$$

where we used Lemma 6 in the last inequality.

Finally, by combining the inequalities (27) and (28) to Eq. (26), we obtain the assertion.

## C.6 COMBINING ALL BOUNDS TOGETHER

Recall that

$$\mathbb{E}[\mathcal{U}(0, \mu_t^N) - \mathcal{U}(t, \mu_0^N)] = \frac{1}{N}\sum_{i=1}^{d}\int_0^t \mathbb{E}\left[\int \left(\partial_{(x_1)_i}\partial_{(x_2)_i}\frac{\delta^2 \mathcal{U}}{\delta\mu^2}(t-s, \mu_s^N)(x, x)\right)\mu_s^N(\mathrm{d}x)\right]\mathrm{d}s.$$

Hence, by applying Lemma 7 and 9 with $\mu_0 = \mu_s^N, \xi = \xi^{[1]} = \xi^{[2]} = e_i$, we obtain

$$\int \left( \partial_{(x_1)_i} \partial_{(x_2)_i} \frac{\delta^2 \mathcal{U}}{\delta \mu^2}(t-s, \mu_s^N)(x, x) \right) \mu_s^N(\mathrm{d}x)$$

$$= \int \frac{\delta^2 \Phi}{\delta \mu^2}(m(t-s, \mu_s^N))(d^{(1)}(t-s; \mu_s^N, \xi, x), d^{(1)}(t-s; \mu_s^N, \xi, x))$$

$$+ \frac{\delta \Phi}{\delta \mu}(m(t-s, \mu_s^N))(d^{(2)}(t-s; \mu_s^N, \xi, \xi, x, x)) \mu_s^N(\mathrm{d}x)$$

$$= \mathcal{O}(\Lambda_{\mu_s^N} \exp(-\lambda \alpha (t - s - T_0)/2)),$$

where $\Lambda_{\mu_s^N} = \frac{1}{N} \sum_{i=1}^{N} \exp(\mathcal{O}(W_2(\mu_s^N, \mu^*) + W_2(\mu_{s \backslash \hat{X}_s^i}^N, \mu^*)))$. Now we only need to evaluate the term $\mathbb{E}[\Lambda_{\mu_s^N}]$.

Suppose $C$ is a constant such that $\Lambda_{\mu_s^N} \leq \frac{1}{N} \sum_{i=1}^{N} \exp(C(W_2(\mu_s^N, \mu^*) + W_2(\mu_{s \backslash \hat{X}_s^i}^N, \mu^*)))$. Then, we can verify $\Lambda_{\mu_s^N} \leq \exp\left(3C\sqrt{\mathbb{E}_{\mu_s^N}[\|X\|^2] + \mathbb{E}_{\mu^*}[\|X\|^2]}\right)$. Since $\mu^*(x) \lesssim \exp(-\lambda c_r \|x\|^{2+\delta})$, $\mathbb{E}_{\mu^*}[\|X\|^2] = \mathcal{O}(1)$ and thus we need to evaluate $\mathbb{E}\left[\exp\left(C\sqrt{\mathbb{E}_{\mu_s^N}[\|X\|^2]}\right)\right]$. This can be upper bounded by $\mathbb{E}\left[\exp\left(\frac{9C^2}{2} + \frac{1}{2}\mathbb{E}_{\mu_s^N}[\|X\|^2]\right)\right] = \mathbb{E}\left[\exp\left(\frac{9C^2}{2} + \frac{1}{2N} \sum_{i=1}^{N} \|\hat{X}_s^i\|^2\right)\right]$. Let $Q_s := \frac{1}{N} \sum_{i=1}^{N} \|\hat{X}_s^i\|^2$. By Ito's formula, we have

$$\frac{\mathrm{d}}{\mathrm{d}t} \mathbb{E}\left[\exp\left(\frac{1}{2N} \sum_{i=1}^{N} \|\hat{X}_t^i\|^2\right)\right]$$

$$= \mathbb{E}\left[\exp\left(Q_t/2\right)\left(-\frac{1}{N} \sum_{i=1}^{N} b(\hat{X}_t^i, \mu_t^N)^\top \hat{X}_t^i + \frac{\lambda}{2N} \sum_{i=1}^{N}(d + \|\hat{X}_t^i\|^2)\right)\right]$$

$$\leq \mathbb{E}\left[\exp\left(Q_t/2\right) \frac{1}{N} \sum_{i=1}^{n}\left(B^2\|\hat{X}_t^i\| - (2+\delta)\lambda c_r\|\hat{X}_t^i\|^{2+\delta} + \frac{\lambda}{2}(d + \|\hat{X}_t^i\|^2)\right)\right].$$

Then, by Young's inequality, there exists a constant $C'$ depending on $B, \lambda, \delta, c_r, d$ such that

$$B^2\|\hat{X}_t^i\| - (2+\delta)\lambda c_r\|\hat{X}_t^i\|^{2+\delta} + \frac{\lambda}{2}(d + \|\hat{X}_t^i\|^2) \leq C' - \frac{(2+\delta)\lambda c_r}{2}\|\hat{X}_t^i\|^{2+\delta}.$$

Moreover, by Jensen's inequality, the term related to $\|\hat{X}_t^i\|^{2+\delta}$ can be further bounded by

$$-\frac{1}{N} \sum_{i=1}^{n} \|\hat{X}_t^i\|^{2+\delta} \leq -\left(\frac{1}{N} \sum_{i=1}^{n} \|\hat{X}_t^i\|^2\right)^{(2+\delta)/2} = -Q_t^{(2+\delta)/2}.$$

In summary, we arrive at

$$\frac{\mathrm{d}}{\mathrm{d}t} \mathbb{E}\left[\exp\left(Q_t/2\right)\right] \leq \mathbb{E}[\exp(Q_t/2)(C' - \epsilon Q_t^{(2+\delta)/2})]$$

with another constant $\epsilon = \frac{(2+\delta)\lambda c_r}{2}$. Here, by noticing that

$$C' - \epsilon Q_t^{(2+\delta)/2} \leq 2C'\mathbf{1}[Q_t \leq (2C'/\epsilon)^{2/(2+\delta)}] - C',$$

it holds that

$$\frac{\mathrm{d}}{\mathrm{d}t} \mathbb{E}\left[\exp\left(Q_t/2\right)\right] \leq \mathbb{E}[\exp(Q_t/2)(2C'\mathbf{1}[Q_t \leq (2C'/\epsilon)^{2/(2+\delta)}] - C')]$$

$$\leq \mathbb{E}\left\{2C' \exp[\tfrac{1}{2}(2C'/\epsilon)^{2/(2+\delta)}] - C' \exp(Q_t/2)\right\}.$$

This means that

$$\mathbb{E}[\exp(Q_t/2)] \leq \max\{2\exp[\tfrac{1}{2}(2C'/\epsilon)^{2/(2+\delta)}], \quad \mathbb{E}[\exp(Q_0/2)]\} = \mathcal{O}(1).$$

One side remark is that we can further show an exponential decay of the term related to $\mathbb{E}[\exp(Q_0/2)]$, but the above is sufficient for our purpose.

Finally, we arrive at

$$\mathbb{E}[\mathcal{U}(0, \mu_t^N) - \mathcal{U}(t, \mu_0^N)] = \frac{1}{N}\mathcal{O}\left(\mathbb{E}[\Lambda_{\mu_s^N}]\int_0^t \exp(-\lambda\alpha(t - s - T_0)/2)\mathrm{d}s\right)$$

$$\leq \frac{1}{N}\mathcal{O}\left(\exp(\lambda\alpha T_0/2)\frac{1}{\lambda\alpha}(1 + \mathbb{E}[\Lambda_{\mu_0^N}])\right).$$

By selecting the initial distribution so that $\mathbb{E}[\Lambda_{\mu_0^N}] = \mathcal{O}(1)$, the right hand side is $\mathcal{O}(1)$. This is satisfied if the support of $\mu_0$ is bounded, e.g., $\mu_0^N = \frac{1}{N}\sum_{i=1}^N \delta_0$.

## D    AUXILIARY LEMMAS

**Lemma 10.** *Suppose that $\Psi : \mathcal{P} \to \mathbb{R}$ has smooth first-variation such that $\|\nabla\frac{\delta\Psi}{\delta\mu}\|_\infty \leq C$. Then, for any $\nu_0, \nu_1 \in \mathcal{P}_2$, it holds that*

$$|\Psi(\nu_1) - \Psi(\nu_2)| \leq CW_2(\nu_0, \nu_1).$$

*Proof.* By the Benamou–Brenier formula, for any $\nu_1, \nu_0 \in \mathcal{P}_2$, it holds that

$$W_2^2(\nu_1, \nu_0) = \inf\left\{\int_0^1 \int \|v_t\|^2 \mathrm{d}\nu_t \mathrm{d}t \mid \partial_t\nu_t + \nabla\cdot(\nu_t v_t) = 0 \ (t \in [0, 1])\right\},$$

where the infimum is taken over all curves $\nu_t : [0, 1] \to \mathcal{P}_2$ continuous with respect to the weak topology (Ambrosio et al., 2005, Chapter 8). Then, for any $v_t$ satisfying $\partial_t\nu_t + \nabla\cdot(\nu_t v_t) = 0$ ($t \in [0, 1]$), we note that $\frac{\mathrm{d}\Phi(\nu_t)}{\mathrm{d}t} = \int v_t^\top\nabla\frac{\delta\Psi}{\delta\mu}(\nu_t)\mathrm{d}\nu_t$, and thus

$$|\Psi(\nu_1) - \Psi(\nu_0)|$$
$$= \left|\int_0^1 \int v_t^\top\nabla\frac{\delta\Psi}{\delta\mu}(\nu_t)\mathrm{d}\nu_t\mathrm{d}t\right|$$
$$\leq \left|\int_0^1 \int \|v_t\|\left\|\nabla\frac{\delta\Psi}{\delta\mu}(\nu_t)\right\|\mathrm{d}\nu_t\mathrm{d}t\right|$$
$$\leq C\sqrt{\int_0^1 \int \|v_t\|^2\mathrm{d}\nu_t\mathrm{d}t}.$$

By taking infimum with respect to $v_t$, we obtain the inequality. $\square$

$f_{\boldsymbol{X}_t}$ in the informal theorem stated in Section 1 formally corresponds to $f_{\mu_t^N}$ in the main text. Then, we have the following corollary as a direct consequence of Theorem 2.

**Corollary 7.** *Under the same setting as Theorem 2, we have*

$$\mathbb{E}[(f_{\mu_t}(z) - f_{\mu_t^N}(z))^2] = \mathcal{O}(1/N),$$

*where $\mu_t = m(t, \mu_0^N)$, and the expectation is taken over the dynamics of $\mu_t^N$ with a fixed initial condition $\mu_0^N$.*

*Proof.* By conditioning the initial distribution $\mu_0^N$, the evolution of $\mu_t$ is deterministic. Hence, for fixed $z \in \mathbb{R}^{d'}$, it holds that

$$\mathbb{E}[(f_{\mu_t^N}(z) - f_{\mu_t}(z))^2]$$
$$= \mathbb{E}[(f_{\mu_t^N}(z) - f_{\mu_t}(z))^2]$$
$$+ 2\mathbb{E}_X[f_{\mu_t^N}(z) - f_{\mu_t}(z)](f_{\mu_t}(z) - f^*(z)) - 2\mathbb{E}[f_{\mu_t^N}(z) - f_{\mu_t}(z)](f_{\mu_t}(z) - f^*(z))$$

$$+ (f_{\mu_t}(z) - f^*(z))^2 - (f_{\mu_t}(z) - f^*(z))^2$$
$$= \mathbb{E}[(f_{\mu_t^N}(z) - f^*(z))^2] - (f_{\mu_t}(z) - f^*(z))^2$$
$$- 2(\mathbb{E}[f_{\mu_t^N}(z)] - f_{\mu_t}(z))(f_{\mu_t}(z) - f^*(z)).$$

Therefore, by considering $\Phi(\mu) = f_\mu(z)$ and $\Phi(\mu) = (f_\mu(z) - f^*(z))^2$, Theorem 2 indicates that

$$\mathbb{E}[(f_{\mu_t^N}(z) - f^*(z))^2] - (f_{\mu_t}(z) - f^*(z))^2 = \mathcal{O}(1/N), \;\; \mathbb{E}[f_{\mu_t^N}(z)] - f_{\mu_t}(z) = \mathcal{O}(1/N),$$

which yields the assertion. □

