# OpenReview forum: "Uniform-in-time propagation of chaos for the mean-field gradient Langevin dynamics"
_ICLR.cc/2023/Conference — ICLR 2023 poster_

### Official Review · Reviewer_713A · 2022-10-20

**Confidence:** 3
**Correctness:** 4
**Technical Novelty And Significance:** 3
**Empirical Novelty And Significance:** Not applicable
**Recommendation:** 8

**Clarity, Quality, Novelty And Reproducibility:**

**Quality, Novelty:** yes.

**Clarity:** In general clear presentation of context and results, however the central technical advance is advertised but not clearly detailed (see weaknesses).

**Reproducibility:** The calculations are clear and reproducible. The code for the numerical experiments should be made available.


**Additional small comments:**  There are a number possible typos \
 -p.1: $h_z(x)=\tanh(r\sigma(w^Tx)) $ seems a typo (while on p.5 below A1 $\tanh(r)\sigma(w^Tx)$ is given) \
-p.2: "using neural network"; "$M$ neurons" ; also $x\in \mathbb{R}^p$ and $X \subset \mathbb{R}^d $ should both be $\mathbb{R}\times\mathbb{R}^d$? \
-p.4: "provides meaningful guarantee" \
-p.5 in Def.1: $\mathbb{R}^p$ should be $\mathbb{R}^d$? \
-p.6: "under quadratic regularizer" \
-p.9: "[our] analysis do not cover" \
-p.13: "xB"; "having form"; "such the"; "(1) uniform"; "(2) local"; "[they showed a..] but it also requires" \
-p.20: "We also a"; "for related quantity" \


**Strength And Weaknesses:**

**Strengths:**
 - Technically demanding proof with relevant outcome (1/N scaling) under mild conditions.

**Weaknesses:**
 - The progress in proof technique is stated multiple times in abstract/intro/conclusion (not requiring weak interaction while showing uniform propagation of chaos), but not technically discussed/explained in the relevant technical section 4 or appendices.



**Summary Of The Paper:**

The paper considers two-layer networks with one output dimension in the mean-field parametrization. In the limit of infinite hidden-layer width, these can be described in the infinite dimensional space of parameter distributions (across hidden unit weights) by a nonlinear Fokker-Plank equation, which has been done in the past. The paper justifies the relevance of this approach by computing bounds on the difference between finite-size discretization and the infinite-width limit, showing that the difference in any sufficiently smooth test function and for any time in the training process scales as $1/N$, given relatively mild assumptions. This extends previous finite-size bounds which suffered from an exponential dependence on training time.

**Summary Of The Review:**

This paper seems mathematically sound and provides a relevant result for the study of feature learning in two layer networks in MF-parametrization. It is suitable for ICLR. The technical description of the novelty in approach should be improved.

---

> ### Author Response · Authors · 2022-11-13
> **Reply to Reviewer 713A**
>
> Thank you for the positive evaluation and helpful feedback. We address the technical comments below.
>
> **Overview of proof strategy.**
>
> Considering the page limit and simplicity of presentation, we decided to skip mathematical details on how the super log-Sobolev inequality is used to relax the small interaction condition.
> Following your suggestion, we have added an overview of the proof just before Section 5.1 of the revised version.
> We also included more discussions on the technical details on our contribution in Section 5.2.
>
> A rough description of the proof strategy is as follows. First, the super log-Sobolev inequality of the proximal Gibbs distributions $\tilde{\mu}_t$ (Lemma 2) entails the density ratio bound between $\tilde{\mu}_t$ and $\mu_t$. This enables us to prove the log-Sobolev inequality of $\mu_t$ (Corollary 1). A key consequence is that this condition enables us to show exponential convergence of $d^{(1)}$ and $d^{(2)}$. Intuitively, the LSI condition ensures that the KL-divergence from $\mu_t$ behaves like a strongly convex function. Therefore, a small perturbation from $\mu_t$ exponentially converges to 0 as $t$ grows, which essentially implies the convergence of $d^{(1)}$ and $d^{(2)}$ because they represent infinitesimal displacement of $\mu_t$.
>
> **Typos**
>
> Thank you pointing out the typos. We have revised the manuscript accordingly. The followings are some additional comments on the typos:
> *  *p.1, the definition of $h_z(x)$:* Indeed our first example can be confusing. We changed this to the more standard two-layer neural network in the mean-field parameterization $h_z(x) = \sigma(w^\top z + b)$, which is also included in the examples presented just after Assumption 1.
> * *Confusing notations for dimensionality of $z$ and $x$:* We fixed them so that $d'$ denotes the dimensionality of $z$ (input) and $d$ denotes that of $x$ (parameter).
>
> We would be happy to clarify any concerns or answer any questions that may come up during the discussion period.

---

### Official Review · Reviewer_FGcq · 2022-10-23

**Confidence:** 3
**Correctness:** 2
**Technical Novelty And Significance:** 2
**Empirical Novelty And Significance:** Not applicable
**Recommendation:** 5

**Clarity, Quality, Novelty And Reproducibility:**

I think the writing and presentation of the document should be improved. In particular, section 4.1.2, which is two pages of the main document, is in my opinion too difficult to parse.
In addition, Some details which are given are inessential whereas more would have required more care.

- For example, it is a bit odd in my opinion to note that a distribution is also called a probability law (p4).
- Why referring particularly to Huang for existence and uniqueness of solutions for (3) while no assumption and detail are given and there are many other results on this topic.
- The use of both $\lambda$ and $\lambda_1$ is really necessary? I think you could only introduce one of them since you work with a generic $r$.
- From assumption on $r$, it is hard to guess functions satisfying such a condition except $\Vert x\Vert^{2\alpha}$, for $\alpha >1$.
- The authors make use of higher order derivative of function $\Phi :\mathcal{P} \to \mathbb{R}$ with respect to a measure $\mu$ but I did not catch how it is defined only based on the definition of the first order derivative. In fact one of the main arguments is based on a second order differential computation derived in Delarue and Tse. However, I was not able to understand the tools used by the authors and how they use them to show Theorem 2 in the main paper. Finally, I think the authors should really comment more on what their contributions differ from those of Delarue and Tse.
- I did not follow how the smoothness of $\Phi$ implies that $\Phi(\mu) - \Phi(\mu^*) \leq C W_2(\mu,\mu^{*})$ (end of p6).
- the notation $\mu_t$ and $m(t,\mu_0)$ is really confusing.
- Finally, I was not been able to recover the informal statement of the authors:
$$
\mathbb{E}[(f_{X_t}-f_{\mu_t})^2] = O(1/N).
$$
In my opinion, it should be
$$
\vert \mathbb{E}[f_{X_t}-f_{\mu_t}] \vert = O(1/N).
$$


**Strength And Weaknesses:**

Strength:
- the result is mathematically interesting and brings some new tools for establishing uniform propagation of chaos for particle systems which emerged from overparametrized neural network optimization.

Weakness:
- I am not really sure how interesting this result is to the machine learning community. I think the authors should better motivate the question they are addressing and provide some insights why their results may help in understanding of the optimization of overparametreized neural networks.  Ideally, could the authors give practical implications of their findings?
- The condition on the regularization term is not really satisfactory. In particular, the authors say that previous works were limited by assuming either weak interaction or large noise. To my view, the authors get rid of this constraint by imposing a very strong regularization term which always counterbalances the interaction drift. This choice could potentially miss important and interesting phase-transition phenomena.
- I felt that some parts of the paper would needed to be rewritten and  proofread if the authors want that their paper is accessible to the machine learning community.

**Summary Of The Paper:**

This paper considers the mean-field regime of an overparametrized two layers neural network $f_X$ where $X\in\mathbb{R}^p$ stands for the trainable parameters. Indeed, it have been shown previously taking the number of neurons $N\to \infty$ that optimizing such network can be seen as minimizing the functional over the space of probability laws
$$
F(\mu)=\frac{1}{n} \sum_{i=1}^n \ell_i\left(f_\mu\right)+\lambda_1 \int r(x) \mathrm{d} \mu(x) \quad , \quad f_\mu(z)=\int f_x(z) \mathrm{d} \mu(x).
$$
In particular, the paper aims to extend and complete existing literature regarding convergence of the neurons following the continuous limit of the "noisy" gradient descent dynamics to a mean-field Langevin equation which is the gradient flow dynamics associated with $F$.
More precisely, the authors consider the particle system

$$
\mathrm{d} \hat{X}_t^i=-b(\hat{X}_t^i, \mu_t^N ) \mathrm{d} t +\sqrt{2 \lambda} \mathrm{d} W_t^i
$$

$$
\mu_t^N=\frac{1}{N} \sum_{i=1}^N \delta_{\hat{X}_t^i},
$$

where $W_t^i$ are independent Brownian motion and $b$ corresponds to the gradient of the first variation $F$
$$
b(x, \mu)=\nabla \frac{\delta F}{\delta \mu}(\mu)(x)=\frac{1}{n} \sum_{j=1}^n \ell_j^{\prime}\left(f_\mu\right) \nabla h_j(x)+\lambda_1 \nabla r(x).
$$

They then show that for a nice class of functional $\Phi$, a uniform propagation holds for the sequence of dynamics $(\mu_t^N)$: there exist $C_1,C_2,C_3$ such that for any $t \geq 0$ and $N \geq 1$,
$$
|\mathbb{E}\left[\Phi\left(\mu_t^N\right)\right]-\Phi\left(\mu^*\right) |\leq C_1 e^{C_2t} + C_3/N.
$$

where $\mu^*$ is the minimizer of
$$
\mathcal{L}(\mu)=\frac{1}{n} \sum_{i=1}^n \ell_i\left(f_\mu\right)+\lambda \mathrm{KL}\left(\mu, \nu_r\right),
$$

and the limit of the mean-field Langevin dynamics
\begin{aligned}
&\mathrm{d} X_t=-b\left(X_t, \mu_t\right) \mathrm{d} t+\sqrt{2 \lambda} \mathrm{d} W_t, \\
&\mu_t=\operatorname{Law}\left(X_t\right).
\end{aligned}

**Summary Of The Review:**

Overall I think that it could be a nice contribution to the literature on propagation of chaos.

However, in my opinion, the writing of the paper has to be improved and the relevance of the contributions for the machine community should be better emphasized.

Second, I do not really know if ICLR is a good venue for the paper which I think should be suitable for more theoretical conferences or journals. Especially because the proofs of the results are quite delicate and require a relatively long time to be verified. To be honest, I did not have the time to do it and I am afraid no reviewer has either.

---

> ### Author Response · Authors · 2022-11-13
> **Reply to Reviewer FGcq**
>
> Thank you for the helpful feedback. We address the technical points below.
>
> **"To my view, the authors get rid of this constraint by imposing a very strong regularization term which always counterbalances the interaction drift."**
>
> We would like to clarify this major misunderstanding. Crucially, our result holds for regularization parameters that can be *arbitrary small*; hence our assumption is not really about the strength of regularization.
> Consequently, in our setting the regularization term *does not overwhelm the interaction*. Indeed, difference between the quadratic and super-quadratic regularizations appears in the region that is far away from the origin; whereas within a radius of the origin, the potential is non-convex (which is the main difficulty to prove convergence) and the interaction between neurons is certainly not weak.
> This is in stark contrast with existing results that only handle weak interactions by imposing strong regularization so that the potential becomes (nearly) convex, in which path-wise convergence can be shown (i.e., discrepancy between two paths $(X_t)_t$ with slightly different initial values $X_0$ converges to 0 exponentially).
> Instead, we can only make use of the convergence of $\mathrm{Law}(X_t)$ due to this non-convexity.
>
> **"The authors should comment more on what their contributions differ from those of Delarue and Tse"**
>
> While our uniform propagation of chaos computation is based on the technical tools developed in Delarue and Tse (2021), the proof direction is completely different. Noticeably, their analysis requires a *small correlation assumption* which cannot be guaranteed in our setting (such small correlation assumption is not necessarily satisfied even for quite small $\lambda$).
> Therefore, we need to develop a novel device to circumvent this difficulty. Our strategy is to utilize the super log-Sobolev inequality via a super-quadratic regularization, which ensures a density ratio condition (Corollary 1) and hence the LSI on $\mu_t$. Such LSI condition along the trajectory is challenging to prove, and to our knowledge our result is the *first* to achieve this in the mean-field neural network setting. We have discussed this aspect in the revise Section 5.
>
> Moreover, even after we can establish the LSI on $\mu_t$, it is still technically challenging to show the convergence of the terms $d^{(1)}$ and $d^{(2)}$ without the weak correlation condition. We solve this problem by carefully constructing a potential function and making use of the Poincaré inequality. This proof strategy is completely different from existing work such as Delarue and Tse (2021).
>
> **"Could the authors give practical implications of their findings?"**
>
> We believe our uniform propagation of chaos result is an important step towards quantitative understanding of neural networks in the presence of feature/representation learning. As mentioned in the Introduction, a fundamental question in the deep learning literature is: *why can neural networks be optimized efficiently?*
> While the neural tangent kernel approach has been well-studied and finite-width results are easy to obtain, this regime fails to capture the feature learning dynamics. On the other hand, the mean-field regime allows for rich feature learning behavior, but quantitative finite-width guarantee is much more challenging to establish.
> Therefore, the propagation of chaos, a.k.a. the discretization error analysis, is a fundamental question in neural network theory yet to be resolved.
>
> To give a concrete example of the potential applications, [Abbe et al. 2022] studied functions that can be efficiently learned by mean-field neural networks but not kernel methods (NTK). However, due to the absence of uniform propagation of chaos result, they can only consider the setting where the input dimensionality $d$ is large but gradient descent is perform for *constant* time (that does not grow with $d$).
> This short time-horizon significantly limits the class of functions that can be learned by neural network; for example, even a simple quadratic target function may require $t=\Theta(\text{log} d)$ which falls beyond scope.
> Equipped with a uniform propagation of chaos analysis, we can now theoretically study neural networks that are trained much longer. This opens up new research directions in deep learning theory; for instance, we may try to establish learning separation between neural networks and kernel methods for a much wider class of functions.
>
> Therefore, we believe that our result is highly relevant to the ICLR community.
>
> Abbe et al., 2022. *The merged-staircase property: a necessary and nearly sufficient condition for SGD learning of sparse functions on two-layer neural networks.*

---

> > ### Author Response · Authors · 2022-11-13
> > **Reply to Reviewer FGcq (continued)**
> >
> > **"Why referring particularly to Huang for existence and uniqueness of solutions for (3)"**
> >
> > We presented the dynamics just for brief overview of the problem setting and deferred the detailed assumptions to the section of the theoretical analysis for concise presentation. We cited Huang et al. (2021) because the assumptions in their paper cover our setting where super-quadratic regularization is employed.
> > Whereas other prior results such as Hu et al. (2019) often assumed Lipschitz continuity on the gradient of the potential function which excludes our tail growth condition.
> >
> > **How to show** $\Phi(\mu) - \Phi(\mu^*) \le CW_2(\mu,\mu^*)$
> >
> > Thank you for the close reading. By the Benamou–Brenier formula, for any $\nu_1,\nu_0 \in \mathcal{P}_2$,
> > it holds that
> > \begin{align*}
> > W_2(\nu_1,\nu_0) = \inf \Big(\int_0^1 \int \|v_t\|^2 \mathrm{d}\nu_t \mathrm{d}t \mid \partial_t \nu_t + \nabla \cdot (\nu_t v_t) = 0 \Big).
> > \end{align*}
> > Then, for any $v_t$ satisfying $\partial_t \nu_t + \nabla \cdot (\nu_t v_t) = 0 $,
> > the chain rule (Chapter 10 of Ambrosio et al. (2005)) yields that
> > \begin{align*}
> > |\Psi(\nu_0) - \Psi(\nu_1)| \\
> > = \left|\int_0^1 \int v_t^\top \nabla \frac{\delta \Psi}{\delta \mu}(\nu_t) \mathrm{d}\nu_t \mathrm{d} t\right| \\
> > \leq
> > \left|\int_0^1 \int \left\|v_t\right\| \left\|\nabla \frac{\delta \Psi}{\delta \mu}(\nu_t)\right\| \mathrm{d} \nu_t \mathrm{d} t\right| \\
> > \leq
> > C \sqrt{\int_0^1 \int \|v_t\|^2  \mathrm{d} \nu_t \mathrm{d} t }.
> > \end{align*}
> > By taking infimum with respect to $v_t$, we obtain the inequality. We have included this auxiliary lemma in Appendix D.
> >
> > **Condition on regularization term.**
> >
> > We note that our assumption on the regularizer can be easily extended $c \|x\|^{2+\delta} \leq r \leq C (1+\|x\|^{2+\delta'})$ for different $\delta,\delta' >0$. We employed $\delta = \delta'$ just for simple presentation. Hence its choice is much wider than $\|x\|^{2\alpha}$.
> > We have added a remark on this in the main text.
> >
> > **Why is $\mathbb{E}[(f_X - f_{\mu_t})^2] = \mathcal{O}(1/N)$.**
> >
> > Thanks again for checking the details. This assertion can be verified via a direct computation which we included in the updated Appendix D.  We have also revised the informal statement in more rigorous way.
> >
> > **Probability law.**
> >
> > In probability theory, the distribution of a random variable $X$ is also called the "probability law". Indeed, the convergence in distribution is also called convergence in law. Therefore, we think it is natural to call $\mu$ the probability law of $X$.
> >
> > **"introduce one of $\lambda$ and $\lambda_1$ since you work with a generic $r$."**
> >
> > Indeed we can absorb $\lambda_1$ into the regularization function $r$. Here we follow the convention in [Nitanda et al. 2021] and write explicitly $\lambda$ in front of $r$, because it becomes easier for readers to identify where the regularization strength appears in the LSI constant and convergence rate.
> >
> > We would be happy to clarify any concerns or answer any questions that may come up during the discussion period.

---

> ### Author Response · Authors · 2022-11-17
> **Follow up**
>
> We hope the reviewer has looked at our reply. We addressed all of your technical questions which, we believe, would resolve your concerns.
> If this is the case, please consider updating your evaluation. Otherwise, we would be happy to address any further concerns before the end of the discussion period.

---

### Official Review · Reviewer_tT78 · 2022-10-23

**Confidence:** 3
**Clarity, Quality, Novelty And Reproducibility:** The paper is generally clear. The res…
**Correctness:** 3
**Technical Novelty And Significance:** 4
**Empirical Novelty And Significance:** Not applicable
**Recommendation:** 6

**Strength And Weaknesses:**

This paper is the first I have known of to prove a uniform-in-time type of result for MF neural networks. This is a good result, since there is currently no applicable tool from the uniform-in-time chaos literature. Uniformity in time is also a highly desirable property, since one drawback of typical MF analyses is that the approximation between the large-width neural nets and the MF limit blows up exponentially with time. The observation that $\mu_t$ satisfies LSI thanks to the super-quadratic regularizer, leading to Theorem 2, is an interesting one. The work also circumvents the trouble of doing the large-$N$ approximation at each time $t$ by going for a large (ideally infinite) $t$, hence allowing for the error decomposition as well as the exponential convergence result that was proven by previous works.

I do not have much to complain about the paper, nor do I have sufficient time and background to check the proofs. Of course, there are typical concerns about the (potentially bad) dependency among the constants, as well as the rather unusually strong regularizer, but I would not worry much about it for now in light of a new result, the first of its kind. I have two questions:

- The numerical illustration basically shows that the approximation is increasingly better for larger $N$, but it doesn’t quite show that the approximation is getting better / no worse with time. Is there a way to better illustrate this?

- The strong tail of the regularizer essentially shapes the curvature at infinity and intuitively prevents bad things to happen at infinity. However the analysis centers around the optimal solution $\mu*$, for which the faraway region likely matters much less. Is there a way to intuitively understand this supposed dilemma? One possibility is that the regularizer might not be needed for uniform-in-time chaos.

Here my concern is that the strong regularizer is more of a technical device than something insightful, so there is some doubt whether this is the right way to understand uniform-in-time chaos.

**Summary Of The Paper:**

The paper proves uniform-in-time chaos in the context of mean-field (MF) Langevin dynamics, motivated by noisy gradient descent for 2-layer MF neural nets. The analysis exploits the fact that a super log-Sobolev inequality (LSI) for the so-called proximal Gibbs measure, enabled by a super-quadratic regularizer, translates to the LSI of the corresponding flow measure at any sufficiently large time.

**Summary Of The Review:**

The paper proves the first uniform-in-time chaos result ever known for MF 2-layer neural nets. This is a good result with new technical insights, though I’m unable to go through the proofs. I would expect researchers in the area take serious interests in this result, even if the strong regularizer does not appear convincing to be the right way to understanding the phenomenon.

---

> ### Author Response · Authors · 2022-11-13
> **Reply to Reviewer tT78**
>
> Thank you for the thoughtful comments and for precisely understanding our contributions. We address the technical points below.
>
> **"The strong tail of the regularizer essentially shapes the curvature at infinity and intuitively prevents bad things to happen at infinity ... Is there a way to intuitively understand this supposed dilemma?"**
>
> At a high level, we agree that the tail growth condition handles "faraway region" that might not matter. This being said, we highlight that LSI is rather subtle and cannot be reduced to simple tail conditions. For example, it is known that a "quadratic" tail condition such as sub-Gaussianity does not imply LSI.
> In our uniform-in-time evaluation, we require the trajectory $\mu_t$ to satisfy the LSI -- this assumption also appeared in prior works on uniform propagation of chaos such as [Lacker and Flem 2022], and is very challenging to establish.
> We have highlighted this point in Section 5 of the revised manuscript.
>
> Our approach to overcome this difficulty is to assume super-quadratic inequality, which is relatively weak since the regularizer can be sufficiently close to the quadratic function, and we do not impose any constraint on the strength of the regularization.
> With such an assumption, we are able to establish a density-ratio bound which then implies LSI for $\mu_t$ (Corollary 1).
> Please refer to our general comment for more details.
>
> As a side remark, in terms of future directions for improvement, we may consider the Poincaré inequality, which is an isoperimetric condition weaker than LSI but also sufficient for our purpose. However, we think that this also require appropriate regularization;  establishing a uniform propagation of chaos without any isoperimetry conditions on the trajectory remains very challenging.
>
> **"numerical illustration... doesn’t quite show that the approximation is getting better/no worse with time"**
>
> Note that we do not show that the approximation error gets better in time. Our result only implies that even if the time goes to infinity, the approximation error remains of order $\mathcal{O}(1/N)$ and does not blow up.
> Therefore, it is natural that the numerical experiments do not show monotonic decrease of the approximation error through time.
>
> We would be happy to clarify any concerns or answer any questions that may come up during the discussion period.

---

> ### Comment · Reviewer_tT78 · 2022-12-08
> **Thanks**
>
> Thanks for the reply!
>
> I would encourage the authors to have a broader view than the technicality (LSI and the likes). Any weakening of the assumption on the regularizer is technically interesting, but does not seem to hold meaning. Again the proof only exploits the regularizer without caring about the main loss term, but conceptually it is the main loss term that matters most at convergence.

---

> > ### Author Response · Authors · 2022-12-08
> > **Reply to Followup Comment**
> >
> > Thank you for the followup comment.
> > In our revision, we will include a broader discussion on the implications of our propagation of chaos result. In brief, a uniform-in-time control of the particle discretization error allows us to run the mean-field dynamics for longer time; consequently, we may show learnability of a larger class of functions by finite-width neural network, and establish a stronger separation against linear estimators such as kernel models (see our reply to reviewer FGcq for more details).
> >
> > Regarding the technical proof, while our convergence rate does require non-vanishing regularizations, it is not the case that structure of the main loss term is dismissed. Similar to prior mean-field analyses, we exploited the *convexity* of loss function to prove convergence (for example in the "entropy sandwich" argument).

---

### Official Review · Reviewer_k2b8 · 2022-10-24

**Confidence:** 3
**Correctness:** 3
**Technical Novelty And Significance:** 3
**Empirical Novelty And Significance:** Not applicable
**Recommendation:** 6

**Clarity, Quality, Novelty And Reproducibility:**

Clarity: The presentation of this paper is overall clearly written. Section 4.1.2 is difficult to follow.

Novelty: While this paper achieves the uniform-in-time type control between the mean-field dynamics and its finite-particle counterpart without adding strong entropy regularization or imposing strong assumptions on the complicated interaction term, the super-quadratic tail of the convex regularizer seems strong to me. I was not able to go through the details of the proof and I believe it is not easy. However, a strong regularization basically confines the parameter in a compact space (especially the gradients are assumed to be bounded) and LSI naturally holds in that case. I am not certain how meaningful this result is.



**Strength And Weaknesses:**

Strength: This paper provides the first rigorous uniform-in-time propagation of chaos result in the context of mean-field neural networks. This is achieved without adding strong entropy regularization or imposing strong assumptions on the complicated interaction term.

Weakness: To achieve the claimed result, the convex regularizer on parameters is assumed to have a super-quadratic tail, an assumption that is not satisfied for the most common $\ell_2$ regularizer.

Some minor comments:
1. Below the equation after Eq.(4), there is a statement "We therefore see that $\mu_t$ decreases ... $\neq 0$.". It is difficult to understand.
2. In the paragraph of "Particle discretization" on page 4, it is assume that $\hat X_0^i \sim \mu_0$, but in Lemma 1, it is claimed that you replaced the KL divergence on the R.H.S. with the Wasserstein distance since $\mu_0$ is a discrete distribution. Why can't we simply take $\mu_0$ to be some simple continuous initial distribution?
3. At the end of Proposition 1, I guess it should be $\mu^* = p_{\mu^*}$.
4. The concept of "test function" used in section 4.1 is not the same as the ones commonly used for defining weak convergences.

**Summary Of The Paper:**

This paper studies the convergence of the mean-field gradient Langevin dynamics for the regularized mean-field neural network model. Under certain regularity assumption, it is proved that the dynamics in mean-field limit and the finite particle approximation are close for all time t>0, and certain statistic of these two dynamics are close with the order of O(1/N). As a consequence, it can be show that the finite particle approximation of the mean-field dynamics exponentially convergence towards the optimal solution up to an error of O(1/N).

**Summary Of The Review:**

This paper shows that the dynamics in mean-field limit and the finite particle approximation are close for all time t>0, and certain statistic of these two dynamics are close with the order of O(1/N). This result is interesting, but is obtained under a relatively strong assumptions on the parameter regularizer.

---

> ### Author Response · Authors · 2022-11-13
> **Reply to Reviewer k2b8**
>
> Thank you for the thoughtful feedback. We address the technical comments below.
>
> **"Why can't we simply take $\mu_0$ to be some simple continuous initial distribution?"**
>
> Indeed we may derive Theorem 1 with a simple absolutely continuous distribution $\mu_0$. However, if we do so, there appears an additional discretization error of the initial distribution, i.e., the distance between $\mu_0$ and $\mu^N_0$, which introduces an additional error term. In general, this discretization error is of order $W_2(\mu_0, \mu^N_0) = O(N^{-1/d})$, which is strongly affected by the dimensionality $d$. To avoid this large deviation, we choose to directly take a discrete distribution as the initial distribution.
>
> **"A strong regularization basically confines the parameter in a compact space ... and LSI naturally holds in that case"**
>
> In our uniform-in-time evaluation, we require the trajectory $\mu_t$ to satisfy the LSI -- this assumption also appeared in prior works on uniform propagation of chaos such as [Lacker and Flem 2022].
> Importantly, while it is easy to verify that the Gibbs measure $\tilde{\mu}_t$ and the optimal $\mu^*$ satisfy the LSI, it is very challenging to prove that $\mu_t$ also enjoys good isoperimetry (in fact, this is the only place we used the strong tail growth condition; we have highlighted this point in Section 5 of the revised manuscript). This is because the LSI requires careful control of $\mu_t$ (e.g., smoothness) beyond basic growth. For example, it is known that a "quadratic" tail condition such as sub-Gaussianity does not imply LSI.
> One of our technical contribution is to utilize a super LSI condition (entailed by our assumption on the regularization term) to control the density ratio between $\mu_t$ and $\tilde{\mu}_t$ (Corollary 1), which allows us to establish the LSI for $\mu_t$.
>
> We also highlight that our result holds for any regularization parameter $\lambda,\lambda_1$. Thus, in a rigorous sense, we are not considering a "strong" regularization but only imposing a tail growth condition -- this contrasts existing results on uniform propagation of chaos that are only valid under strong regularizations (as remarked in the Introduction).
> Please refer to our general comment for more details.
>
> **"Below the equation after Eq.(4), there is a statement 'We therefore see that  decreases ... .'. It is difficult to understand"**
>
> Thank you for pointing this out. There was typo in the sentence. The correct statement is "$\mu_t$ decreases $\mathcal{L}(\mu_t)$ unless $\frac{\delta \mathcal{L}(\mu)}{\delta \mu}= 0$."
>
> "**At the end of Proposition 1, I guess it should be** $\mu^* = p_{\mu^*}$"
>
> Here we used the notation $\mu^*$ to indicate the probability measure instead of the density function, and then we denote the density function of $\mu^*$ by $\mathrm{d}\mu^*/\mathrm{d}x$. We have modified this sentence to avoid confusion.
>
> **"The concept of "test function" used in section 4.1 is not the same as the ones commonly used for defining weak convergences"**
>
> Thank you for the close reading. We have changed the terminology as "objective function" to describe $\Phi$.
>
> We would be happy to clarify any concerns or answer any questions that may come up during the discussion period.

---

### Official Review · Reviewer_LXtr · 2022-10-26

**Confidence:** 1
**Correctness:** 3
**Technical Novelty And Significance:** 4
**Empirical Novelty And Significance:** 2
**Recommendation:** 6

**Clarity, Quality, Novelty And Reproducibility:**

The authors clearly stated the contributions and limitations of the proof.

Novelty: I am not an expert in neural network theory and I am not able to evaluate the novelty.

The technical tools based on gradient flow stuff seem standard and reasonable.

Typos: $\mu_t$ decreases $\mathcal{L}(\mu_t)$ unless $\frac{\delta \mathcal{L}(\mu)}{\delta\mu}=0$?

**Strength And Weaknesses:**

**Pros:**
1. The first quantitative discretization error ensures analysis based on neural networks of finite/ limited neurons. The uniform nature ensures that the deviation of the output reduces rapidly.
2. The analysis leverages the advantages of propagation of chaos and overcomes the bottleneck suffered by the Growall inequality in Mei'18 and the error bound remains stable when the time t is large.

**Cons:**
The assumption of the super-quadratic tail of the regularization term is strong indeed.



**Summary Of The Paper:**

This paper proposed a uniform-in-time analysis of the propagation of chaos for the mean-field Langevin dynamics to conduct neural network optimization. The authors avoid the double-loop structure and enable to deal with convergence guarantee based on finite-width neural network with vanilla noisy gradient descent algorithms.

**Summary Of The Review:**

Convergence analysis for stochastic gradient descent on a finite-width neural network where the upper bound is table w.r.t. time.

---

> ### Author Response · Authors · 2022-11-13
> **Reply to Reviewer LXtr**
>
> Thank you for your helpful comments. We address the technical comments below.
>
> **Assumption on the super-quadratic tail of regularization.**
>
> We agree that the super-quadratic regularization is the main limitation of our current result. In Section 5 of the revised manuscript, we provided more explanations on where this assumption is needed in our analysis.
> This being said, we note that the uniform-in-time propagation of chaos is a challenging mathematical problem in general, and some additional assumptions are usually required. We believe our contribution is significant, as it is the first uniform propagation of chaos result in the context of mean-field Langevin dynamics for neural network optimization.
> Despite requiring a rapid tail growth, our problem setting does not impose convexity around the origin; and importantly, our characterization holds for any regularization strength, which is much different from existing researches that assumed large noise / weak interaction conditions (as remarked in the Introduction and Appendix A).
> Please refer to our general comment for more details.
>
>
> **Typos.**
>
> Thank you for pointing this out. Indeed the correct statement is ``unless $\frac{\delta \mathcal{L}(\mu)}{\delta \mu}= 0$.''
>
>
> We would be happy to clarify any concerns or answer any questions that may come up during the discussion period.

---

### Author Response · Authors · 2022-11-13
**General Response**

We appreciate the reviewers' helpful feedback that helped us improve the paper. To best respond to the comments, we revised our paper with additional clarifying content as suggested by the reviewers.
The modifications are summarized as follows.

* We revised Section 5 to improve readability and highlight our technical contributions. Specifically, before Section 5.1 we included a discussion on the LSI, super quadratic regularization, and technical difficulties of the proof. In Section 5.2 we deferred Proposition 3 to the Appendix and added more explanations on the proof outline.
* For the Appendix, we re-organized the proof,  included more detailed derivations, and cleaned up potentially confusing notations.
* We corrected typos in the main text pointed out by the reviewers.

**On super-quadratic regularization**

More than one reviewer raised concerns regarding our assumed super-quadratic tail growth. Here we provide a general reply to clarify this assumption and explain its role in our proof. Indeed we agree that the super-quadratic regularization is the main limitation of
our current analysis, and it is possible that such assumption can be removed in the future. This being said, we believe our contribution is significant, as we provide the *first* uniform propagation of chaos result in the context of mean-field Langevin dynamics for neural network optimization, which is an important step towards quantitative understanding of neural networks in the presence of feature/representation learning.
We make the following remarks on the significance of our theoretical result.


* The uniform-in-time propagation of chaos is a challenging mathematical problem in general, and some additional assumptions are usually required. Note that despite requiring a rapid tail growth on the regularizer, our problem setting does not impose convexity around the origin.
More importantly, our characterization holds for *any regularization strength*, which does not imply large noise / weak interaction conditions often assumed in prior works (as remarked in the Introduction and Appendix A). Thus, in a rigorous sense, we are not considering a "strong" regularization but only imposing a tail growth condition – this contrasts existing results on uniform propagation of chaos that are only valid under strong regularizations.

* In terms of our proof strategy, we are able to show uniform-in-time propagation of chaos if the trajectory $\mu_t$ satisfies the
LSI  – this assumption also appeared in prior works on uniform propagation of chaos such as [Lacker and Flem 2022], and is very challenging to establish (see discussion in Section 5.2 of the revised manuscript for more details). This difficulty is partly because the LSI is a rather subtle condition and cannot be reduced to simple tail conditions; for example, it is known that a "quadratic" tail condition such as sub-Gaussianity does not imply LSI.
We overcame this difficulty by utilizing a *super LSI condition* (entailed by our assumption on the regularization term), which allows us to control density ratio between $\mu_t$ and $\tilde{\mu}_t$ (Corollary 1), and then translate the LSI to $\mu_t$ via the Holley-Stroock argument. We believe this derivation is novel as we are not aware of any similar characterizations in the context of mean-field neural networks (i.e., without assuming convexity or weak interactions).

Please refer to our detailed response to each reviewer where we address all individual comments.

Daniel Lacker and Luc Le Flem, 2022. *Sharp uniform-in-time propagation of chaos.*

---

### Decision · Program_Chairs · 2023-01-20

**Decision:**

Accept: poster

**Justification For Why Not Higher Score:**

While this is a very nice paper I did not see any particular reason for a highlight.



**Justification For Why Not Lower Score:**

No reason to reject.



**Metareview: Summary, Strengths And Weaknesses:**

Reviewers overall agree this paper should be accepted. I think the reviews summarize very well the strengths and weaknesses of the paper as well as points that the authors should include in the revised version. This paper is a valuable addition to the conference.

**Note From Pc:**

if the above contains the word "oral" or "spotlight" please see: "oral" presentation means -> notable-top-5% and "spotlight" means -> notable-top-25%. As stated in our emails, we are disassociating presentation type from AC recommendations